

# Increasing water stress in Chile evidenced by novel datasets of water availability, land use and water use

Juan Pablo Boisier[1,2,*], Camila Alvarez-Garreton[1,2,*], Rodrigo Marinao[1,3], Mauricio Galleguillos[1,4]

[1]Center for Climate and Resilience Research (CR2, FONDAP 1523A0002), Santiago, Chile
[2]Department of Geophysics, Universidad de Chile, Santiago, Chile
[3]Department of Civil Engineering, Universidad de La Frontera, Temuco, Chile
[4]Faculty of Engineering and Science, Universidad Adolfo Ibáñez, Santiago, Chile

*Equal contribution

*Correspondence to*:  Camila Alvarez-Garreton (calvarezgarreton@gmail.com)

**Abstract.** Many regions in Chile experienced an unprecedented drought from 2010 to 2022, driven by climate change and natural variability. This so-called megadrought led to severe water scarcity, causing conflicts and exposing issues in Chilean water regulations. Water-intensive agriculture in areas with limited water availability has worsened these problems, raising
questions about the contributions of water extraction and climate on high water stress levels.

In this study, we evaluate water stress in Chile over the long term, from the mid-20th century to the end of the 21st century, under various climate and socio-economic scenarios. To this end, novel datasets of water availability, land use and water use were developed. Using these, we calculated the Water Stress Index (WSI) for all major basins in the country and assessed the impact of increasing water use and climate change on water stress over different time periods. Results show that most basins
in semi-arid regions experienced high to extreme water stress (WSI > 40% and WSI > 70%, respectively) during the megadrought, mainly due to reduced water availability, but worsened by high water demand. Over time, increasing water stress in central Chile is primarily linked to rising water consumption, with a smaller contribution from water availability changes, leading to consistently high water stress levels (1990-2020 average) in several basins from Santiago northward. Under an adverse climate scenario (SSP3-7.0), megadrought-like conditions could become permanent by the end of the 21st century,
with a projected 30% drop in precipitation, resulting in high to extreme water stress in most basins in central Chile. We argue that using the WSI to assess one of the several aspects of water security offers a valuable strategy for adaptation plans. If public policy agrees on establishing quantifiable water security goals based on metrics like the WSI, different pathways of water use combined with alternative water sources can be evaluated to achieve them.





## 1 Introduction

Water security is defined as the capacity of a population to safeguard access to sufficient quantities of water of acceptable quality for sustaining livelihoods, promoting human well-being, fostering socio-economic development, and preserving ecosystems. Achieving water security is part of the Sustainable Development Goals adopted by all United Nations member
states in 2015 (UNESCO, 2019), and many countries have incorporated this goal into their national development plans (High-Level Political Forum on Sustainable Development, https://hlpf.un.org/countries, last access: 20 August 2024). Pursuing this goal requires understanding the climate system and change, its regional effects, and its interaction with local human activities. Then, effective governance and infrastructure are essential to set water security goals and implement actions to achieve them.

There are various ways to assess water security in a territory, with methodologies varying in complexity based on the factors
considered. A common and straightforward approach is to contrast water use and water availability estimates at the basin scale. A basin is deemed to have high water stress when the Water Stress Index (WSI) –the ratio of water use to availability– exceeds 40% over the medium term (5 to 10 years). A high WSI indicates low water security, as it increases the risk of water scarcity (Falkenmark and Lundqvist, 1998; Vörösmarty et al., 2000; Oki and Kanae, 2006). While assessing water security using the WSI does not account for factors such as governance and accessibility, it serves as a baseline for more in-depth evaluations.
Studies that quantify the WSI rely on robust estimates of water availability and use (Gain et al., 2016; Liu et al., 2017; Oki and Kanae, 2006; Wada et al., 2011). However, obtaining accurate information can be challenging due to the poor quality and limited availability of data and ground observations in some regions (Condom et al., 2020; IPCC, 2022).

Quantifying water availability presents methodological challenges due to the complex interaction of climate with landscape across regions. In South America, the Andes Cordillera makes these estimates particularly challenging due its intricate
topography, which affects atmospheric motions and precipitation patterns that global models struggle to capture (Arias et al., 2021; Espinoza et al., 2020). Water use estimates typically rely on national inventories, which are often incomplete, unavailable, or even non-existent. As a result, these datasets can carry significant uncertainties in some regions, as noted by Wada and Bierkens (2014) for South America.

In this study, we complement global water security assessments by implementing a WSI-based methodology in Chile,
identifying the main causes of its variations over the past decades and projecting future trends. Although the study focuses on Chile, the approach can be applied to any region that meets the data requirements (see Sect. 2). Moreover, Chile's diverse hydroclimate and landscape make the insights from this research applicable to other regions of the world. The country spans 4300 kilometers along the southern Andes (17°S−56°S), with altitudes reaching up to 6900 meters above sea level, and features a variety of extratropical climates and biomes, from the Atacama Desert to very humid regions with temperate rainforests in
the south. The country's central regions (30°S−40°S) are covered by a mosaic of different land uses, including major urban areas, agricultural lands in the valleys, exotic forest plantations in hilly regions with non-arable soils, and natural vegetation in the steepest areas. Natural and managed prairies for cattle are also abundant in the south-central regions and Patagonia.



Central Chile has experienced increasing water scarcity problems, leading to social and legal conflicts (e.g., Rivera et al., 2016). This situation is partly due to a long-term drying trend and decreased water availability in the region (Boisier et al., 2018), which has been exacerbated by a decade-long drought since 2010 (Boisier et al., 2016; Garreaud et al., 2017, 2019). Water scarcity issues have also been attributed to limitations in Chile's water management system, in which the legal framework for granting Water Use Rights does not account for climate variability and decreasing water availability (Alvarez-Garreton et al., 2023a; Barría et al., 2021a). Case studies have shown that water scarcity issues also relate to increased water withdrawal, notably from the agriculture sector, but with very contrasting results regarding the magnitude of the impacts of local water use and climate variability (e.g., Muñoz et al., 2020; Barría et al., 2021b; Valdés-Pineda, 2022). More consistently, recent research has shown that, despite lower surface available water, increasing water supply has been sustained by the overexploitation of groundwater, leading to continuous depletion of the water table in the major basins of central Chile (Jódar et al., 2023; Alvarez-Garreton et al., 2024; Taucare et al., 2024).

Despite the increasing evidence of intense water use in a region with decreasing precipitation rates, changes in water stress at the national level in Chile have not been thoroughly described, nor have the contributions of climate variability and water use to water security been widely quantified. Understanding the causes of water stress is critical for designing strategies aimed at mitigating unsustainable water uses and adapting to future climate conditions. In this study, we fill this gap by addressing the following research questions: What have been the historical water stress levels of the basins in Chile? What have been the drivers of changes in these levels? What can be expected under future climate scenarios?

To address these questions, we developed national-scale data products on freshwater availability, land use, and water use, enabling a consistent assessment of water stress from the mid-20th century to projections for the 21st century. This new knowledge can serve as a concrete input for designing climate-resilient strategies to achieve water security. To illustrate this, at the end of the manuscript, we consider a specific goal for maximum water stress to be achieved by 2050 (a moderate level of water stress, or WSI below 40%) and analyze different scenarios to reach that goal, taking into account various climate change projections, water use trends, and alternative sources of water availability.

## 2 Data products and methods

The results presented in this paper are based on recently developed, spatially distributed datasets of variables that are essential for assessing water stress over several decades and across all watersheds in Chile (outlined in Table 1). Climate information includes both observation-based data to evaluate historical changes, and model-based data for scenario projections (Sect. 2.1 and 2.2). Water availability is derived from a water balance rationale, using an estimation of evapotranspiration under near-natural conditions, described in Sect. 2.3 and Appendix B. Based on a consistent approach, reconstructions of land use/cover and water use are described in Sect. 2.4, Sect. 2.5, Appendix B and Appendix C. In Sect. 2.6, we describe the computation of the Water Stress Index (WSI) and the method used to attribute its evolution over time to climate and water use changes.



**Table 1: Main datasets used in this study.**

| Dataset | Variables[a] | Domain & resolution | Description | Data repository |
|---|---|---|---|---|
| Observations-based hydroclimate (CR2MET) | $P$, $T_N$, $T_X$, $ET_0$, $ET_N$ | Chile, 1960-present, daily, 0.05° lat-lon | Meteorological variables based on statistical downscaling of ERA5 (DGA, 2022a). $ET_0$, $ET_N$ are post processed. | https://doi.org/10.5281/zenodo.7529682 (Boisier, 2023) |
| Model-based hydroclimate | $P$, $T_N$, $T_X$, $ET_0$, $ET_N$ | Chile, 1960-2100, daily, 0.05° lat-lon | Statistical downscaling of 17 CMIP6 Earth System Models (Table A1). $ET_0$, $ET_N$ are post processed. | - |
| Land use/cover (CR2LUC) | $F_V$, $I_V$ | Chile, 1950-2020, yearly, 0.01° lat-lon | Land use and land cover reconstruction, including fractional cover and irrigation data. | https://doi.org/10.5281/zenodo.13324250 (Boisier et al., 2024a) |
| Water use (CR2WU) | $U_{CO}$, $U_{NC}$ | Chile, 1960-2020, yearly, commune, sectors | Consumptive and non-consumptive water use reconstruction (this study). | https://doi.org/10.5281/zenodo.13324235 (Boisier et al., 2024b) |

[a] Precipitation ($P$), daily minimum ($T_N$) and maximum ($T_X$) temperature, Potential evaporation ($ET_0$), naturalized actual evapotranspiration ($ET_N$), land-cover fraction ($F_V$), irrigation fraction ($I_V$) fraction, consumptive ($U_{CO}$) and non-consumptive ($U_{NC}$) water use.

## 2.1 Historical climate

Daily precipitation and temperature information were obtained from the Center for Climate and Resilience Research Meteorological dataset (CR2MET) version 2.5, available at https://doi.org/10.5281/zenodo.7529682 (Boisier, 2023). The
CR2MET product has been developed to fill key data gaps regarding the hydroclimatic mean regime and variability in Chile, and has significantly contributed to scientific and technical knowledge over the past years. In particular, the updated national water balance project led by the National Water Bureau (DGA) has been based on and has boosted the development of CR2MET (DGA, 2022a; and references therein). CR2MET has been used in a wide range of hydro-climatic research, including the development of the CAMELS-CL catchment dataset for Chile (Alvarez-Garreton et al., 2018), hydrological modeling
applications (Baez-Villanueva et al., 2021; Sepúlveda et al., 2022; Araya et al., 2023; Cortés-Salazar et al., 2023), the study of drought dynamics (Alvarez-Garreton et al., 2021; Baez-Villanueva et al., 2024), glacier dynamics (Ruiz Pereira and Veettil, 2019; Amann et al., 2022), water management assessment (Muñoz et al., 2020), and climate change studies (Gateño et al., 2024, Carrasco-Escaff et al., 2024).

The CR2MET dataset includes daily precipitation and diurnal maximum/minimum near-surface temperatures on a regular
0.05° latitude-longitude grid over continental Chile, covering the period from 1960 to the present. These variables are partially built upon a downscaling of the European Centre for Medium-Range Weather Forecasts (ECMWF) ERA5 reanalysis (Hersbach et al., 2020), which uses statistical models calibrated against large, quality-controlled records for the corresponding variables ($P$, $T_N$, $T_X$). These records gather observations from various national agencies, including DGA, the Weather Service



(DMC), the Army Weather Service (SERVIMET-DIRECTEMAR), the Agricultural Research Institute (INIA), and the
Foundation for Fruit Development (FDF). This observational dataset is also used in this study for comparison with CR2MET
in Sect. 3. The CR2MET statistical models use several atmospheric variables as predictors, selected for the specific predictand
(e.g., moisture fluxes are included for P), in addition to land-surface temperature estimates from the Moderate Resolution
Imaging Spectroradiometer (MODIS) satellite sensor (Hulley and Hook, 2021) for $T_N$ and $T_X$. The predictor variables are
combined with invariant local topographic features to achieve the target resolution. A detailed description and validation of
CR2MET are provided in DGA (2022a) and references therein.

For this study, potential evapotranspiration ($ET_0$) is derived as a post-processed CR2MET variable over the same
spatiotemporal domain. Due to data availability from CR2MET and downscaled climate models (Sect. 2.2), the method is
based on the temperature-dependent Hargreaves-Samani (HS) formula (Hargreaves and Samani, 1985), which is then corrected
using two criteria: (1) accounting for systematic biases between the HS estimate and the more precise, multivariate Penman-
Monteith (PM) formula (e.g., Allen et al., 1998), and (2) accounting for mean biases in wind conditions. For a given grid cell
($x$) and day ($t$), the adjusted $ET_0$ is computed as follows:

$$ET_0(x,t) = A(x,t_0)\, ET_{0,HS}(x,t) + B(x,t_0) \tag{1}$$

where $ET_{0,HS}$ is the HS estimate as described by Samani (2000). The formula is used with $T_N$ and $T_X$ from CR2MET and daily
mean temperature calculated as the average of these two variables. Coefficient A is computed as the climatological mean
(1981-2015) PM to HS-based $ET_0$ ratio ($ET_{0,PM}/ET_{0,HS}$), using a consistent set of atmospheric variables; in this case, the ERA5
reanalysis. The ERA5 HS $ET_0$ is calculated in the same manner as that derived with CR2MET, but with the corresponding
$T_N/T_X$ data. The ERA5 PM-based $ET_0$ corresponds to the Singer et al. (2021) dataset. It should be noted that the use of factor
A assumes that PM/HS corrections, valid —by construction— for ERA5 data, also apply to CR2MET data, likely introducing
some inaccuracies. Coefficient B addresses a wind bias correction, based on the comparison of near-surface wind speed data
from ERA5 (consistent with the PM to HS correction applied with coefficient A) and a source with more spatial detail than
ERA5. In this case, we used 1-km simulations (aggregated to the 0.05° CR2MET grid) conducted in Chile with the Weather
Research and Forecasting (WRF) model for the Wind Energy Explorer (https://eolico.minenergia.cl/; see Muñoz et al., 2018).
This correction is based on simplified aerodynamic expressions typically used to estimate open water evaporation, which scales
with the air vapor pressure deficit (VPD) and a coefficient depending on wind speed (the so-called wind functions; e.g.,
Penman, 1948). We used the wind function parameterization described by McJannet et al. (2012) and VPD from ERA5 to
compute B as 0.167 ($U_{WRF} - U_{ERA5}$) VPD. Here, $U_{WRF}$ and $U_{ERA5}$ correspond to the climatological mean 2-meter wind speed
from the WRF and ERA5, respectively. Both coefficients A and B are spatially distributed and computed separately for each
month of the year, and then interpolated to Julian days ($t_0$ in Eq. 1).



## 2.2 Climate model downscaling

To assess the impacts of anthropogenic climate change on water availability and water stress, we used an ensemble of Earth System Model (ESM) simulations from the sixth phase of the Coupled Model Intercomparison Project (CMIP6). The ESM data were bias-corrected for the period 1960-2100 to accurately represent Chile's hydroclimate and properly contrast with observation-based data.

The applied bias correction method is based on the Quantile Delta Mapping (QDM) approach as described by Cannon et al.
(2015), using CR2MET as the reference dataset. Unlike conventional quantile mapping, the QDM method adjusts the frequency distribution of a given variable to match a reference (typically an observational record or an observations-based time series) while preserving the modeled changes in the variable's quantiles over time. The method was applied on a daily time scale to precipitation outputs from 17 ESMs and to a subset of 11 ESMs for daily extreme temperatures, according to data availability at the time of the analysis (Table A1). Simulations include historical runs and future projections forced by two
socio-economic scenarios: Shared Socioeconomic Pathways (SSPs) 1 and 3 (Global Sustainability and Regional Rivalry), respectively paired with carbon emissions pathways leading to radiative forcing levels of 2.6 and 7.0 W m-2 by 2100 (Eyring et al., 2016; O'Neill et al., 2016). These two scenarios, SSP1-2.6 and SSP3-7.0, represent high and medium-to-low climate change mitigation efforts during the current century, with associated climate simulations available for a large ensemble of models.

The QDM computations were applied additively for absolute changes in $T_N$ and $T_X$, and multiplicatively for relative changes in P. The parametric density distributions Generalized Extreme Value and Exponential were used to fit the empirical frequency distributions of $T_N$/$T_X$ and P, respectively. To ensure accurate seasonal representation, the method was applied separately for each month of the year. Additionally, post-corrections were made to address any evident physical inconsistencies; for example, if $T_X$ was found to be less than TN, both variables were adjusted to their average value.

The downscaled temperature data is then used to estimate $ET_0$ in models using the same HS-based approach described in Sect. 2.2. Precipitation and $ET_0$ data both from in CR2MET and ESMs are used to estimate a naturalized evapotranspiration, as described in next section.

## 2.3 Hydrological balance and water availability

Water availability (A) is considered here as a naturalized runoff, that is, the remaining flow from precipitation (P) and
evapotranspiration (ET), without considering local disturbances. Thus, the surface water budget in a hydrological unit is represented as follows:

$$A = P - ET_N = R_N \qquad (2)$$



where $ET_N$ and $R_N$ stand for naturalized evapotranspiration and runoff. For simplicity, since A is assessed on a yearly to multi-year basis, changes in water storage are omitted in Eq. (2), but soil water content dynamic is accounted for in smaller-scale ET computation (Appendix B). The disturbed water budget is represented as:

$$A - \Delta ET_{LU} - U_{\sim LU} = R_N - \Delta R \tag{3}$$

where $\Delta ET_{LU}$ and $U_{\sim LU}$ indicate, respectively, modified ET due to land use/land cover changes (defined as positive for a perturbation increasing ET) and losses by other means (consumptive water use from other activities), both of which will induce a change in runoff ($\Delta R$).

Fluxes of ET were computed over continental Chile with the same spatiotemporal resolution as CR2MET, using a simplified 'bucket' ET scheme described in Appendix B. This model runs on daily time steps, forced by P and $ET_0$ from CR2MET or from CR2MET-based downscaled climate model data to maintain a consistency in observed and modeled data. The ET scheme also depends on soil and land cover parameters, and account for well-watered surfaces to represent water bodies and irrigated areas, allowing the estimation of both $ET_N$ and $\Delta ET_{LU}$.

For $ET_N$, simulations were performed using the CR2MET forcing between 1960 and 2020, with irrigation turned off and prescribing a static land-cover map of 1950, according to the CR2LUC product (Sect. 2.4). Thus, more precisely, $ET_N$ represents a climate-dependent ET flux under past land-cover conditions, without irrigation or temporal land use changes. Subsequently, annual time series of A for each basin were computed as the difference between catchment-scale annual P and $ET_N$ (Eq. 2). Projections of water availability following the *SSP1-2.6* and *SSP3-7.0* scenarios were derived in the same way but using the downscaled ESM data as climate forcing. As part of the water use reconstruction workflow, the computation of $\Delta ET_{LU}$ is described in Sect. 2.5.

**2.4 Land cover reconstruction**

Long-term land-use dynamics is a key factor to consider when estimating changes in water use over time and for many other applications. The CR2LUC product was developed for this end, describing land use and land cover changes in Chile since 1950. As it encompasses a pre-satellite period, this reconstruction was developed using a different approach from typical land use/cover estimates available for more recent decades. A dedicated paper describing the methodology and results of CR2LUC is under development at the time of writing this paper, so here we provide a general description of it.

Although satellite data are used for fine spatial distribution guidance, the reconstruction is based on national bookkeeping of land use activities in Chile, particularly agricultural censuses conducted since 1930. The methodological challenge in developing CR2LUC lies in the homogenization and coherent integration of information from different sources, as well as the data preprocessing of each source (e.g., some historical documents required data digitization/tabulation). The assessed set of inventoried data includes the Agricultural and Livestock Censuses carried out by the National Institute of Statistics (INE) for



the years 1955, 1965, 1976, 1997 and 2007 (see references in Table C1), the Continuous Statistics of annual crops from the Office of Agricultural Studies and Policies since 1979 (ODEPA, 2023b), the Fruit Cadastres maintained by the Natural
Resources Information Centre and ODEPA since 1998 (CIREN-ODEPA, 2023), the Viticultural Cadastres from the Agricultural and Livestock Service (SAG) since 1997 (ODEPA, 2023a), and the Vegetation Cadastres from the National Forestry Corporation, maintained since 1993 (CONAF, 2023).

The first stage of the CR2LUC workflow involves computing continuous time series describing annual quantities, primarily the fractional land area occupied by a given land cover class, for every commune in the country between 1950 and 2020,
ensuring alignment with data provided by agricultural censuses in specific years. To achieve this, solutions were developed to transform census data provided for different territorial units over time into a unified classification (current communes). In most cases, this was done by extrapolating proportional relationships in given variables between nested or related units (e.g., current communes vs. old provinces in Chile, or communes created from a larger, former one). Interpolation methods were used to fill in the time series, ranging from simple statistical criteria (e.g., linear interpolation) to guidance based on other sources,
depending on data availability in different periods (e.g., cadastres provided by ODEPA).

A finer spatial disaggregation (gridding) of the communal-level values was primarily driven by satellite information and some inventories available at high resolutions in recent years (e.g., fruit orchard cadastres), while ensuring consistency with aggregations provided at larger (territorial) unit scales. Cropland and urban areas were constrained by the corresponding classes in Landsat-based land cover datasets developed for Chile by Zhao et al. (2016) and globally by Chen et al. (2015), with the
latter including three scenes for 2000, 2010, and 2020. Vegetation cadastres of CONAF was the main input used to distribute natural vegetation classes and tree plantations, as well as to define a potential vegetation for past periods. Other inputs include fine national inventories (polygons) of water bodies and wetlands (MMA, 2020), saltpans (CEDEUS, 2015) and glaciers (DGA, 2022b).

As result, CR2LUC provides fractional land cover data on a 1-km latitude-longitude grid for continental Chile, with annual
data from 1950 to 2020. It also includes information on irrigation (grid fraction and type) and yields for specific crops. Organized into three levels of aggregation, the dataset includes a total of 48 classes of particular relevance in Chile (Table B1), encompassing various agricultural types (25), natural forests (5), planted trees (3), shrubland (3), grasslands (3), natural and artificial water bodies (5), salt pans, glaciers, built-up areas and bare soil. Hence, in addition to the temporal coverage, the dataset is characterized by a detailed classification. Version 1.0 of CR2LUC, used in this study, is available on
https://doi.org/10.5281/zenodo.13324250 (Boisier et al., 2024a).

## 2.5 Water uses reconstruction

The CR2WU product was developed to assess historical water demand in Chile. This dataset includes water uses as volumetric fluxes (U) from various sectors for each commune in continental Chile, with annual resolution from 1960 to 2020. The



computation involves two distinct methodologies: one for sectors encompassing land use, land-use change, and forestry
(LULUCF) and another for other water-consuming sectors.

For LULUCF sectors, water uses were derived using the ET scheme introduced in Sect. 2.3 (further described in Appendix B),
as the difference between alternative historical ET scenarios. One scenario represents the actual ET, following the historical
land use/cover, irrigation, and climate conditions ($ET_{FULL}$), and another –control– scenario represents ET with no land
use/cover changes, corresponding to $ET_N$ (Sect. 2.3). A third simulation was performed in the same way as $ET_{FULL}$ but without
irrigation ($ET_{NI}$), allowing the estimation of the ET change component driven by rainfed agriculture and forestry (the 'green'
water use). Hence, the consumptive total water use from LULUCF ($U_{CO,LU}$) and the non-irrigated component ($U_{CO,LU,NI}$) were
computed as:

$$U_{CO,LU}(yr) = \mathrm{ET}_{FULL}(yr) - ET_N(yr) \qquad (4)$$

$$U_{CO,LU,NI}(yr) = ET_{NI}(yr) - ET_N(yr) \qquad (5)$$

Irrigation fractions from CR2LUC is used to prescribe unlimited soil moisture in the corresponding grid cells and agricultural
classes, while the irrigation type is used to estimate the water supply typically needed for irrigation practices, a portion of
which returns to the system as infiltration (a non-consumptive use). Irrigation efficiency ($e_{IR}$), defined as the ratio between
$U_{CO,LU}$ and the total irrigation volume, was set to 0.5, 0.6, and 0.75 for the three types of irrigation included in CR2LUC:
traditional (e.g., furrow), mechanized (sprinkler), and localized, according to Brouwer et al. (1989). In this way, the non-
consumptive water use component of agriculture ($U_{NC,LU}$) was derived as:

$$U_{NC,LU}(yr) = (e_{IR}^{-1} - 1)\, U_{CO,LU}(yr) \qquad (6)$$

Sectors other than LULUCF were grouped into drinking water, livestock, energy, mining, and manufacturing industries, each
with various subsectors (Sect. 4 and Appendix C). Water use for these sectors was estimated using methodologies from
previous studies commissioned by DGA (e.g., DGA, 2017). These studies quantified water consumption for a specific year
based on sector-specific forcing data (e.g., mineral mass production for the mining sector) and consumption rates specific to
each class (e.g., water usage per ton of copper concentrates). Having established the consumption rate values specific to each
sector or process, the major challenge for historical water use estimation was reconstructing the evolution of sector-specific
drivers. To obtain continuous time series from the mid 20th century to the present, we followed a methodology similar to that
described for the CR2LUC dataset. Various documents and national inventories were homogenized to get continuous spatio-
temporal series of the forcing data for each sector ($F_S$). This information was then combined with water consumption rates
($K_S$) to estimate the volumetric water use as follows:

$$U_S(x, yr) = K_S \cdot F_S(x, yr) \qquad (7)$$



Appendix C provides a full list of sectors, their consumption factors, and the forcing variables and sources. The CR2WU dataset includes both consumptive and non-consumptive uses from LULUCF and non-LULUCF sectors. While estimates for
LULUCF are computed on a regular grid, the data is aggregated and reported at the communal scale, with annual estimates from 1960 to 2020. CR2WU data are available at https://doi.org/10.5281/zenodo.13324235 (Boisier et al., 2024b).

**2.6 Computation of water stress index and its drivers of change**

Similar to previous studies (e.g., Falkenmark and Lundqvist, 1998; Vörösmarty et al., 2000; Oki and Kanae, 2006), the WSI is defined here as the ratio $U_{co}/A$, where $U_{co}$ and $A$ represent total consumptive water use and near-natural water availability,
respectively, as described in previous sections. The WSI was computed for hydrological units, specifically over the major watersheds defined by the DGA in the National Inventory of Watersheds (BNA).

Given the simplicity of the WSI, it allows for a straightforward estimation of the relative impact on water stress levels resulting from variations in both water availability (climate-driven supply) and water usage (demand), as presented in Sect. 5. The evolution over time driven by changes in water availability ($\Delta WSI|_A$) and use ($\Delta WSI|_U$) are estimated by decomposing the WSI
ratio into its partial derivatives, leading to:

$$\Delta WSI|_A = -C\, \overline{WSI}\, \frac{\Delta A}{\overline{A}} \tag{8}$$

$$\Delta WSI|_U = C\, \overline{WSI}\, \frac{\Delta U}{\overline{U}} \tag{9}$$

Terms with $\Delta$ denote the difference of the corresponding variable between two periods of interest, and the overbar indicates the average over the two periods. Since the calculation is not performed on an infinitesimal scale, the sum of the partial
derivative terms does not precisely match the total change in WSI. To address this, both terms are adjusted proportionally (constant C) so that their sum equals the absolute change in WSI between the evaluated periods.

**3 Historical and future changes in water availability (1960-2100)**

Due to the latitudinal distribution and complex topography of Chile's continental territory, there is a large contrast in precipitation regimes from the hyperarid Atacama Desert (with virtually no precipitation) to extremely rainy areas in the
southern Andes and Patagonia (with annual accumulations exceeding three meters, Fig. 1).

As in other regions in the globe, potential evapotranspiration ($ET_0$) in Chile follows an inverse pattern to precipitation, with maximum values (above 2000 mm yr$^{-1}$) in the north of the country due to the high insolation and atmospheric water demand in the desert (Fig. 1b). The surface water limitation north of ~30°S and the $ET_0$ threshold south of ~40°S (below 1000 mm yr$^{-1}$) define the distribution of actual ET. The region in between, central Chile, features a mixed hydroclimate resulting from a
sharp transition between water-limited regimes in the north and energy-limited areas in the south. This region, home to most



of the country's population and economic activities, is also characterized by Mediterranean-like precipitation seasonality with dry summers and high inter-annual variability (Boisier et al., 2018; Aceituno et al., 2021).



**Figure 1: Long-term (1990-2020) mean annual precipitation (a), potential evapotranspiration (b), near-natural evapotranspiration (c), and water availability (d) in continental Chile. Panel e shows the zonal average of each water flux and the balance between P, $ET_N$ and A. Thin polygons in panel d indicate the country's major watersheds (BNA).**

On average across continental Chile, the mean annual rates of P and $ET_N$ are estimated at 1200 mm and 430 mm, respectively, which leads to a surface water availability of approximately 770 mm yr$^{-1}$ (corresponding to a volumetric flux of 680 km³ yr$^{-1}$). This average surface availability exceeds the global mean in continental areas, close to 300 mm yr$^{-1}$ (Oki and Kanae, 2006). However, the reality varies greatly between different regions of the country, as the gap between P and $ET_N$ defines a pronounced north-to-south gradient of water availability (see panels d and e in Fig. 1). Specifically, the administrative regions



of Los Lagos, Aysén, and Magallanes (south of 40°S) together account for more than 75% of the total available water volume in the country, while the regions from Valparaíso (~32°S) northward add up to less than 1% of the total, highlighting very different challenges in terms of water security in Chile.


**Figure 2: Changes in mean annual precipitation in Chile between the periods 1960-1990 and 1990-2020 (a-c), and projected towards the end of the 21st century (2070-2100) under global scenarios SSP1-RCP2.6 (d) and SSP3-RCP7.0 (e). Historical changes are based on local observations (a), the CR2MET dataset (b), and CMIP6 model simulations (historical scenario and SSP3 for 2015-2020). The three modeled changes (c-e) are based on downscaled simulations from 17 CMIP6 ESMs (showing the multi-model mean).**

Besides geographical differences, Chile's hydroclimate exhibits significant temporal variability. In the long term, local precipitation records show a clear downward trend between 1960 and 2020 across much of the country (Fig. 2). The spatially



distributed CR2MET dataset shows a precipitation decline consistent with observations, a match that matters given the subsequent use of this dataset for basin-scale assessments.

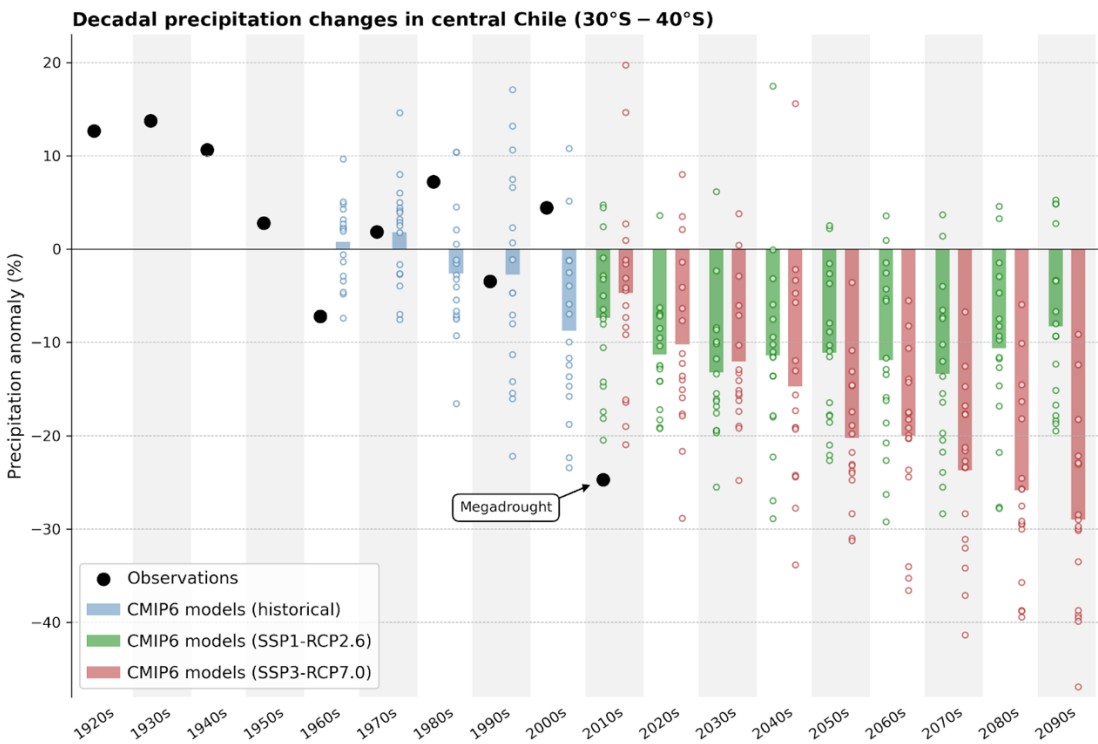

**Figure 3: Precipitation changes in central Chile (30°-40°S). Decadal mean precipitation anomalies relative to the 1960-1990 period, based on observations (black circles) and downscaled simulations from 17 ESMs (circles and bars show single and multi-model mean values). Model data based on CMIP6 simulations under the historical (blue), SSP1-RCP2.6 (green), and SSP3-RCP7.0 (red) scenarios.**

Studies addressing the driving mechanisms of the recent megadrought and long-term climate trends in Chile indicate that changes in precipitation result from natural climate variability and anthropogenic climate change, with the latter being more prominent in the long-term (Boisier et al., 2016, 2018; Garreaud et al., 2017, 2019; Damiani et al., 2020; Villamayor et al., 2021). These conclusions are based on the contrast between observations and simulations with global climate models, which systematically simulate a decrease in precipitation over the South Pacific in response to changes in greenhouse gas and stratospheric ozone concentrations (IPCC, 2022). This decrease precisely affects the country's regions where trends towards a drier climate are observed. The downscaled simulations from the CMIP6 ensemble assessed here are broadly consistent with observation-based changes, although slightly lower in magnitude (Fig. 2). This difference could be related to an additional drying driven by natural multidecadal changes and the recent megadrought in central Chile (Fig. 3), and/or to a higher actual regional sensitivity to anthropogenic forcing compared to the model average (Boisier et al., 2018).



The drying tendency overlaps with more substantial short-term variability, as seen in decadal precipitation anomalies in central
Chile (Fig. 3). The magnitude of the recent megadrought stands out on this time scale and highlights the importance of
considering low-frequency natural climate variability in water governance and planning, as well as its role in climate projection
uncertainty. Following Hawkins and Sutton (2009), the projected precipitation changes depicted in Fig. 3 are highly variable
and depend on three main sources of uncertainty: (1) the overlap of the climate change signal with phases of natural variability
that can last for decades, (2) the intensity of the global and regional climate change signal, with differences among climate
models, and (3) the global socioeconomic scenario considered. Considering these factors, in an optimistic case, a decrease in
precipitation of less than 10% can be expected in central Chile by the end of the 21st century. This projection is based on a
global scenario with high mitigation of greenhouse gas emissions (SSP1-RCP2.6, O'Neill et al., 2016) and models with low
regional climate sensitivity to anthropogenic forcing. In a pessimistic case, the deficit may exceed 30%, resulting in conditions
similar to the megadrought but permanently. This regime represents an average condition, and even drier decades (above 40%)
can be anticipated in the region due to the superposition of natural droughts with climate change. This scenario is projected
with medium to high global greenhouse gas emissions (SSP3-RCP7.0) and models with high climate sensitivity.

## 4 Current conditions and historical changes in water uses (1960-2020)

Water withdrawals for both human consumption and productive purposes have diverse impacts on water balances (Wada et
al., 2011). These impacts largely depend on whether the withdrawn water is returned to the basin, termed non-consumptive
use (e.g., in hydroelectric generation), or not returned, known as consumptive use (e.g., water evaporated in industrial and land
use activities).

In Chile, most activities with significant water use concentrate in the country's central and northern regions (Fig. 4).
Considering both consumptive and non-consumptive uses, the total water use in Chile is estimated to be around 100 km³ per
year at present. This value is similar to, albeit slightly higher than, other independent estimates (DGA, 2017; Fundación Chile,
2018; FAO and UN Water, 2021). Compared to other countries, total water use in Chile is high, mainly due to the significant
role of hydroelectric power generation in mountainous basins in the central-southern zone. This process involves using large
volumes of water, but the water is almost entirely returned to the system except for evaporation from the plant's reservoirs.
Thus, hydroelectric water use primarily alters the flow dynamics of the intervened river but does not significantly affect the
long-term water balance of the basin at its discharge into the sea.

Consumptive water uses are primarily attributed to the LULUCF sector, notably in central Chile (Fig. 4). With a flow close to
12 $km^3$ $yr^{-1}$ (400 $m^3$ $s^{-1}$), this sector represents three-quarters of the country´s consumptive water use. Specifically, irrigated
agriculture consumes a large volume of water (285 $m^3$ $s^{-1}$) due to high rates of crop evapotranspiration, often located in water-
limited areas with elevated $ET_0$. Non-irrigated LULUCF activities also account for a significant portion (nearly 15%) of
consumptive water use –a green water footprint–, and is the principal sector of use in several communes of central Chile (Fig.



4e). This water use is mainly associated with forestry plantations and, to a lesser extent, with rainfed agriculture and evaporation from artificial water bodies. Agriculture also has a non-consumptive water use component, as some irrigation water returns to the system through infiltration and percolation. This partition depends on irrigation technology, efficiency, and management practices. According to our methodology, land use activities may result in negative water use, that is, a landscape transformation that reduces ET compared to $ET_N$. This response is obtained in urban areas and some pasture lands

in southern Chile, though its magnitude is very low nationally.

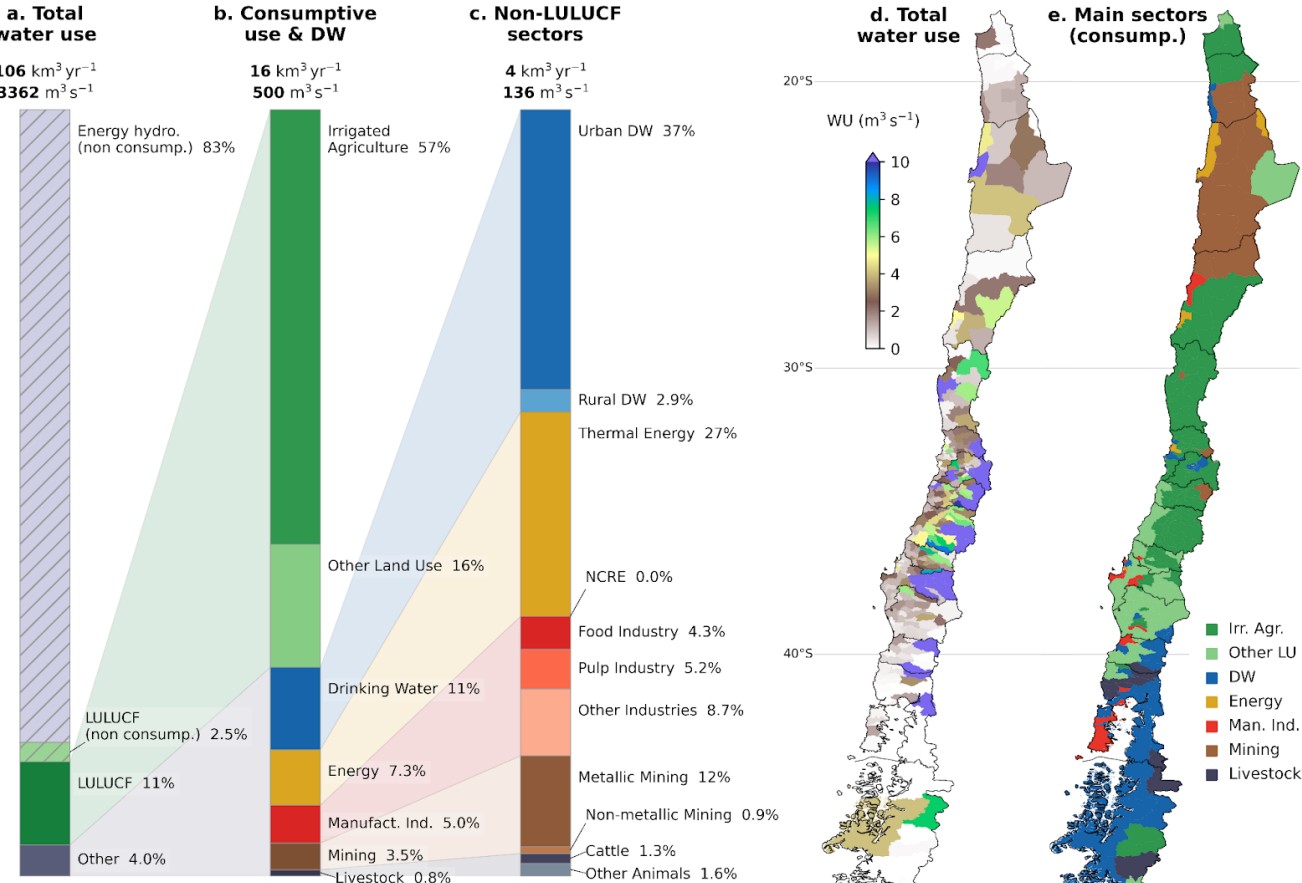

**Figure 4: Current water use in Chile (2010-2020 average). Bars (a-c) indicate the total national water use and sectoral contributions, including the details of consumptive water sectors (b) and the non-LULUCF sectors (c). Maps show the distribution of water use by commune across the central and northern regions of the country (d), and the sectors leading the consumptive use (e).**

Water supply systems, from extraction to treatment and wastewater return, constitute a partially consumptive water use sector totalling around 55 m³ s⁻¹ in Chile. This use is mainly associated with supply in urban areas, of which 30 m³ s⁻¹ (equivalent to 145 litters per person per day) corresponds to domestic consumption. Hence, on average, the provision of drinking water for human consumption in Chile meets the minimum standard of 100 litters per person per day, although there are significant

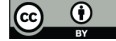



differences within the country, including areas with serious supply problems (Muñoz et al., 2020; Meza et al., 2014; Duran-
Llacer et al., 2020; Alvarez-Garreton et al., 2023b). Consumptive water uses of the energy sector (mostly thermoelectric power plants), mining, livestock, and manufacturing industry play a secondary role in the national total but are significant—and often dominant—at the local or watershed scale, particularly in the arid northern areas (see Fig. 4e).

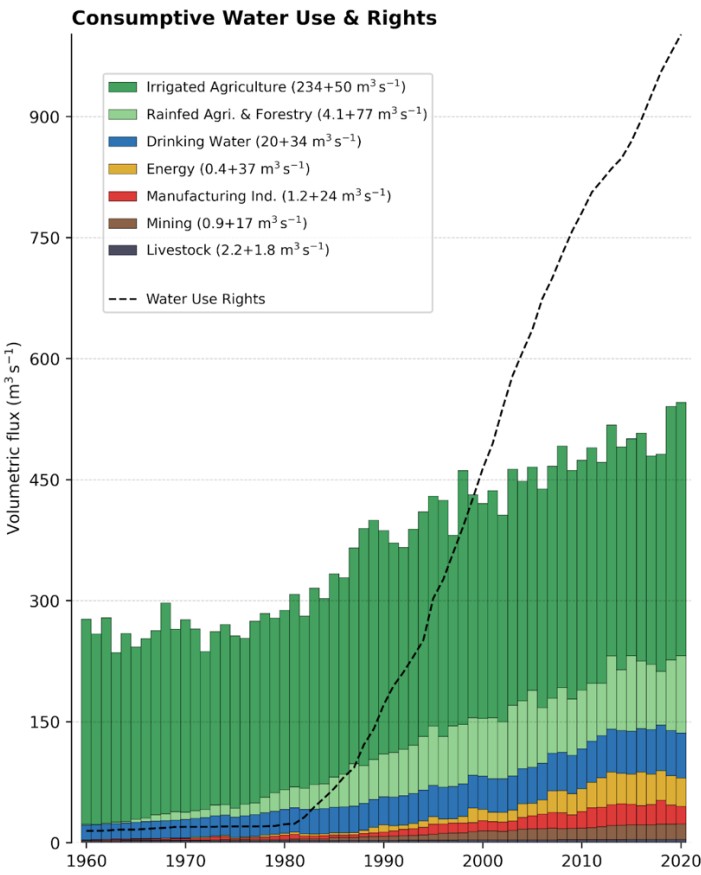

**Figure 5: Historical evolution (annual means) of consumptive water use and allocated rights (WURs) in Chile. The values in the**
**legend indicate the average water use during the 1960s and the increase towards the 2010s for each sector.**

Alongside population growth and socioeconomic development, water use in Chile increased substantially over the past six decades (Fig. 5). The LULUCF sector saw an increase from about 240 to 365 $m^3 \, s^{-1}$ (53%), explaining most of the rise in consumptive water use since the 1960s. Water use in irrigated agriculture has risen by approximately 20% due to increased production of annual crops and strong development in the fruit orchard industry, the latter with greater relevance within the 385 central-north valleys, featuring high insolation and drier conditions. This water use increase occurs despite a widespread transition from gravitational to sprinkler and drip irrigation systems, which has enhanced irrigation efficiency over the last four decades. Indeed, among the various sectors, irrigated agriculture's non-consumptive water usage component has been the



only one with a downward trend (not shown). Part of these changes can be explained by the irrigation efficiency paradox, suggesting that the reduced non-consumptive use associated with increased efficiency does not translate into effective water

savings at the basin scale; instead, it leads to greater consumptive use, as it allows for more crops to be irrigated when total water extractions are not limited (Grafton et al., 2018).

The forestry industry, which developed primarily between the 1970s and 2000s, has significantly contributed to the increase in water use (about 80 $m^3$ $s^{-1}$) and to greater pressure on water resources in watersheds with intensive Pinus radiata and Eucalyptus plantations, particularly along the coastal range of central Chile. This finding aligns with previous assessments of

water consumption by tree plantations in Chile (e.g., Alvarez-Garreton et al., 2019; Galleguillos et al., 2021; Balocchi et al., 2023) and in other regions worldwide under similar conditions (e.g., Farley et al., 2005; Beets and Oliver, 2007).

Due to changes in the LULUCF sector, and the water use increases in other productive areas and drinking water, the total consumptive water demand has doubled over the study period (Fig. 5). Additionally, during the second half of the 20th century, non-consumptive uses grew more than fourfold due to the major implementation of hydroelectric power plants (not shown).

In the 21st century, the installed energy capacity and generation continued to grow, primarily through thermoelectric plants, which have significant consumptive water use (Fig. 5), and more recently, through solar and wind power plants, which have very low water consumption.

The water use estimates presented in this section are based on actual socioeconomic activities in Chile (Sect. 2.5), regardless of whether these activities have a Water Use Right (WUR) granted by the State. The regulation regarding the access to

freshwater sources for consumptive or non-consumptive uses through WURs has been systematic since the introduction of the Water Code in 1981, which is still in force in the country (Congreso Nacional de Chile, 2022). This regulation has formalized customary rights and granted new ones, leading to the steeply increasing curve of the total allocated flow in Chile since the early 1980s (dashed curve in Fig. 5). As expected, the national water use estimated here for the present time is below the exploitable volume according to the allocated consumptive WURs, which totals about 1000 $m^3$ $s^{-1}$ across the country. However,

there are exceptions in some basins where uses are supplemented by desalinated sea water, which is not recorded under WURs, as they only consider terrestrial freshwater sources. An example is the coastal basin of Quebrada Caracoles in the Antofagasta region (~23°S), where water usage is primarily for the thermoelectric industry and is supplied by desalination plants. Other basins where estimated water use exceeds the WURs are related to forestry activities, particularly in coastal basins within the Maule and Biobío regions (~37°S). This discrepancy occurs because it is not required to have a WUR to make use of water

naturally contained in the soil from precipitation, and rekindles the discussion regarding the legal monitoring and regulation of consumptive uses that do not involve direct extraction from a river or a pumping well (e.g., Prosser and Walker, 2009; Rockström et al., 2010, Alvarez-Garreton et al., 2019), even though these uses can be very significant in some basins (Fig. 4e). Further details regarding the water allocation system in Chile and its compatibility with water security can be found in previous studies (e.g., Barria et al. 2021a, Alvarez-Garreton et al., 2023a).



## 5 Current conditions and drivers of change in water stress

In this section, we present the results addressing the primary question concerning the current diagnosis of water stress in Chile and the historical factors contributing to changes in stress levels. Considering the entire territory of continental Chile, current consumptive uses represent only 2 to 3% of total water availability. However, due to the large regional contrasts in water availability (Fig. 1) and the particular mismatch between regions with high availability and those with higher water demand (Fig. 4), water security levels are highly unequal across the country. Indeed, water demands approach or even exceed the available surface water in a number of basins today.

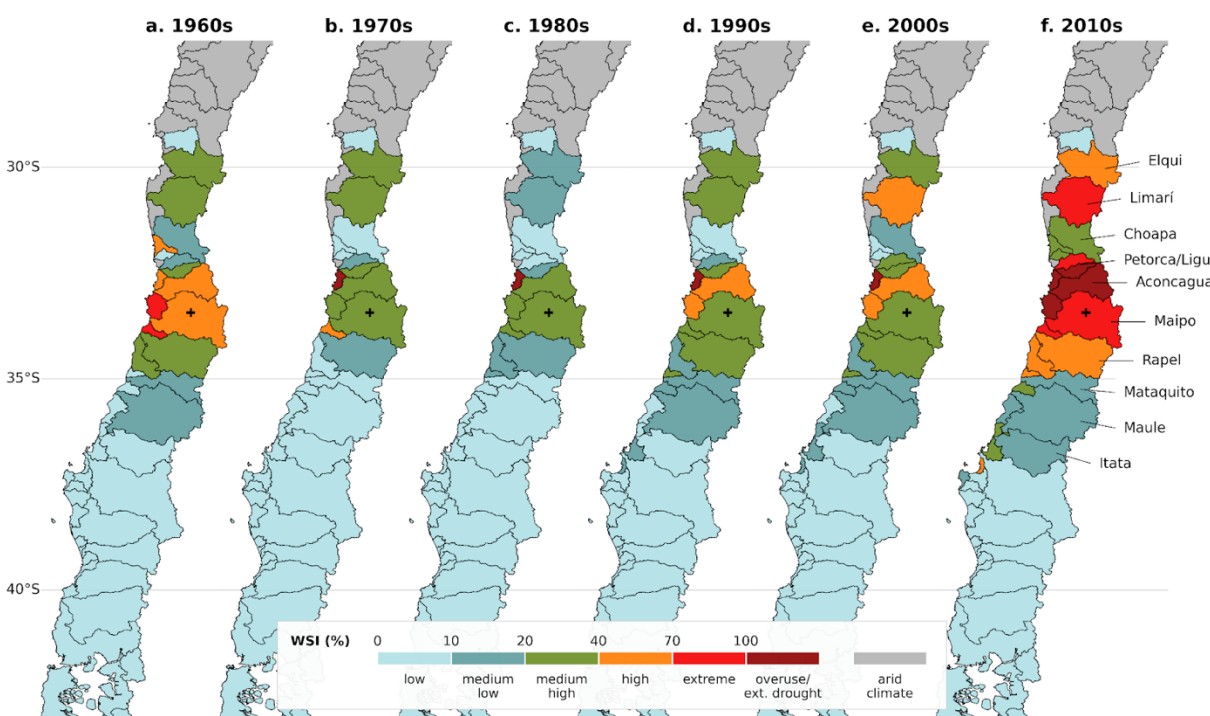

**Figure 6: Decadal mean Water Stress Index (WSI) for the major watersheds in central Chile between 1960 and 2020. The names of the basins analyzed further are indicated in the right-hand panel. The cross indicates the location of Santiago city.**

During the second half of the 20th century, most basins between the administrative regions of Coquimbo and Biobío (30 to 37°S) remained at low to medium levels of water stress (Fig. 6). In the 1960s, some basins, including those of Maipo and Aconcagua rivers (which supply the metropolitan area of Santiago and the region directly to the north, respectively), reached WSI levels above 40% due to an intense drought that hit the region at the end of that decade. In the following period until 2010, only Aconcagua, along with some coastal basins in the central zone, experienced elevated stress levels. By contrast, in the 2010-2020 decade, the combination of low water availability due to the megadrought and higher water use rates significantly increased water stress levels in several basins. During this period, the Maipo River basin reached an extreme water stress level, while other basins, such as La Ligua and Aconcagua exhibited critical levels, with WSI values exceeding



100%, indicating that water use surpasses available surface water. These elevated water stress levels have been associated with unsustainable use of groundwater reserves, as evidenced by sustained declines in the water table in this region (Alvarez-
Garreton et al., 2024; Jódar et al., 2023; Taucare et al., 2024).

The Aconcagua River basin represents a case of extreme water stress. In this watershed, covering about 7300 km$^2$, urban and rural areas coexist alongside multiple productive activities with high and increasing water consumption, sometimes surpassing surface water availability (Fig. 7). Another indication of water demand pressure in this basin is the present-day small gap between the estimated water use and the legal withdrawal limit according to the total WURs (red bars and black curve in Fig.
7a). As in other regions with natural water limitations, Aconcagua's supplies rely heavily on reservoirs and groundwater withdrawals (the groundwater to total WURs ratio currently reaches about 80%).

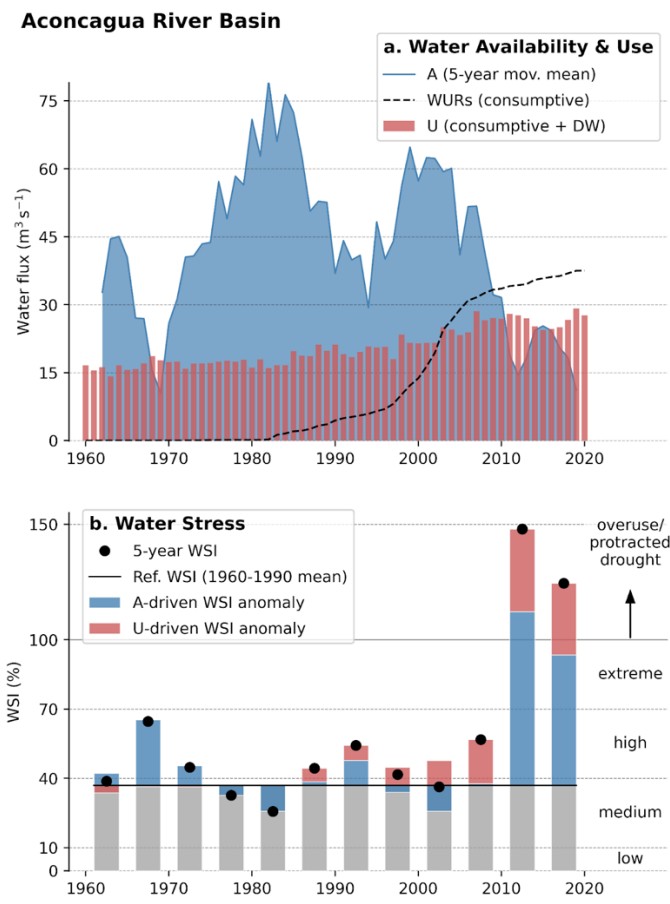

**Figure 7: (a) Water availability (A, blue), consumptive water uses (U, red) and rights (WURs, black) in the Aconcagua River basin from 1960 to 2020. (b) Five-year Water Stress Index (WSI) averages in the basin (black dots). Anomalies for each quinquennium**
**relative to the 1960-1990 mean are highlighted in color. The WSI anomalies are disaggregated into two components related changes in water availability (blue) and changes in water use (red).**



During the decades before the megadrought, consumptive water uses in Aconcagua represented about 40% of the water availability in the basin, indicating a medium to high level of water stress. The water use to availability ratio narrows during periods of drought, as observed during the second half of the 1960s and the recent megadrought (2010-2020), where the 5-
year mean WSI exceeded 65% (high to extreme water stress) and 100% (a critical condition), respectively.

The primary role that climate variability plays in WSI is evident in the case of the Aconcagua River basin, as shown by quinquennial WSI anomalies attributed to changes in water availability (blue bars in Fig. 7b). However, it is also clear that, in addition to the increased WSI during periods of precipitation deficits, the long-term growth in water use has progressively increased water stress between 1960 and 2020 (red bars in Fig. 7b). Particularly, the increased water use amplified stress levels
during 2010-2020, significantly intensifying the impact of the megadrought on water resources.

**Figure 8: Mean WSI for the periods 1960-1990, 1990-2020 and 2035-2065 in the major basins of central Chile. Changes in WSI compared to a reference period (black line), and the components related to water availability and water use are highlighted in blue and red, respectively. The WSI for the two historical periods are based on actual water use/availability estimates (O). Water**
**availability projections for the mid-21st century are based on 11 ESM simulations (showing the multi-model mean) and two socioeconomic scenarios: one with high (E1, SSP1-RCP2.6) and the other with low (E2, SSP3-RCP7.0) global greenhouse gas emission mitigation. Water use projections assume a linear extrapolation of the trend observed between 2000 and 2020.**



As in Aconcagua, the increase in water demand since the 1960s has narrowed the gap between availability and use, leading to a substantial rise in water stress levels in most basins in central Chile. The 30-year mean WSI for the period 1990-2020 reflects
this condition (Fig. 8). Compared with the previous 30-year period (1960-1990), the WSI increase is mainly associated with the growth in water demand and, to a lesser extent, with the long-term decrease in water availability between the two periods (see red and blue bars in Fig. 8).

Water stress levels in central Chile are projected to worsen in the future due to the continuous decline in water availability and the potential strengthening of this trend under unfavorable climate scenarios (Fig. 8). The water availability decline is primarily
attributed to lower precipitation rates and, less significantly, to increased evapotranspiration caused by higher temperatures (not shown). As mentioned in Sect. 3, climate projections involve various sources of uncertainty, and the actual future conditions will depend on the global climate change scenario and regional sensitivity to large-scale climatic disturbances. However, climate model projections consistently show a direction of change towards lower precipitation in central Chile (Fig. 3), leading to high and extreme WSI values in the mid-21st century in the basins of the Elqui, Limarí, Petorca/La Ligua,
Aconcagua, and Maipo rivers (Fig. 8). This result considers only changes in water availability under a future climate scenario with a pathway (greenhouse gas increase) similar to one of the recent decades (SSP3-7.0). A further discussion regarding future trends and mitigation options to reduce water stress in Chile is presented next.

**6 How to advance toward water security goals?**

Given the limited control that local governance and actions have over climate evolution, and considering the precautionary
principle regarding climate and water availability projections (e.g., Martin, 1997; Costa, 2014), the effects of water demand on the evolution of water security are particularly relevant in Chile.

As seen during the megadrought, increased water stress under unfavorable future climate conditions could be significantly exacerbated if water demand continues to rise in basins at high risk of scarcity. To evaluate this threat, along with climate projections, a simple future scenario of water use to the mid-21st century was considered based on recent trends (2000-2020)
extrapolation. Under these conditions, several basins are projected to reach extreme levels of water stress, in some cases exceeding the threshold of physical sustainability (WSI > 100%, Fig. 8). WSI values near or above 100% indicate a structural condition of water overuse with major social and environmental impacts, including even greater pressure on groundwater reserves, as Alvarez-Garreton et al. (2024) reported.

Public policy in Chile is aware of the impacts of droughts and the challenges that climate change poses to water security. In
particular, the Framework Law on Climate Change (MMA, 2022) establishes a series of legal instruments and adaptation plans, many of which are under development at the time of writing this paper. However, some of these instruments, explicitly oriented towards water resources (MOP and DGA, 2024), have not yet defined quantitative metrics of water security. In our opinion,





metrics such as the water stress index assessed here are necessary to establish goals and monitor the efficacy of potential actions to achieve those goals. Regarding WSI, stress levels are well-defined worldwide in relation to their impacts on
watersheds (e.g., Falkenmark, 2013a; 2013b; Rockström et al., 2014; Oki and Kanae, 2006; Grafton et al., 2012).

Given global climate scenarios, there are only two ways to alleviate water stress in a basin: reducing consumptive water use or increasing water availability through alternative sources. If public policy sets a target of, e.g., reaching WSI values of 40% or lower by 2050, different actions to adjust water availability and demand could help achieve this goal. As an illustrative example, we look at different cases leading to that target in the Aconcagua River basin, considering the basin history and the
climate projections assessed in this study. The consumptive water use in the basin nearly doubled over the past six decades, primarily due to the growth of irrigated agriculture (Fig. 9). A period of large water availability in the 1980s, along with international trading opportunities, encouraged policy decisions aimed at boosting agricultural development and positioning Chile as a global food-exporting leader (Villalobos et al., 2006). However, the onset of the megadrought and the climatic projections for the region question the feasibility of sustaining these consumption levels if water security is to be
achieved.

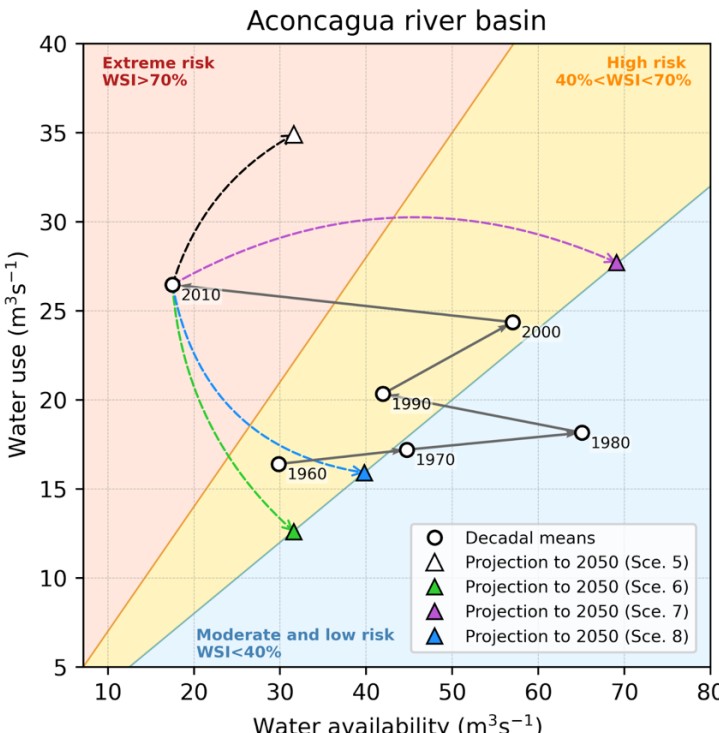

**Figure 9: The decadal evolution of consumptive water use, water availability, and the corresponding WSI category (background) in the Aconcagua River basin for the period 1960-2020 (circles). Triangle markers indicate different scenarios for the mid-21st century under a global climate scenario with low greenhouse gas emission mitigation (SSP3-7.0, scenarios 5 to 8 in Table 2).**



Different pathways to meet the water security target (WSI < 40%) by 2050 in the basin are presented in Table 2, including extreme cases where the target is met either by exclusively reducing water use (scenarios 2 and 6 in Table 2) or by solely increasing water availability (scenarios 3 and 7) through engineering solutions, such as inter-basin water transfers or seawater desalination. In the first case, adjustment measures would need to significantly reduce consumptive uses (by nearly 50% under unfavourable climate conditions). In the second case, alternative sources would need to increase water availability by 35 m³ s⁻

¹, that is, doubling the natural availability in Aconcagua.

Both solutions involve socio-economic and environmental costs that should be carefully assessed. The best approach will likely involve combined efforts. Desalination has rapidly developed in Chile and is projected as a key solution for mitigating risks to basic water provision and industry in northern Chile (Vicuña et al., 2022). However, given the current energy, economic, and environmental costs, the expansion of this technology to supply high water-demand sectors such as agriculture

still seems distant. As a rough estimate, considering only the economic cost and assuming a very low production rate of desalinated seawater (0.25 USD m⁻³; about half of the lowest published costs; Vicuña et al., 2022), an annual investment of USD 65 million, equivalent to 10% of the GDP of the agricultural sector in Valparaiso (the administrative region that houses the Aconcagua basin), would lead to a water flux of about 8 m³ s⁻¹. Even with this alternative water source, consumptive water use would still need to be reduced by 20 to 35% to achieve the target of a 40% WSI by mid-21st century, depending on the

climate trends (Table 2). These values can serve as benchmarks for further in-depth assessments of actions focused on climate change adaptation and on mitigating the water use impacts of current land use activities.

It is important to distinguish between total and consumptive water uses when considering alternatives to reduce water stress, particularly in the design of water use efficiency plans. Irrigation strategies can alter how water is used in a watershed by either redistributing it more effectively or by reducing plants' consumptive water use. In the first case, studies have shown that it is

possible to reduce irrigation by 20% in table grape crops while maintaining transpiration and reducing percolation, thereby avoiding negative impacts on water quality, such as the transport of agrochemicals to groundwater (Pizarro et al., 2021). This approach reduces total water use, but not consumptive use. In the second case, a 25 to 30% reduction in transpiration can be achieved through controlled plant stress management, as demonstrated locally in avocado cultivation (Beya et al., 2023). However, these management strategies can only reduce basin-scale water stress if they result in a reduction in total consumptive

water use (Grafton et al., 2018).

It should be noted that non-renewable water reserves, such as aquifers that are not in equilibrium or melting glaciers, are not considered as alternative sources to reduce the WSI. These water reserves, along with those from artificial reservoirs, play a key role in water security primarily through temporal regulation and water accessibility. However, they do not represent a long-term additional source of water since their storage is limited by surface recharge rates. Given this limitation, water

consumption at rates close to or exceeding surface availability will not be sustainable over time, regardless of whether the access is from underground, surface sources, or reservoirs (Alvarez-Garreton et al., 2024).



**Table 2: Projected WSI for the mid-21st century (2035-2065) in the Aconcagua River basin under different scenarios of climate and related water availability ($A_C$), alternative water availability sources ($A_A$), such as seawater desalination or inter-basin transfer, and consumptive water use ($U_{CO}$). Relative changes compared to the reference period 1990-2020 are provided in parentheses. Projections of $A_C$ based on eight ESM simulations and two socio-economic scenarios: one with high global mitigation of greenhouse gas emissions (SSP1-RCP2.6) and another with low mitigation (SSP3-RCP7.0).**

| Reference 1990-2020 | $A_C$ 2035-2065 | $A_A$ 2050 | $U_{CO}$ 2050 | WSI 2036-2065 | Scenario description |
|---|---|---|---|---|---|
| $A_C$: 38.4 m$^3$ s$^{-1}$ <br><br> $A_A$: 0 m$^3$ s$^{-1}$ <br><br> $U_{CO}$: 23.8 m$^3$ s$^{-1}$ <br><br> WSI: 62% | 39 m$^3$ s$^{-1}$ (+2%) | 0 | 34.9 m$^3$ s$^{-1}$ (+46%) | 90% | (1) U-trend/CC-optimistic: Climate under SSP1-RCP2.6, water use after following recent trend (2020-2020). |
| | | 0 | 15.5 m$^3$ s$^{-1}$ (−35%) | 40% | (2) U-driven target/CC-optimistic: Climate under SSP1-RCP2.6, water use change required to meet 40% WSI target. |
| | | 30.4 m$^3$ s$^{-1}$ | 27.7 m$^3$ s$^{-1}$ (+16%) | 40% | (3) A-driven target/CC-optimistic: Climate under SSP1-RCP2.6, water use fixed to 2020 value, alternative water sources required to meet 40% WSI target. |
| | | 8.2 m$^3$ s$^{-1}$ | 18.8 m$^3$ s$^{-1}$ (−21%) | 40% | (4) Mixed-solutions target/CC-optimistic: Climate under SSP1-RCP2.6, combined actions to meet 40% WSI target, $A_A$ limited by desalination investment equivalent to 10% of current regional/sectoral GDP (Valparaiso/Agriculture, refs). |
| | 31.6 m$^3$ s$^{-1}$ (−18%) | 0 | 34.9 m$^3$ s$^{-1}$ (+46%) | 106% | (5) As scenario (1) but climate under SSP3-RCP7.0. |
| | | 0 | 12.6 m$^3$ s$^{-1}$ (−47%) | 40% | (6) As scenario (2) but climate under SSP3-RCP7.0. |
| | | 37.5 m$^3$ s$^{-1}$ | 27.7 m$^3$ s$^{-1}$ (−16%) | 40% | (7) As scenario (3) but climate under SSP3-RCP7.0. |
| | | 8.2 m$^3$ s$^{-1}$ | 15.9 m$^3$ s$^{-1}$ (−33%) | 40% | (8) As scenario (4) but climate under SSP3-RCP7.0. |

## 8 Conclusions

This paper offers a comprehensive evaluation of Chile's current, historical, and future water stress conditions, utilizing novel datasets on water availability and use. These datasets address critical information gaps in Chile and are relevant for various applications, including climate, land use, and water management.

We highlight three main conclusions of this study:

1. Most basins in central Chile experienced high to extreme water stress between 2010 and 2020. This situation was primarily driven by reduced water availability during the megadrought and was further exacerbated by high water demand in the region.



2. From a historical perspective, water stress has steadily increased over the last six decades in central Chile, leading to permanent (30-year mean) high levels of water stress (WSI > 40%) in several basins from Santiago to the north. This increase is mainly attributed to rising water consumption and, to a lesser extent, to a reduction in surface water availability. During this period, consumptive water use in the country has doubled, largely due to the expansion of the agricultural and forestry industries.

3. In a scenario of adverse climate change, conditions similar to those of the megadrought are projected to become permanent by the end of the 21st century, with near 30% reduction in precipitation. Under such circumstances, it is likely that most basins in the central and northern regions of the country will face permanently high and extreme levels of water stress by the mid-21st century.

Given these results and the contrast with historical Water Use Rights (WURs) allocations, it is important that decision-makers and water users acknowledge that Chilean regulations have permitted and continue to allow consumptive water uses that exceed recognized sustainability levels in the central-northern regions of the country. Then, measures can be evaluated and agreed upon to mitigate the impact of high water consumption activities. These actions should be implemented alongside effective adaptation strategies to address current and future climate trends, particularly concerning the impacts on the Andes Cordillera. In addition to the projected decline in precipitation and in fresh water availability, a warmer climate will likely reduce the snow accumulation capacity of headwater basins in central Chile, leading to lower meltwater flows during the summer, when agricultural water demand is highest (Vicuña et al., 2011; Stehr and Aguayo, 2017; Bozkurt et al., 2018; Alvarez-Garreton et al., 2023b). This scenario poses a significant risk to water security, food security, and socio-economic stability in the region. Similar risks are faced by mountainous regions worldwide that rely on snow-dominated headwater basins (Drenkhan et al., 2023; Adam et al., 2009).

The assessed pathways of water use and alternative sources of water availability to achieve a water security target under climate change scenarios provide valuable insights for adaptation plans currently being developed. While having a comprehensive set of water security indices is essential for evaluating adaptation strategies, a key challenge lies in making political decisions to establish goals for these indices and determining the necessary changes and associated costs to achieve them.

The methods used in this study can be applied to any region that meets the necessary data requirements. The WSI is a straightforward, catchment-scale index that complements metrics assessing other aspects of water security, such as water quality, sustainable groundwater use, ecosystem needs, and water accessibility. The estimates of water availability, water use and stress presented here carry uncertainties related to climate, hydrological, and land cover data that should be contrasted and complemented with independent approaches. Monitoring key variables related to water security and facilitating access to data, particularly from public agencies, is also crucial for a comprehensive assessment of water stress and effective planning.



## Appendix A: Earth System Models

**Table A1: List of Earth System Models used from CMIP6. Downscaled variables include precipitation (P) and daily**
**minimum/maximum temperatures ($T_N$, $T_X$) in a subset of models, depending on data availability at the time of the analysis.**

| Model | Institution | Reference | Dowscalled and derived variables |
|---|---|---|---|
| ACCESS-CM2 | Commonwealth Scientific and Industrial Research Organisation and Bureau of Meteorology, Australia | Bi et al. (2020) | P |
| ACCESS-ESM1-5 | | Ziehn et al. (2020) | P, $T_N$, $T_X$, $ET_0$, $ET_N$ |
| BCC-CSM2-MR | Beijing Climate Center, China | Wu et al. (2019) | P, $T_N$, $T_X$, $ET_0$, $ET_N$ |
| CanESM5 | Canadian Centre for Climate Modelling and Analysis, Canada | Swart et al. (2019) | P, $T_N$, $T_X$, $ET_0$, $ET_N$ |
| CMCC-ESM2 | Centro Euro-Mediterraneo sui Cambiamenti Climatici, Italy | Cherchi et al. (2019) | P |
| CNRM-CM6-1 | Météo-France/Centre National de Recherches Météorologiques, France | Voldoire et al. (2019) | P |
| CNRM-ESM2-1 | | Séférian et al. (2019) | P |
| EC-Earth3 | EC-Earth consortium, Europe | Döscher et al. (2021) | P, $T_N$, $T_X$, $ET_0$, $ET_N$ |
| FGOALS-g3 | Chinese Academy of Sciences, China | Li et al. (2020) | P |
| GFDL-ESM4 | Geophysical Fluid Dynamics Laboratory, USA | Dunne et al. (2020) | P, $T_N$, $T_X$, $ET_0$, $ET_N$ |
| IPSL-CM6A-LR | Institut Pierre-Simon Laplace, France | Boucher et al. (2020) | P, $T_N$, $T_X$, $ET_0$, $ET_N$ |
| MIROC6 | Japan Agency for Marine-Earth Science and Technology, Japan | Tatebe et al. (2019) | P, $T_N$, $T_X$, $ET_0$, $ET_N$ |
| MPI-ESM1-2-HR | Max Planck Institute for Meteorology, Germany | Mauritsen et al. (2019) | P, $T_N$, $T_X$, $ET_0$, $ET_N$ |
| MRI-ESM2-0 | Meteorological Research Institute, Japan | Yukimoto et al. (2019) | P, $T_N$, $T_X$, $ET_0$, $ET_N$ |
| NorESM2-MM | Norwegian Climate Centre, Norway | Seland et al. (2020) | P, $T_N$, $T_X$, $ET_0$, $ET_N$ |
| TaiESM1 | Research Center for Environmental Changes, Taiwan, China | Lee et al. (2020) | P, $T_N$, $T_X$, $ET_0$, $ET_N$ |
| UKESM1-0-LL | Met Office Hadley Centre, United Kingdom | Sellar et al. (2019) | P |

## Appendix B: Evapotranspiration model

An evapotranspiration (ET) model was developed to assess changes in water availability and estimate water use in the LULUCF sector across Chile. This model employs a bucket scheme, similar to those used in many surface or hydrological models (e.g., Sellers et al., 1997). The model includes a soil water reservoir, replenished by precipitation (P) up to its maximum
capacity and depleted through evapotranspiration (ET). The ET flux scales linearly with the soil water content and is limited by a maximum ET ratio relative to potential evapotranspiration ($ET_0$), defined by a parameter $\beta_X$. Similar to crop coefficients, $\beta_X$ represents plants' inherent transpiration (T) intensities in the absence of moisture stress, reflecting canopy conductance and other physiological/morphological properties that affect the water use efficiency of different functional types. Intercepted precipitation in the vegetation canopy is explicitly accounted for through a secondary small tank, leading to evaporation ($E_I$)



with no limitations other than $ET_0$ and the available canopy water (as a water body). The model runs on a daily time step following these main equations:

$$ET(x,t) = a\,[\,E_I(x,t) + f_{IWB}\,ET_{IWB}(x,t)\,+\,(1 - f_{IWB})\,ET_{ML}(x,t)\,] \tag{B1}$$

$$E_I(x,t) = min\,[\,1.2\,ET_0(x,t), W_I(x,t)\,] \tag{B2}$$

$$ET_{IWB}(x,t) = \beta_X\,ET_0(x,t) \tag{B3}$$

$$ET_{ML}(x,t) = min\,[\,\beta(x,t)\,ET_0(x,t), W_S(x,t)\,]. \tag{B4}$$

The main output in Eq. B1 represents an ET flux composed of $E_I$ and two fluxes estimated under unlimited ($ET_{IWB}$) and limited ($ET_{ML}$) soil moisture conditions. The relative importance of $ET_{IWB}$ and $ET_{ML}$ is controlled by fraction $f_{IWB}$, which serves as an activation flag for irrigated areas or water bodies. Factor a is applied to each component to limit ET to a maximum rate of 1.2 $ET_0$. The factor of 1.2 defines the maximum possible ET flux, such as in open water conditions ($ET_0$ estimates used here stand

for a reference ET over grass; see Finch and Hall, 2001). The interception component $E_I$ is a flux of maximum evaporation until the canopy reservoir $W_I$ is emptied. $ET_{IWB}$ is only limited by $\beta_X$, while $ET_{ML}$ is constrained both by $\beta_X$ and the soil water content ($W_S$) through coefficient $\beta$:

$$\beta(x,t) =\ \beta_X\,\frac{W_S(x,t)}{w_{SX}(x)} \tag{B5}$$

where $w_{SX}$ represents the water capacity of the soil tank, which depends on a rooting depth parameter ($d_R$) and a fraction of

available water capacity (AWC), as follows:

$$w_{SX}(x) = AWC(x)\,d_R. \tag{B6}$$

While $d_R$ is prescribed for a given land cover class, AWC varies spatially according to soil texture and bulk density. In this study, AWC was derived from the latter properties for six soil layers (0-5, 5-15, 15-30, 30-60, 60-100 and 100 to 200 cm) using the Rosetta V3 pedotransfer functions (Zhang and Shaap et al., 2017), and it is provided by CLSoilMaps, a database of

gridded physical and hydraulic soil properties for Chile (Dinamarca et al., 2023; Galleguillos et al., 2024). Consequently, the model does not use a fixed soil layer depth. Instead, it employs the effective soil bucket capacity (wSX), which depends on the hydraulic properties of a given location (AWC) and the rooting depth ($d_R$) of a specific vegetation type.

Based on previous conditions, at the beginning of each time step, water reservoirs are partially or totally filled by fractions of total precipitation, which are allocated to the canopy ($P_I$, interception) and soil ($P_S$) components as:

$$P_I(x,t) = min\,[\,w_{IX} - W_I(x,t-1), P(x,t)\,] \tag{B7}$$

$$P_S(x,t) = min\,[\,w_{SX}(x) - W_S(x,t-1), P(x,t) - P_I(x,t)\,] \tag{B8}$$



where $w_{IX}$ stands for water holding capacity of the canopy. The remaining precipitation $(P - P_I - P_S)$ is considered as the model runoff. At the end of each cycle, the soil and canopy reservoirs are partially emptied following the $ET_{ML}$ and $E_I$ losses, respectively.

Hence, in addition to the forcing and internally computed variables, four parameters control the model: the coefficient $\beta_X$ (dimensionless), the rooting depth $d_R$ (mm), the canopy interception capacity $w_{IX}$ (mm) and the saturation fraction $f_{IWB}$ (boolean). Water fluxes and storages defined by Eqs. B1-8 are calculated independently for different land cover conditions, each with its own set of parameters (Table B1). The final ET flux is determined as a weighted average across the various land cover types coexisting in a spatial unit (grid cell), using a mosaic approach similar to that employed in most land surface

models (e.g., Blyth et al., 2021). For the purposes of this study, the model runs in a fully distributed manner over the domain and with the spatial resolution defined by the atmospheric forcing CR2MET (continental Chile, 0.05° latitude-longitude grid), and using the 48 land cover classes included in CR2LUC.

Soil evaporation is not explicitly included in Eq. B1 but is accounted for through the flux $ET_{ML}$ as an independent tile. The fractional area of this tile results from the barren land fraction prescribed in CR2LUC (code 600) and the residual fraction from

the effective coverage $(F_K^{(eff)})$ of other classes. In a similar approach to other models (e.g., Krinner et al., 2005), the fraction $F_K^{(eff)}$ reduces the area prescribed for vegetated classes $(F_K)$ based on its canopy cover, thereby accounting for differences in foliar density and the phenological cycle of plants. This is modeled following the Beer-Lambert law:

$$F_K^{(eff)}(x,t) = F_K(x,t)\,(1 - e^{0.6\,LAI_K(t)}) \tag{B9}$$

where $LAI_K$ is the Leaf Area Index of a vegetation class K. The factor 0.6 has been chosen as a typical value for the light

extinction coefficient (e.g., van Dijk and Bruijnzeel, 2001). Consequently, the actual grid area fraction of a vegetated class seen by the model will range between 0 and a value that tends to $F_K$ for elevated LAI. In this study, we computed the monthly mean LAI (2014-2019) for each vegetated class based on the 333-m PROBA-V LAI satellite dataset from the Copernicus Global Land Service (CGLS; Smets et al., 2018). The data were processed for equivalent grid cells with a dominant fraction of the corresponding CR2LUC class, using data sources from this product (Sect. 2.4), particularly the fruit cadastres from

ODEPA, the vegetation cadastres from CONAF, and the 30-m Land Cover for Chile (Zhao et al., 2016).

The main objective of this scheme is to accurately represent the ET mean rate, its seasonal and long-term variability, rather than modeling runoff as in a full hydrological model. As such, snow accumulation and melt are not included, nor are any processes related to runoff propagation. Sublimation over permanent snow/glacier conditions is accounted for in the same way as any other land cover type with its set of parameters. With these limitations, the model does not fully capture runoff

seasonality and variability, especially in mountain watersheds influenced by snow. However, the long-term mean runoff within watersheds were contrasted with observed river streamflows to adjust parameters and validate water balance of the model.





Land cover classes included in CR2LUC, along with the set of parameters used in this study, are listed in Table B1. Most parameters were based on published data. Rooting depth values can vary significantly depending on the metric used, such as

maximum depths or percentiles in vertical root distribution (e.g., Canadell et al., 1996; Schenk and Jackson, 2002). For agricultural classes, we prescribed the maximum effective rooting depth as specified in the Food and Agriculture Organization's Irrigation and Drainage Paper No. 56 (FAO56; Allen et al., 1998), similar to the values of maximum rooting depths reported by Fan et al. (2016). For natural vegetation and tree plantations, prescribed rooting depth values range from 0.3 meters (Tundra) to 3.0 meters (sclerophyllous and eucalyptus trees), based on the data from Canadell et al. (1996), Schenk

and Jackson (2002), Fan et al. (2017) and Tumber-Davila et al. (2022).

Parameter $w_{IX}$ varies between 0.3 and 2 mm. For agricultural classes, the values were derived using the model of van Dijk and Bruijnzeel (2001) with the leaf and stem water capacity from Zhong et al. (2022), and the LAI data described earlier. For natural vegetation and tree plantations, we used the storage capacity per unit of canopy area, also based on Zhong et al. (2022).

For $\beta_X$, we adopted the mid-season crop coefficients from FAO56 for agricultural classes, ranging from 0.7 to 1.2. Setting $\beta_X$

for natural classes and tree plantations is not straightforward. In this case, we were guided by published observations-based data on $ET/ET_0$ ratios (Liu et al., 2017), ET/P ratios (White et al., 2022; Balocchi et al., 2023), and T/ET ratios (Schlesinger and Jasechko, 2014; Benyon and Doody, 2015; Nelson et al., 2020). However, this information serves only as a reference since these quantities are influenced by factors such as moisture stress, leaf area, and intercepted evaporation, which do not precisely correspond to the unstressed $T/ET_0$ rates accounted for by $\beta_X$. In general, we aimed to maintain differences between functional

types (e.g., shrubs vs. trees, broadleaf vs. needleleaf), but the final $\beta_X$ values were calibrated by assessing the water balance in basins with monitored streamflow and distinct land covers (Fig. B1), resulting in values ranging from 0.5 to 0.9.

Non-vegetated classes were also characterized through these parameters. For barren land and built-up areas, topsoil layers of 200 mm and 50 mm were assigned, respectively. Additionally, a $w_{IX}$ value of 1.0 mm was included for the latter to account for intercepted water on impervious surfaces. Water bodies evaporate at maximum rates ($\beta_X = 1.2$), except for wetlands and

salt pans, which are only partially covered by saturated areas. Sublimation from glaciers and permanently snow-covered areas was modeled as water bodies with limited evaporation rates ($\beta_X = 0.2$), mimicking the $ET/ET_0$ ratios shown by ERA5 land over glacier areas, such as in the Patagonian Ice Fields.

Figure B1 presents a validation summary of the ET simulations. We used the CAMELS-CL dataset (Alvarez-Garretón et al., 2018) to compare long-term (1990-2020) average river streamflows (Q) observed in Chile with watershed-averaged

precipitation (P) and evapotranspiration (ET). A total of 147 basins were selected based on data availability (at least 75% coverage during 1990-2020) and the absence of disturbances significantly affecting river discharges. The sum of the observed Q and simulated ET shows a strong inter-basin correlation with the CR2MET P ($R^2 = 0.95$) and exhibits low overall biases (<1%). More noticeable and systematic biases, where Q+ET values exceed P, are found in humid watersheds (P > 2000 mm yr[-1]). We attribute these biases primarily to an underestimation of CR2MET P rather than an overestimation of ET, since the



latter is mostly limited by ET0 and the runoff coefficients (Q/P) in these watersheds approach or even exceed 100%. These

basins are predominantly covered by native, temperate rainforest. Aside from this group, there is no systematic bias across

other watersheds with distinct land cover (Fig. B1c). For benchmarking, we conducted the same analysis using independent P

and ET data from ERA5-Land reanalysis (Muñoz-Sabater et al., 2021), which yielded poorer results (Fig. B1b), highlighting

the added value of these new datasets for Chile.

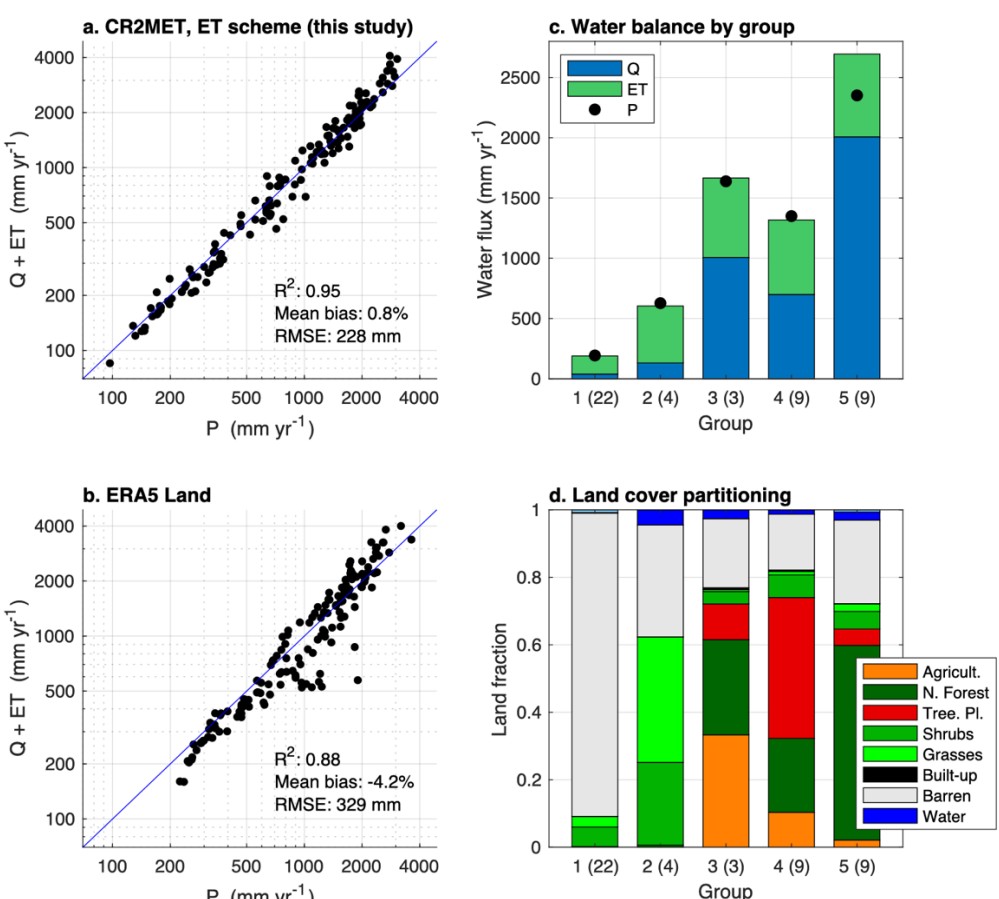


**Figure B1: Comparison of mean annual precipitation (P) in 147 watersheds in Chile with observed river streamflow (Q) and modeled evapotranspiration (ET). The estimates of P and ET in panels a and b correspond to those assessed in this study (CR2MET-driven) and from ERA5-Land, respectively. Panel c shows the CR2MET-based flux comparison for groups of watersheds with significant fractions of barren land (>85%, group 1), shrubland and grasses (>40%, group 2), agriculture (>30%, group 3), tree plantations**
**(>30%, group 4), and native forest (>50%, group 5). The number of basins and mean land cover partitioning of each group is shown in panel d.**



**Table B1: CR2LUC classes and parameters used for ET simulations**

| Level 1 | Level 2 | Level 3 | $\beta_X$ (1) | $d_R$ (mm) | $w_{IX}$ (mm) | $f_I$ (1) |
|---|---|---|---|---|---|---|
| (100) Agricultural Land | (110) Cereals | (111) Wheat | 1.15 | 1300 | 1.00 | 1 |
| | | (112) Maize | 1.20 | 1300 | 1.00 | 1 |
| | | (113) Oat | 1.05 | 1250 | 1.00 | 1 |
| | | (114) Rice | 1.20 | 750 | 0.50 | 1 |
| | | (115) Barley | 1.15 | 1250 | 0.80 | 1 |
| | | (116) Other | 1.15 | 1200 | 1.00 | 1 |
| | (120) Legumes and Tubers | (121) Potato | 1.15 | 500 | 0.80 | 1 |
| | | (122) Other | 1.15 | 500 | 0.80 | 1 |
| | (130) Industrial Crops | | 1.15 | 1250 | 0.80 | 1 |
| | (140) Vegetables and Flowers | (141) Vegetables | 1.05 | 500 | 0.50 | 1 |
| | | (142) Flowers | 1.00 | 500 | 0.50 | 1 |
| | (150) Forage Crops and Meadows | (151) Annual | 1.15 | 1200 | 1.00 | 1 |
| | | (152) Permanent | 1.15 | 1500 | 1.00 | 1 |
| | (160) Orchards | (161) Table Grapevine | 0.85 | 1500 | 0.75 | 1 |
| | | (162) Prunus | 0.95 | 1500 | 1.00 | 1 |
| | | (163) Pomes | 0.95 | 1500 | 0.90 | 1 |
| | | (164) Avocado | 0.85 | 1250 | 0.90 | 1 |
| | | (165) Citrus | 0.80 | 1200 | 0.70 | 1 |
| | | (166) Walnut Tree | 1.10 | 2000 | 1.00 | 1 |
| | | (167) Hazel Tree | 0.80 | 1800 | 1.00 | 1 |
| | | (168) Olive | 0.70 | 1500 | 0.60 | 1 |
| | | (169) Other | 0.85 | 1500 | 0.80 | 1 |
| | (170) Vineyard | (171) Wine | 0.85 | 1500 | 0.60 | 1 |
| | | (172) Pisco | 0.85 | 1500 | 0.60 | 1 |
| | (180) Pasture and Fallow | | 0.85 | 800 | 0.50 | 0 |
| (200) Forest | (210) Native | (211) Mediterranean Evergreen | 0.50 | 3000 | 1.50 | 0 |
| | | (212) Conifer | 0.65 | 2000 | 1.50 | 0 |
| | | (213) Temperate Broadleaf Deciduous | 0.65 | 2000 | 1.50 | 0 |
| | | (214) Temperate Broadleaf Evergreen | 0.70 | 2500 | 1.50 | 0 |
| | | (215) Chilean Wine Palm | 0.65 | 1500 | 1.00 | 0 |
| | (220) Tree Plantation | (221) Pine | 0.65 | 2500 | 2.00 | 0 |
| | | (222) Eucalyptus | 0.70 | 3000 | 1.50 | 0 |
| | | (223) Other | 0.65 | 2500 | 1.50 | 0 |
| (300) Shrubland | (310) Shrubland with Sparse Trees | | 0.50 | 1500 | 1.00 | 0 |
| | (320) Shrubland with Succulent Plants | | 0.60 | 800 | 0.50 | 0 |
| | (330) Other | | 0.50 | 1500 | 1.00 | 0 |
| | (410) Steppe | | 0.90 | 1000 | 0.50 | 0 |



| (400) Natural Grassland and Tundra | (420) Other Grassland | | 0.90 | 800 | 0.50 | 0 |
|---|---|---|---|---|---|---|
| | (430) Tundra | | 0.90 | 500 | 0.30 | 0 |
| (500) Built-up land | | | 1.00 | 50 | 1.00 | 0 |
| (600) Barren land | | | 1.00 | 200 | 0.00 | 0 |
| (700) Water Bodies and Salt Pans | (710) Lakes | | 1.20 | 0 | 0.00 | 1 |
| | (720) Rivers | | 1.20 | 0 | 0.00 | 1 |
| | (730) Reservoirs | | 1.20 | 0 | 0.00 | 1 |
| | (740) Other Water Bodies | | 1.20 | 0 | 0.00 | 1 |
| | (750) Wetland | | 0.50 | 0 | 0.00 | 1 |
| | (760) Salt Pans | | 0.10 | 0 | 0.00 | 1 |
| (800) Glacier and Permanent Snow | | | 0.20 | 0 | 0.00 | 1 |

# Appendix C

**Table C1: CR2WU classes, homogenized forcing data and water consumption rates for non-land cover sectors. All water consumption rates are expressed in liters per forcing unit per day.**

| Level 1 | Level 2 | Level 3 | Forcing data at communal and yearly resolution (the national total in 2020 is provided). | Water consumption rate |
|---|---|---|---|---|
| Drinking water (100) | Urban (110) | Domestic (111) | Population in urban areas, based on data from the National Institute of Statistics (INE, 1964; 1988; 1993; 2018a; 2018b; 2018c) and INE-CELADE (1972). Urban population in 2020: 15,880,000 | 173 L/person/day, based on (SISS (2020a) and SISS (2020b). |
| | | Commercial (112) | GDP in MUSD adjusted to 2020 USD value (millions), based on Banco Central (2020a, 2020b, 2020c, 2020d) and World Bank (2021). Commercial GDP in 2020: 25,670 MUSD | 16,490 L/MUSD/day, based on SISS (2020a) and SISS (2020b). |
| | | Industrial (113) | GDP in MUSD adjusted to 2020 USD value, based on Banco Central (2020a, 2020b, 2020c, 2020d) and World Bank (2021). Industrial GDP in 2020: 27,360 MUSD | 2,412 L/MUSD/day, based on SISS (2020a) and SISS (2020b). |
| | | Green areas (114) | Population in urban areas, based on INE (1964, 1988, 1993, 2018a, 2018b, 2018c) and INE-CELADE (1972). Urban population in 2020: 12,438,000 (*) | 6 L/person/day, based on (SISS (2020a) and SISS (2020b). |
| | Rural (120) | Domestic (121) | Population in rural areas, based on INE (1964, 1988, 1993, 2018a, 2018b, 2018c) and INE-CELADE (1972). Rural population in 2020: 2,174,462 | 135 L/person/day based on DGA (2017). |
| Unbilled drinking water (200) | Urban (210) | Domestic (211) | Population in urban areas, based on data from the National Institute of Statistics (INE, 1964; 1988; 1993; 2018a; 2018b; 2018c) and INE-CELADE (1972). Urban population in 2020: 15,880,000 | 60 L/person/day, based on SISS (2020a) and SISS (2020b) (**). |
| | | Commercial (212) | GDP in MUSD adjusted to 2020 USD value, based on Banco Central (2020a, 2020b, 2020c, 2020d) and World Bank (2021). Commercial GDP in 2020: 25,670 MUSD | 5,667 L/MUSD/day, based on SISS (2020a) and SISS (2020b) (**). |



| | | Industrial (213) | GDP in MUSD adjusted to 2020 USD value, based on Banco Central (2020a, 2020b, 2020c, 2020d) and World Bank (2021). Industrial GDP in 2020: 27,360 MUSD | 877 L/MUSD/day, based on SISS (2020a) and SISS (2020b) (**). |
|---|---|---|---|---|
| | | Green areas (214) | Population in urban areas, based on INE (1964, 1988, 1993, 2018a, 2018b, 2018c) and INE-CELADE (1972). Urban population (applicable) in 2020: 12,438,000 (*) | 2 L/person/day, based on (SISS (2020a) and SISS (2020b) (**). |
| | Rural (220) | Domestic (221) | Population in rural areas, based on INE (1964, 1988, 1993, 2018a, 2018b, 2018c) and INE-CELADE (1972). Rural population in 2020: 2,174,462 | 27 L/person/day, based on DGA (2017) (**). |
| Mining (300) | Metallic (310) | Copper concentrates (311) | Fine copper production in tons, based on COCHILCO (2006, 2020a, 2021a, 2021b, 2021c). Production in 2020: 4,170,000 t | 234 L/Ton/day, based on COCHILCO (2020b, 2021d) and DGA (2017) (***). |
| | | Copper cathodes (312) | Fine copper production in tons, based on COCHILCO (2006, 2020a, 2021a, 2021b, 2021c). Production in 2020: 1,560,000 t | 128 L/Ton/day, based on COCHILCO (2020b, 2021d) and DGA (2017) (***). |
| | | Molybdenum (313) | Fine metal production in tons, based on COCHILCO (2006, 2020a, 2021a). Production in 2020: 55,142 t | 2,045 L/Ton/day, based on COCHILCO (2020b, 2021d) and DGA (2017) (***). |
| | | Silver (314) | Fine metal production, based on COCHILCO (2006, 2020a, 2021a). Production in 2020: 81 Ton (Valparaíso Region) (****) | 558,000 L/Ton/day, based on COCHILCO (2020b, 2021d) and DGA (2017) (***). |
| | | Gold (315) | Fine metal production in tons, based on COCHILCO (2006, 2020a, 2021a). Production in 2020: 34 t | 5,000,000 L/Ton/day, based on COCHILCO (2020b, 2021d) and DGA (2017) (***). |
| | | Iron (316) | Fine metal production in tons, based on COCHILCO (2006, 2020a, 2021a). Production in 2020: 9,891,000 Ton | 1.14 L/Ton/day, based on COCHILCO (2020b, 2021d) and DGA (2017) (***). |
| | | Zinc (317) | Fine metal production in tons, based on COCHILCO (2006, 2020a, 2021a). Production in 2020: 22,600 t (Aysén Region) (****) | 39 L/Ton/day, based on COCHILCO (2020b, 2021d) and DGA (2017) (***). |
| | Non Metallic (320) | Lithium Carbonate (321) | Ore production in tons, based on COCHILCO (2006, 2020a, 2021a). Production in 2020: 114,260 t | 55 L/Ton/day, based on MOP (2012) and DGA(2017). |
| | | Nitrates (322) | Ore production in tons, based on COCHILCO (2006, 2020a, 2021a). Production in 2020: 996,515 t | 27 L/Ton/day, based on MOP (2012) and DGA(2017). |
| | | Iodine (323) | Ore production in tons, based on COCHILCO (2006, 2020a, 2021a). Production in 2020: 21,941 t | 3,836 L/Ton/day, based on MOP (2012) and DGA(2017). |
| | | Coal (324) | Ore production in tons, based on COCHILCO (2006, 2020a, 2021a). Production in 2020: 191,562 t | 0.8 L/Ton/day, based on MOP (2012) and DGA(2017). |
| | | Ulexite (325) | Ore production in tons, based on COCHILCO (2006, 2020a, 2021a). Production in 2020: 28,8103 t | 7.7 L/Ton/day, based on MOP (2012) and DGA(2017). |
| Energy (400) | Thermal (410) | Gas (411) | Gross energy produced in power plants, based on CNE (2019a, 2019b, 2019c, 2021a, 2021b, 2022). Production in 2020: 13,705,802 MWh | 37 L/MWh/day, based on DGA (2017) and Hardy & Garrido (2010). |
| | | Oil (412) | Gross energy produced in power plants, based on CNE (2019a, 2019b, 2019c, 2021a, 2021b, 2022). Production in 2020: 851,902 MWh | 67 L/MWh/day, based on DGA (2017) and based on Hardy & Garrido (2010). |





| | | Coal (413) | Gross energy produced in power plants, based on CNE (2019a, 2019b, 2019c, 2021a, 2021b, 2022). Production in 2020: 27,236,019 MWh | 85 L/MWh/day, based on DGA (2017) and based on Hardy & Garrido (2010). |
|---|---|---|---|---|
| | | Biomass (414) | Gross energy produced in power plants, based on CNE (2019a, 2019b, 2019c, 2021a, 2021b, 2022). Production in 2020: 1,866,415 MWh | 85 L/MWh/day, based on DGA (2017) and based on Hardy & Garrido (2010). |
| | | Geothermal (415) | Gross energy produced in power plants, based on CNE (2019a, 2019b, 2019c, 2021a, 2021b, 2022). Production in 2020: 246,876 MWh | 20 L/MWh/day, based on DGA (2017) and based on Hardy & Garrido (2010). |
| | Hydraulic (420) | Reservoir (421) | Gross energy produced in power plants, based on CNE (2019a, 2019b, 2019c, 2021a, 2021b, 2022). Production in 2020: 7,925,011 MWh | 12,271 L/MWh/day, based on DGA (2017). |
| | | ROR (422) | Gross energy produced in power plants, based on CNE (2019a, 2019b, 2019c, 2021a, 2021b, 2022). Production in 2020: 12,293,579 MWh | 11,338 L/MWh/day, based on DGA (2017). |
| | NCRE (430) | Wind (431) | Gross energy produced in power plants, based on CNE (2019a, 2019b, 2019c, 2021a, 2021b, 2022). Production in 2020: 5,510,712 MWh | - |
| | | Solar (432) | Gross energy produced in power plants, based on CNE (2019a, 2019b, 2019c, 2021a, 2021b, 2022). Production in 2020: 6,118,180 MWh | - |
| Industry (500) | Manufacture (510) | Food (511) | GDP in MUSD adjusted to 2020 USD value. Based on Banco Central (2020a, 2020b, 2020c, 2020d) and World Bank (2021). GDP in 2020: 7,276 MUSD | 67,741 L/USD/day, based on DGA (2017), Statistics Canada (2011) and SMA (2021). |
| | | Beverage (512) | GDP in MUSD adjusted to 2020 USD value. Based on Banco Central (2020a, 2020b, 2020c, 2020d) and World Bank (2021). GDP in 2020: 2,364 MUSD | 7,698 L/USD/day, based on DGA (2017), Statistics Canada (2011) and SMA (2021). |
| | | Cellulose (513) | GDP in MUSD adjusted to 2020 USD value. Based on Banco Central (2020a, 2020b, 2020c, 2020d) and World Bank (2021). GDP in 2020: 1,891 MUSD | 258,601 L/USD/day, based on DGA (2017), Statistics Canada (2011) and SMA (2021). |
| | | Wood (514) | GDP in MUSD adjusted to 2020 USD value. Based on Banco Central (2020a, 2020b, 2020c, 2020d) and World Bank (2021). GDP in 2020: 946 MUSD | 196,316 L/USD/day, based on DGA (2017), Statistics Canada (2011) and SMA (2021). |
| | | Chemistry ((515) | GDP in MUSD adjusted to 2020 USD value. Based on Banco Central (2020a, 2020b, 2020c, 2020d) and World Bank (2021). GDP in 2020: 2,407 MUSD | 100,951 L/USD/day, based on DGA (2017), Statistics Canada (2011) and SMA (2021). |
| | | Minerals and Metal Products (516) | GDP in MUSD adjusted to 2020 USD value. Based on Banco Central (2020a, 2020b, 2020c, 2020d) and World Bank (2021). GDP in 2020: 4,853 MUSD | 52177 L/USD/day, based on DGA (2017), Statistics Canada (2011) and SMA (2021). |
| | | Oil Refining (517) | GDP in MUSD adjusted to 2020 USD value. Based on Banco Central (2020a, 2020b, 2020c, 2020d) and World Bank (2021). GDP in 2020: 702 MUSD | 287,882 L/USD/day, based on DGA (2017), Statistics Canada (2011) and SMA (2021). |



| | | Textile (518) | GDP in MUSD adjusted to 2020 USD value. Based on Banco Central (2020a, 2020b, 2020c, 2020d) and World Bank (2021). GDP in 2020: 26 MUSD | 14,707 L/USD/day, based on DGA (2017), Statistics Canada (2011) and SMA (2021). |
|---|---|---|---|---|
| Livestock (600) | Bovine (610) | Cows (611) | Number of animals, based on INE (1959, 1969, 1983, 1998, 2008). Heads in 2020: 1,681,333 | 50 L/head/day, based on DGA (2017). |
| | | Heifers (612) | Number of animals, based on INE (1959, 1969, 1983, 1998, 2008). Heads in 2020: 464,642 | 27 L/head/day, based on DGA (2017). |
| | | Steers (613) | Number of animals, based on INE (1959, 1969, 1983, 1998, 2008). Heads in 2020: 520,108 | 35 L/head/day, based on DGA (2017). |
| | | Calves (614) | Number of animals, based on INE (1959, 1969, 1983, 1998, 2008). Heads in 2020: 1,049,858 | 23 L/head/day, based on DGA (2017). |
| | | Oxen (615) | Number of animals, based on INE (1959, 1969, 1983, 1998, 2008). Heads in 2020: 46,425 | 50 L/head/day, based on DGA (2017). |
| | | Bulls (616) | Number of animals, based on INE (1959, 1969, 1983, 1998, 2008). Heads in 2020: 55,395 | 50 L/head/day, based on DGA (2017). |
| | Equines (620) | Horses (621) | Number of animals, based on INE (1959, 1969, 1983, 1998, 2008). Heads in 2020: 210,029 | 45 L/head/day, based on DGA (2017). |
| | | Mules and Donkeys (622) | Number of animals, based on INE (1959, 1969, 1983, 1998, 2008). Heads in 2020: 15,752 | 45 L/head/day, based on DGA (2017). |
| | Camelids (630) | Llamas (631) | Number of animals, based on INE (1959, 1969, 1983, 1998, 2008). Heads in 2020: 47,382 | 3 L/head/day, based on DGA (2017). |
| | | Alpacas (632) | Number of animals, based on INE (1959, 1969, 1983, 1998, 2008). Heads in 2020: 31,429 | 3 L/head/day, based on DGA (2017). |
| | Birds (640) | Chickens (641) | Number of animals, based on INE (1959, 1969, 1983, 1998, 2008). Heads in 2020: 61,445,698 | 0.3 L/head/day, based on DGA (2017). |
| | | Turkeys (642) | Number of animals, based on INE (1959, 1969, 1983, 1998, 2008). Heads in 2020: 10,378,782 | 0.8 L/head/day, based on DGA (2017). |
| | | Gooses (643) | Number of animals, based on INE (1959, 1969, 1983, 1998, 2008). Heads in 2020: 102,368 | 0.5 L/head/day, based on DGA (2017). |
| | | Ducks (644) | Number of animals, based on INE (1959, 1969, 1983, 1998, 2008). Heads in 2020: 196,976 | 0.5 L/head/day, based on DGA (2017). |
| | | Other Birds (645) | Number of animals, based on INE (1959, 1969, 1983, 1998, 2008). Heads in 2020: 180,610 | 0.31 L/head/day, based on DGA (2017). |



| | Other (650) | Pigs (651) | Number of animals, based on INE (1959, 1969, 1983, 1998, 2008). Heads in 2020: 5,536,314 | 30 L/head/day, based on DGA (2017). |
|---|---|---|---|---|
| | | Sheeps (652) | Number of animals, based on INE (1959, 1969, 1983, 1998, 2008). Heads in 2020: 4,296,651 | 3.25 L/head/day, based on DGA (2017). |
| | | Goats (653) | Number of animals, based on INE (1959, 1969, 1983, 1998, 2008). Heads in 2020: 337,810 | 2.5 L/head/day, based on DGA (2017). |
| | | Rabbits (654) | Number of animals, based on INE (1959, 1969, 1983, 1998, 2008). Heads in 2020: 49,929 | 0.35 L/head/day, based on DGA (2017). |

(*)For classes associated with green areas (114 and 214), the total population counted only considered communes where data on water use for green area irrigation was available in SISS (2020a).
(**) For these classes, an auxiliary factor is applied that characterizes the percentage of water loss that is not billed. At the national level, the representative factor is 35% for urban uses, and 20% for rural uses. These variables are available in the associated Zenodo repository with different values for each commune.
(***) In the classes associated with silver and zinc production (314 and 317) there were limitations in finding the location of the respective mines. In the case
of silver, the national production is actually 1,580 Ton and for zinc it is 28,660 Ton. In the case of silver, the identification of the specific use is still pending, which is regularly associated with copper extraction, so the use of water is mixed.
(****) These rates are expressed per unit of fine metal. The utilization rates found for metal production are originally expressed per unit of ore processed, which is why it is necessary to have an auxiliary variable called ore grade. These variables are also available in https://doi.org/10.5281/zenodo.13324235 (Boisier et al., 2024b), with spatial and temporal variability for the core classes (311 and 312) and spatial variability for the other metals.

**Data availability**

The CR2LUC and CR2WU datasets developed in this study can be accessed from https://doi.org/10.5281/zenodo.13324250 and https://doi.org/10.5281/zenodo.13324235 (last access: 20 August 2024), respectively. The climate projections from downscaled ESMs can be shared upon request. The CR2MET dataset was obtained from https://doi.org/10.5281/zenodo.7529682 (last access: 20 September 2023). The water use rights were obtained from CAMELS-
CL dataset (Alvarez-Garreton et al., 2018).

**Author contributions**

JPB and CAG conceived the idea, designed the experimental setup, performed de analyses, prepared the figures and wrote an early draft of the manuscript. JPB developed CR2MET and ET simulations. JPB and RM developed the CR2LUC and CR2WU products. All the authors advised on the data production, experimental setup and wrote the final paper.

**Competing interests**

The contact author has declared that none of the authors has any competing interests.



**Acknowledgements**

This research was developed within the framework of the Center for Climate and Resilience Research (CR2, ANID/FONDAP/1523A0002) and the research projects ANID/FSEQ210001, FONDECYT/11190952, and

FONDECYT/11240924. We thank Mauricio Abel Herrera and Sergio Gonzalez for their support with LAI and CMIP6 model data processing. This publication has been prepared using European Union's Copernicus Land Monitoring Service information. We acknowledge the World Climate Research Programme, which, through its Working Group on Coupled Modelling, coordinated and promoted CMIP6. We thank the climate modeling groups for producing and making available their model output, the Earth System Grid Federation (ESGF) for archiving the data and providing access, and the multiple funding

agencies who support CMIP6 and ESGF.

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
