# Peer review of "Increasing water stress in Chile revealed by novel datasets of water availability, land use and water use"

_EGUsphere, 2024_

## Community Comment (CC1)

**Review of "Increasing water stress in Chile evidenced by novel datasets of water availability, land use and water use" by Boisier, J. P., Alvarez-Garreton, C., Marinao, R., & Galleguillos, M.**

*Kyra Boek*

**I.     SUMMARY, CONTRIBUTION AND RECOMMENDATION**

The manuscript by Boisier et al.  assesses water stress in Chile from the mid-20th century to the end of the 21st century under various scenarios, specifically focusing on the megadrought from 2010 to 2022. The authors developed spatially distributed national-scale datasets on hydroclimatic data, water availability, land use, and water use. They used these to calculate the Water Stress Index (WSI) for all major basins to quantify the intensity of water stress during past, present, and future time periods. Findings indicate high to extreme water stress in semi-arid regions during the megadrought, with future projections indicating worsening conditions due to increased water consumption and reduced availability. The study suggests using the WSI for effective water security assessment and adaptation planning.

The novelty of this research lies in its integration of newly created high-resolution datasets with long-term projections, providing an overview of water stress in Chile. This is interesting, as not many papers focus on water stress in a whole country (but rather in a regional or basin-specific context); especially not a country that is a large as Chile. If successful, this approach can inform national-scale policymaking and improve water management strategies across varying climate zones.

The manuscript is well-written, with clear and concise language that effectively communicates complex concepts. The structure is logical, making it easy to follow the progression of the research and its findings. The figures are displayed clearly, enhancing the understanding of the data and results.

In principle, the study fits the scope of HESS, as it uses an interdisciplinary approach to look at spatial and temporal characteristics of water resources in Chile and gives concrete examples of socio-economic impacts of water stress. The use of novel datasets also contributes to the advancement of hydrological modelling. However, the use of these novel datasets in the context of the WSI – a simple index which is typically used on a continental or global scale – may reduce the robustness of the study's outcomes, especially given the lack of model validation and sensitivity analysis. Therefore, while this study provides a comprehensive initial indication of water stress in Chile, I believe there are several potential shortcomings that might need to be addressed before final publication in HESS. In addition, I will discuss

several smaller issues and suggestions that I encountered during my study of the manuscript and that I believe could benefit the authors in improving their work.

**II.   GENERAL COMMENTS**

Firstly, I question the applicability of the WSI at a national scale in this study. While commonly used for continental or global assessments (e.g. Gosling & Arnell, 2013; Pfister & Bayer, 2013), its use on regional or national scales is limited. When used on smaller scales, the WSI typically focuses on specific contexts, such as food security (Gheewala et al., 2014) or cross-region comparisons (Milano et al., 2013). However, this study goes further by suggesting the WSI can inform policy, but as Milano et al. (2013) note, sub-regional studies with local stakeholder input and detailed catchment data are essential for effective water management. The WSI can compare future water stress across regions but provides only a rough indication of stress, missing key regional differences in water use, governance, and access, which are key for effective water management. Although the authors suggest that water security goals can be achieved through metrics like the WSI (p1, line 27; p22, line 498), they do not specify which complementary metrics are necessary for a complete assessment. To strengthen the study, the authors could discuss the limitations of the WSI and recommend using more detailed hydrological models (e.g., SWAT, VIC) and additional metrics to better address regional variability.

The study also lacks model validation and sensitivity analysis for the WSI and key variables. Validating the WSI output against (independent) previous estimates – for example by using the Nash-Sutcliffe Efficiency or Kling-Gupta Efficiency – would improve the scientific contribution. Validating the output of the current model to those obtained in previous studies that followed similar methodologies would provide insights into the consistency and reliability of the model under different conditions. For instance, the assessment by Ferreira et al. (2023) on precipitation and hydrological droughts in South America and work by Alvarez-Garreton et al. (2022) in which the WSI was calculated across 277 basins in Chile could serve as valuable references. Additionally, the study by Perveen and James (2011) on the scale invariance of water stress and scarcity indicators provides a critical perspective on the robustness of these indicators across different spatial scales, highlighting the need for such validation.

Furthermore, the results of the WSI are not supported by a sensitivity analysis, leaving it unclear how variations in water availability, land use, or water demand affect the final WSI values. Adding a sensitivity analysis would give a clearer understanding of the model's reliability under different conditions. An example of how to implement such a sensitivity analysis can be found in the paper by Milano et al. (2013), where a regional sensitivity analysis is performed. The authors could add this analysis either in the methodology section or as part of the results, explaining the specific assumptions tested and the range of variability explored.

A second general comment concerns the use of the Hargreaves-Samani (HS) formula to estimate potential evapotranspiration (PET). Since this formula is primarily based on temperature, this raises concerns about its suitability for projecting future PET under changing climate conditions. Previous studies (e.g. Alexandris et al., 2008; Temesgen et al., 2005; De Souza Nóia Júnior et al., 2019) show that the HS method tends to systematically overestimate PET by up to 20%, particularly in regions with high average temperature, wind speed, or relative humidity. This overestimation is a critical issue when using the HS formula for long-term projections, as the projected increase in temperature may lead to an overestimation of future drought conditions. While the authors adjust for wind biases and align HS with

the Penman-Monteith (PM) formula, uncertainties remain, particularly for long-term trend projections. I suggest adopting the PM method or conducting an uncertainty analysis to assess the reliability of HS estimates. Although Appendix B validates the ET simulations, it doesn't fully address how the corrected HS method performs over time, especially under future climate conditions. A validation process using the same datasets and temporal scales as the analysis would enhance confidence in the projections.

The authors should also clarify whether the novel land and water use datasets were cross validated with independent datasets to enhance the credibility of the findings. Additionally, including uncertainty ranges for key variables, such as precipitation and water demand, would further strengthen the results by providing a clearer understanding of potential limitations and the confidence in the model's outcomes.

Finally, the study does not clarify whether water availability is estimated based on hydrological years or calendar years. This distinction is important since omitting storage changes between years can lead to inaccuracies in estimating water availability, particularly in regions where hydrological processes are influenced by large seasonality. Other studies on droughts (e.g. Xiao et al., 2017; Yuan et al., 2024) use hydrological years (or water years) to ensure that as much as possible of the surface runoff during that (hydrological) year is attributable to the precipitation during the same hydrological year. I would advise to use hydrological years for the hydrological balance (Section 2.3, p6-7), or to explain the use of calendar years. Given Chile's diverse climate, the authors should consider whether a single hydrological year is appropriate nationwide or if region-specific years should be applied. At a minimum, a brief discussion on this issue, with suggestions for future research, would improve the study's clarity.

**III. SPECIFIC COMMENTS**

Firstly, I randomly fact-checked some of the statements made by the authors. The results of this check can be found in the following two specific comments.

i.    WSI threshold value

The use of the value of 0.4 (or 40%, line 41-42, p2) to indicate water-stressed regions seems arbitrary, as some studies use a value of 0.2-0.4 to indicate moderate water stress (e.g. Milano et al., 2013; Gosling & Arnell, 2013) or even 0.1-0.5 (Gheewala et al., 2014). Oki and Kanae (2006) describe 0.4 as "a reasonable, although not definitive, threshold value." In the current paper, the threshold of 0.4 seems to be definitive, but the authors should clarify that this is not the case. While in line 499-500 (p22) the authors say that "regarding the WSI, stress levels are well-defined worldwide in relation to their impacts on watersheds," the five references that are listed to support this statement do not carry the same information:

- Falkenmark, 2013a: this paper talks about water stress in relation to climate change on national and regional scale, but it does not mention WSI levels nor their impact on the watershed-scale.
- Falkenmark, 2013b: this paper also talks about consequences of water stress, but again it does not mention the WSI (or any other index).
- Rockstrom et al., 2014: although the WSI is mentioned (as Falkenmark Index), the paper does not mention stress levels in relation to their impact on watersheds. General problems regarding water scarcity are listed, but it seems like the WSI is used to give a general idea of water stress worldwide rather than a national-scale/watershed-scale indicator.
- Oki and Kanae, 2006: this paper uses the WSI on a global scale.
- Grafton et al., 2012: this paper is not listed in the reference section at all.

A simple solution to this problem is to include a discussion on the value of 0.4 as a threshold, and to remove the statement that WSI stress levels are well-defined worldwide in relation to their impacts on watersheds. Moreover, if the authors still wish to use the paper by Grafton et al., 2012, they should include this paper in the reference section.

ii.     Total water use in Chile at present

In line 348-349 (p14), the authors state: "Considering both consumptive and non-consumptive uses, the total water use in Chile is estimated to be around 100 km$^3$ per year at present. This value is similar to, albeit slightly higher than, other independent estimates." I checked the three references that were used for this statement.

- DGA, 2017: this document is written in Spanish but if I am not mistaken, I would say that based on "Cuadro 4.17-1" and "Cuadro 4.17-2", p47 & 48, the total water use is over 160 km$^3$/year (sum of "Demanda Consuntiva 2015" and "Demanda No Consuntiva 2015" (consumptive and non-consumptive demand)). These numbers are based on a calculation of the water demand per district.
- Fundacion Chile: Based on table 5, the "registered" water consumption is 3.335,44 m3/s which corresponds roughly to 105 km$^3$/y. This is the water consumption based on the Water Use Rights as mentioned in the current paper, so the actual water consumption is likely much higher than that.
- FAO and UN Water, 2021: This source only shows water stress per continent so I am not sure where they found a specific number for Chile.

If my conclusions based on these documents are correct, the authors can omit the problem by stating that the value is *lower* than other independent estimates.

iii.     Research questions

The research questions could be more specific. Instead of "What have been the historical water stress levels of the basins in Chile?" (p3, line 78), it could specify: "What have been the historical water stress levels of the major basins in Chile according to the Water Stress Index (WSI)?" In the second research question (p4, line 78-79), it is asked what the drivers of changes in water stress levels are. While the authors look at climatic and anthropogenic drivers, they do not specifically look at factors such as population growth and technological advances. Therefore, I suggest to specify the drivers that this study looks at. E.g., "What have been the climatic and anthropogenic drivers of changes in these levels?" or: "What have been the drivers of changes in water stress levels in Chile, and how have factors such as climate change, land use, and water use contributed to these changes?"

iv.     Final remarks

The term "significant(ly)" occurs 21 times. This word suggests that these claims have statistical validation, but this is usually not the case. Consider replacing this word with terms like "substantial(ly)" or "notable(ly)" where appropriate (e.g., p14, line 353; p17, line 392; p18, line 436; p20, line 460)

p1, line 1: the use of the word "evidenced" in the title conveys the idea that the increase in water stress is not merely observed but backed by strong, concrete data. In my opinion, this level of certainty is insufficient supported by the research methodology. A simple solution to this problem is to replace "evidenced by" by a term like "indicated by" or "revealed through". This way, the title still reflects that the study sheds light on the issue but does not claim to provide absolute evidence.

p1, line 13: Clarify why the term "megadrought" is used, as it is not explained in the introduction.

p1, line 15: the authors mention that water-intensive agriculture raises questions about the contributions of water extraction to high stress levels. However, they do not come back to this question. Therefore, I think it would make more sense to ask about the contribution of water consumption, as this is what the paper is largely about. Alternatively, the authors could mention the answer to this original question in the conclusion section.

p1, line 15: "the contributions (…) _on_ high water stress levels"; should be "to".

p1, line 26: using the WSI to assess "one" of the several aspects of water security seems like a vague description; what aspect of water security is assessed with the WSI?

p2, line 32: while this definition is important, the opening of the introduction could be stronger by starting with a statement related to water security to catch the reader's attention.

p2, line 41-42: the authors could consider providing a table with an overview of WSI thresholds (no/high/extreme water stress).

p3, line 65: "decade-long drought", what type of drought is referred to? E.g. hydrological/meteorological/socio-economic/…

p3, line 67: what is meant by Water Use Rights?

p3, line 72: Groundwater overexploitation is mentioned as an important contributor to water stress, but it is not fully addressed in the paper. Consider including it in the discussion or conclusion.

p4, Table 1, line 96: "irrigation fraction ($I_V$) fraction"; the word "fraction" appears twice.

p6, line 162: "To ensure accurate seasonal representation, (…)"; does this also include important weather phenomena like El Niño?

p6, line 164: TN should be $T_N$ (in subscript).

p7, line 183: "account" should be "accounts".

p7, line 193: "Long-term land-use dynamics _is_ (…)"; should be "are".

p7, line 193-194: "many other factors"; could the authors give an example?

p9, line 245: "is" should be "are".

p9, line 247: Subscript "IR" unclear (it looks like eIR now).

p9, line 250: UNC,LU should be $U_{NC,LU}$ (in subscript).

p10, line 286: "regions in the globe" should be "regions of the globe"

p12, Figure 2: the color used to indicate a ΔP of +25 and -25 mm/yr are too similar. This problem can be solved by making the values above 0 greener (rather than yellow).

p12, line 311: the authors refer to Fig. 2 but this should be Fig. 3.

p13, line 311-313: "The spatially distributed CR2MET dataset shows a precipitation decline consistent with observations, a match that matters given the subsequent use of this dataset for basin-scale assessments." Is this statement based on a statistical test or purely on visual analysis? The authors say that it matters that the observations match the CR2MET data, but this statement would be stronger if they could show that these datasets show significant correlation.

p13, line 319-328: Could the Andes Cordillera also explain discrepancies between this study and others, given the hydroclimatic effects noted earlier (p2, lines 49-50)?

p15, line 372 and 373: "litters" should be "litres" or "liters".

p23, line 524: "water-demand" should be "water demand" to match other occurrences of this term in the document.

p24, Table 2: in the first scenario description, "(2020-2020)" should be "(2000-2020)"

p25, line 568: "near 30% reduction" should be "nearly a 30% reduction".

p25, line 586-591: the conclusion could be slightly improved by concluding with a strong final statement, e.g. something along the lines of "Achieving long-term water security in Chile will require not only better governance and infrastructure but also a commitment to sustainable water use practices across all sectors. This study provides critical insights into how such goals can be realized."

**IV.   REFERENCES**

Adler, R.; Wang, J.J.; Sapiano, M.; Huffman, G.; Chiu, L.; Xie, P.P.; Ferraro, R.; Schneider, U.; Becker, A.; Bolvin, D.; Nelkin, E.; Gu, G.; and NOAA CDR Program (2016). Global Precipitation Climatology Project (GPCP) Climate Data Record (CDR), Version 2.3 (Monthly). National Centers for Environmental Information. http://doi.org/10.7289/V56971M6

Alexandaris, S.; Stricevic, R.; Petkovic, S. (2008). Comparative analysis of reference evapotranspiration from the surface of rainfed grass in central Serbia calculated by six empirical methods against the Penman-Monteith formula. *Eur. Water*, *21*, 17–28

Alvarez-Garreton, C., Boisier, J. P., Billi, M., Lefort, I., Marinao, R., & Barría, P. (2022). Protecting environmental flows to achieve long-term water security. *Journal of Environmental Management*, *328*, 116914. https://doi.org/10.1016/j.jenvman.2022.116914

De Souza Ferreira, G. W., Reboita, M. S., Ribeiro, J. G. M., & De Souza, C. A. (2023). Assessment of Precipitation and Hydrological Droughts in South America through Statistically Downscaled CMIP6 Projections. *Climate*, *11*(8), 166. https://doi.org/10.3390/cli11080166

De Souza Nóia Júnior, R., Fraisse, C. W., Cerbaro, V. A., Karrei, M. a. Z., & Guindin, N. (2019). Evaluation of the Hargreaves-Samani Method for Estimating Reference

DGA. (2017). Estimación de la demanda actual, proyecciones futuras y caracterización de la calidad de los recursos hídricos en Chile. SIT 419. Realizado por: Unión temporal de proveedores Hídrica Consultores SPA y Aquaterra ingenieros LTDA.

Falkenmark, M. (2013a). Adapting to climate change: towards societal water security in dry-climate countries. *International Journal of Water Resources Development*, *29*(2), 123–136. https://doi.org/10.1080/07900627.2012.721714

Falkenmark, M. (2013b). Growing water scarcity in agriculture: future challenge to global water security. *Philosophical Transactions of the Royal Society a Mathematical Physical and Engineering Sciences*, *371*(2002), 20120410. https://doi.org/10.1098/rsta.2012.0410

FAO: El estado de los recursos de tierras y aguas del mundo para la alimentación y la agricultura - Sistemas al límite. (2021). https://doi.org/10.4060/cb7654es

Fundacion Chile: Radiografia del Agua: Brecha y Riesgo Hidrico en Chile. (2018). https://escenarioshidricos.cl/publicacion/radiografia-del-agua-brecha-y-riesgo-hidrico-en-chile

Gheewala, S., Silalertruksa, T., Nilsalab, P., Mungkung, R., Perret, S., & Chaiyawannakarn, N. (2014). Water footprint and impact of water consumption for food, feed, fuel crops production in Thailand. *Water*, *6*(6), 1698–1718. https://doi.org/10.3390/w6061698

Gosling, S. N., & Arnell, N. W. (2013). A global assessment of the impact of climate change on water scarcity. *Climatic Change*, *134*(3), 371–385. https://doi.org/10.1007/s10584-013-0853-x

Milano, M., Ruelland, D., Fernandez, S., Dezetter, A., Fabre, J., Servat, E., Fritsch, J., Ardoin-Bardin, S., & Thivet, G. (2013). Current state of Mediterranean water resources and future trends under climatic and anthropogenic changes. *Hydrological Sciences Journal*, *58*(3), 498–518. https://doi.org/10.1080/02626667.2013.774458

Oki, T., & Kanae, S. (2006). Global hydrological cycles and world Water Resources. *Science*, *313*(5790), 1068–1072. https://doi.org/10.1126/science.1128845

Pfister, S., & Bayer, P. (2013). Monthly water stress: spatially and temporally explicit consumptive water footprint of global crop production. *Journal of Cleaner Production*, *73*, 52–62. https://doi.org/10.1016/j.jclepro.2013.11.031

Rockström, J., Falkenmark, M., Allan, T., Folke, C., Gordon, L., Jägerskog, A., Kummu, M., Lannerstad, M., Meybeck, M., Molden, D., Postel, S., Savenije, H., Svedin, U., Turton, A., & Varis, O. (2014). The unfolding water drama in the Anthropocene: towards a resilience-based perspective on water for global sustainability. *Ecohydrology*, *7*(5), 1249–1261. https://doi.org/10.1002/eco.1562

Temesgen, B., Eching, S., Davidoff, B., & Frame, K. (2005). Comparison of Some Reference Evapotranspiration Equations for California. *Journal of Irrigation and Drainage Engineering, 131*(1), 73-84. https://doi.org/10.1061/(ASCE)0733-9437(2005)131:1(73)

Xiao, M., Koppa, A., Mekonnen, Z., Pagán, B. R., Zhan, S., Cao, Q., Aierken, A., Lee, H., & Lettenmaier, D. P. (2017). How much groundwater did California's Central Valley lose during the 2012–2016 drought? *Geophysical Research Letters*, *44*(10), 4872–4879. https://doi.org/10.1002/2017gl073333

Yuan, R., Xu, R., Zhang, H., Qiu, C., & Zhu, J. (2024). Satellite-Derived indicators of drought severity and water storage in estuarine reservoirs: a case study of Qingcaosha Reservoir, China. *Remote Sensing*, *16*(6), 980. https://doi.org/10.3390/rs16060980

---

## Author Comment (AC3)

**EGUSPHERE-2024-2695**

**Responses to Community Comments**

This document provides the responses (in blue) to community comments.

Juan P. Boisier, on behalf of all co-authors.

**Comments by Kyra Boek**

I. SUMMARY, CONTRIBUTION AND RECOMMENDATION
The manuscript by Boisier et al. assesses water stress in Chile from the mid-20th century to the end of the 21st century under various scenarios, specifically focusing on the megadrought from 2010 to 2022. The authors developed spatially distributed national-scale datasets on hydroclimatic data, water availability, land use, and water use. They used these to calculate the Water Stress Index (WSI) for all major basins to quantify the intensity of water stress during past, present, and future time periods. Findings indicate high to extreme water stress in semi-arid regions during the megadrought, with future projections indicating worsening conditions due to increased water consumption and reduced availability. The study suggests using the WSI for effective water security assessment and adaptation planning.

The novelty of this research lies in its integration of newly created high-resolution datasets with long-term projections, providing an overview of water stress in Chile. This is interesting, as not many papers focus on water stress in a whole country (but rather in a regional or basin-specific context); especially not a country that is a large as Chile. If successful, this approach can inform national-scale policymaking and improve water management strategies across varying climate zones.

The manuscript is well-written, with clear and concise language that effectively communicates complex concepts. The structure is logical, making it easy to follow the progression of the research and its findings. The figures are displayed clearly, enhancing the understanding of the data and results.

In principle, the study fits the scope of HESS, as it uses an interdisciplinary approach to look at spatial and temporal characteristics of water resources in Chile and gives concrete examples of socio-economic impacts of water stress. The use of novel datasets also contributes to the advancement of hydrological modelling. However, the use of these novel datasets in the context of the WSI – a simple index which is typically used on a continental or global scale – may reduce the robustness of the study's outcomes, especially given the lack of model validation and sensitivity analysis. Therefore, while this study provides a comprehensive initial indication of water stress in Chile, I believe there are several potential shortcomings that might need to be addressed before final publication in HESS. In addition, I will discuss several smaller issues and suggestions that I encountered during my study of the manuscript and that I believe could benefit the authors in improving their work.

R: We appreciate the recognition of our paper's strengths and the constructive feedback on its limitations, which we respond below.

II. GENERAL COMMENTS

1) Firstly, I question the applicability of the WSI at a national scale in this study. While commonly used for continental or global assessments (e.g. Gosling & Arnell, 2013; Pfister & Bayer, 2013), its use on regional or national scales is limited. When used on smaller scales, the WSI typically focuses on

specific contexts, such as food security (Gheewala et al., 2014) or cross-region comparisons (Milano et al., 2013). However, this study goes further by suggesting the WSI can inform policy, but as Milano et al. (2013) note, sub-regional studies with local stakeholder input and detailed catchment data are essential for effective water management. The WSI can compare future water stress across regions but provides only a rough indication of stress, missing key regional differences in water use, governance, and access, which are key for effective water management. Although the authors suggest that water security goals can be achieved through metrics like the WSI (p1, line 27; p22, line 498), they do not specify which complementary metrics are necessary for a complete assessment. To strengthen the study, the authors could discuss the limitations of the WSI and recommend using more detailed hydrological models (e.g., SWAT, VIC) and additional metrics to better address regional variability.

R1: We agree with the reviewer on the limitations of using WSI to establish water security goals. In our responses to the Referees 1 and 2, we address these limitations and propose modifications in the revised manuscript. We invite the reviewer to read the response document, particularly R4 to RC1 and R2 to RC2.

2) The study also lacks model validation and sensitivity analysis for the WSI and key variables. Validating the WSI output against (independent) previous estimates – for example by using the Nash-Sutcliffe Efficiency or Kling-Gupta Efficiency – would improve the scientific contribution. Validating the output of the current model to those obtained in previous studies that followed similar methodologies would provide insights into the consistency and reliability of the model under different conditions. For instance, the assessment by Ferreira et al. (2023) on precipitation and hydrological droughts in South America and work by Alvarez-Garreton et al. (2022) in which the WSI was calculated across 277 basins in Chile could serve as valuable references. Additionally, the study by Perveen and James (2011) on the scale invariance of water stress and scarcity indicators provides a critical perspective on the robustness of these indicators across different spatial scales, highlighting the need for such validation.

R2: Rather than a model, the WSI is a metric derived from water availability and use data—datasets that are themselves estimated with models. We agree that, beyond what is presented in Appendix B, a more detailed validation of the datasets developed in our study would help better assess uncertainty.

Directly comparing WSI values from our study with those reported in previous studies is not feasible due to significant methodological differences. Alvarez-Garreton et al. (2022) estimated WSI based on allocated water rights rather than actual water consumption. This approach can significantly misrepresent actual water use, and addressing that limitation is one of the key contributions of our work. Beyond global assessments, we are not aware of other peer-reviewed publications that assess the WSI across Chile.

Regarding a more detailed comparison of datasets, we refer the reviewer to our responses to Referee 1—specifically R6, R7, R8, and R9 for water availability, and R6 for water use benchmarks. Regarding the latter, in the revised manuscript we will include a new table summarizing national water use totals by major sector and comparing them with estimates from other sources (see R6 to RC1 for further details).

3) Furthermore, the results of the WSI are not supported by a sensitivity analysis, leaving it unclear how variations in water availability, land use, or water demand affect the final WSI values. Adding a sensitivity analysis would give a clearer understanding of the model's reliability under different conditions. An example of how to implement such a sensitivity analysis can be found in the paper by Milano et al. (2013), where a regional sensitivity analysis is performed. The authors could add this analysis either in the methodology section or as part of the results, explaining the specific assumptions tested and the range of variability explored.

R3: We agree that a sensitivity analysis of the WSI would be useful. However, this type of analysis falls outside the scope of the present manuscript, which already includes a substantial amount of new data and analysis. In this study, we focused our efforts on developing new datasets on water availability and water use in Chile, with a spatio-temporal resolution that did not previously exist. We evaluated these datasets through mass balance assessments at the basin scale (see response R9 to RC1) and by comparing our water use estimates with other available sources (see response R6 to RC1). A full uncertainty analysis of the WSI would require error estimation for the underlying datasets, which is beyond the scope of this study. However, we acknowledge its importance and consider it a valuable direction for future work.

4) A second general comment concerns the use of the Hargreaves-Samani (HS) formula to estimate potential evapotranspiration (PET). Since this formula is primarily based on temperature, this raises concerns about its suitability for projecting future PET under changing climate conditions. Previous studies (e.g. Alexandris et al., 2008; Temesgen et al., 2005; De Souza Nóia Júnior et al., 2019) show that the HS method tends to systematically overestimate PET by up to 20%, particularly in regions with high average temperature, wind speed, or relative humidity. This overestimation is a critical issue when using the HS formula for long-term projections, as the projected increase in temperature may lead to an overestimation of future drought conditions. While the authors adjust for wind biases and align HS with the Penman-Monteith (PM) formula, uncertainties remain, particularly for long-term trend projections. I suggest adopting the PM method or conducting an uncertainty analysis to assess the reliability of HS estimates. Although Appendix B validates the ET simulations, it doesn't fully address how the corrected HS method performs over time, especially under future climate conditions. A validation process using the same datasets and temporal scales as the analysis would enhance confidence in the projections.

R4: We largely agree that a more physically constrained formulation of PET is desirable. The approach to PET estimation was extensively discussed within our team during the methodological design stage. The method we ultimately adopted aimed to meet several objectives—most importantly, to provide a consistent way to estimate PET for both the historical period and future projections. The reference atmospheric data is based on CR2MET, a dataset designed to accurately capture precipitation and temperature over continental Chile. However, it does not include other variables required to compute Penman-Monteith PET (wind speed and humidity). Similarly, for climate model projections, these variables were not considered due to limitations in CR2MET (which served as the reference dataset for downscaling) and, in part, to enable the evaluation of multiple climate models for uncertainty assessment, rather than restricting the analysis to those providing a broader set of variables (we note that the processed data for this study sum various TBs in volume).

Given these constraints, we chose to estimate PET using temperature data via the Hargreaves-Samani method, and to partially improve this estimate through a climatological adjustment using Penman-Monteith, as described in Section 2.1. We acknowledge that this approach does not capture certain effects, such as changes in wind speed over time, which can influence PET. However, for the purposes of this paper, we estimate that such effects are of second-order importance. Although ET in some regions of Chile is energy-limited, in general, ET variability is largely controlled by soil moisture within the region of study, which is primarily driven by precipitation. Additionally, changes in ET have less influence on water availability than changes in precipitation. Properly assessing these mechanisms within a contrasting territory like Chile is of great interest but would require a regional-scale coupled model simulations, rather than statistically downscaled ESM data, to account for land–atmosphere feedbacks. This, however, is far beyond the scope of this paper.

A revised text will state more clearly the limitations on the method adopted for estimating PET.

5) The authors should also clarify whether the novel land and water use datasets were cross validated with independent datasets to enhance the credibility of the findings. Additionally, including uncertainty ranges for key variables, such as precipitation and water demand, would further

strengthen the results by providing a clearer understanding of potential limitations and the confidence in the model's outcomes.

R5: We agree. Please see our response on this matter above (R2).

6) Finally, the study does not clarify whether water availability is estimated based on hydrological years or calendar years. This distinction is important since omitting storage changes between years can lead to inaccuracies in estimating water availability, particularly in regions where hydrological processes are influenced by large seasonality. Other studies on droughts (e.g. Xiao et al., 2017; Yuan et al., 2024) use hydrological years (or water years) to ensure that as much as possible of the surface runoff during that (hydrological) year is attributable to the precipitation during the same hydrological year. I would advise to use hydrological years for the hydrological balance (Section 2.3, p6-7), or to explain the use of calendar years. Given Chile's diverse climate, the authors should consider whether a single hydrological year is appropriate nationwide or if region-specific years should be applied. At a minimum, a brief discussion on this issue, with suggestions for future research, would improve the study's clarity.

R6: CR2MET as well as the mass balance model used to estimate evapotranspiration operates on a daily scale. Annual computations of water availability, or for water use data, are based on calendar years. Please note that the choice between hydrological and calendar years is not particularly relevant in this study, as the results are presented in terms of climatological (multi annual) averages.

III. SPECIFIC COMMENTS

Firstly, I randomly fact-checked some of the statements made by the authors. The results of this check can be found in the following two specific comments.

i. WSI threshold value

The use of the value of 0.4 (or 40%, line 41-42, p2) to indicate water-stressed regions seems arbitrary, as some studies use a value of 0.2-0.4 to indicate moderate water stress (e.g. Milano et al., 2013; Gosling & Arnell, 2013) or even 0.1-0.5 (Gheewala et al., 2014). Oki and Kanae (2006) describe 0.4 as "a reasonable, although not definitive, threshold value." In the current paper, the threshold of 0.4 seems to be definitive, but the authors should clarify that this is not the case. While in line 499-500 (p22) the authors say that "regarding the WSI, stress levels are well-defined worldwide in relation to their impacts on watersheds," the five references that are listed to support this statement do not carry the same information:

• Falkenmark, 2013a: this paper talks about water stress in relation to climate change on national and regional scale, but it does not mention WSI levels nor their impact on the watershed-scale.

• Falkenmark, 2013b: this paper also talks about consequences of water stress, but again it does not mention the WSI (or any other index).

• Rockstrom et al., 2014: although the WSI is mentioned (as Falkenmark Index), the paper does not mention stress levels in relation to their impact on watersheds. General problems regarding water scarcity are listed, but it seems like the WSI is used to give a general idea of water stress worldwide rather than a national-scale/watershed-scale indicator.

• Oki and Kanae, 2006: this paper uses the WSI on a global scale.

• Grafton et al., 2012: this paper is not listed in the reference section at all.

A simple solution to this problem is to include a discussion on the value of 0.4 as a threshold, and to remove the statement that WSI stress levels are well-defined worldwide in relation to their impacts on watersheds. Moreover, if the authors still wish to use the paper by Grafton et al., 2012, they should include this paper in the reference section.

We generally agree with the reviewer. Regarding the 40% WSI thresholds, please refer to our responses to related referee comments, particularly R2 to RC2. We will remove Grafton et al., 2012 from the text, as it was a mistake. Thank you for pointing that out. The other references are being adapted in our proposed manuscript revision. In particular, we will replace Falkenmark´s 2013 papers with Falkenmark et al. (2007) and remove Rockström et al. (2014) in this context (see the list of new references in our response to the referee comments).

ii. Total water use in Chile at present

In line 348-349 (p14), the authors state: "Considering both consumptive and non-consumptive uses, the total water use in Chile is estimated to be around 100 km3 per year at present. This value is similar to, albeit slightly higher than, other independent estimates." I checked the three references that were used for this statement.

• DGA, 2017: this document is written in Spanish but if I am not mistaken, I would say that based on "Cuadro 4.17-1" and "Cuadro 4.17-2", p47 & 48, the total water use is over 160 km3/year (sum of "Demanda Consuntiva 2015" and "Demanda No Consuntiva 2015" (consumptive and non-consumptive demand)). These numbers are based on a calculation of the water demand per district.

• Fundacion Chile: Based on table 5, the "registered" water consumption is 3.335,44 m3/s which corresponds roughly to 105 km3/y. This is the water consumption based on the Water Use Rights as mentioned in the current paper, so the actual water consumption is likely much higher than that.

• FAO and UN Water, 2021: This source only shows water stress per continent so I am not sure where they found a specific number for Chile.

If my conclusions based on these documents are correct, the authors can omit the problem by stating that the value is lower than other independent estimates.

This was addressed in our responses to the referee comments. In particular, we are adding a new table that includes a proper comparison with those datasets. Please refer to our response R6 to RC1 for further details.

iii. Research questions

The research questions could be more specific. Instead of "What have been the historical water stress levels of the basins in Chile?" (p3, line 78), it could specify: "What have been the historical water stress levels of the major basins in Chile according to the Water Stress Index (WSI)?" In the second research question (p4, line 78-79), it is asked what the drivers of changes in water stress levels are. While the authors look at climatic and anthropogenic drivers, they do not specifically look at factors such as population growth and technological advances. Therefore, I suggest to specify the drivers that this study looks at. E.g., "What have been the climatic and anthropogenic drivers of changes in these levels?" or: "What have been the drivers of changes in water stress levels in Chile, and how have factors such as climate change, land use, and water use contributed to these changes?"

We agree and change the formulation of these questions:

> (...). Understanding the causes of water stress is critical for designing strategies aimed at mitigating unsustainable water uses and adapting to future climate conditions. **Therefore, this study aims to fill this gap by addressing the following research questions: What have the historical water stress levels of the basins in Chile been, according to the water use-to-availability ratio? How have trends in climate and water use influenced changes in these levels?** What can be expected under future climate scenarios?

iv. Final remarks

The term "significant(ly)" occurs 21 times. This word suggests that these claims have statistical validation, but this is usually not the case. Consider replacing this word with terms like

"substantial(ly)" or "notable(ly)" where appropriate (e.g., p14, line 353; p17, line 392; p18, line 436; p20, line 460)

We agree with the suggestion and will carefully review the text. We will adjust or replace instances where "significant(ly)" may have been misleading, opting for the suggested terms such as "substantial(ly)" or "notable(ly)" where appropriate.

p1, line 1: the use of the word "evidenced" in the title conveys the idea that the increase in water stress is not merely observed but backed by strong, concrete data. In my opinion, this level of certainty is insufficient supported by the research methodology. A simple solution to this problem is to replace "evidenced by" by a term like "indicated by" or "revealed through". This way, the title still reflects that the study sheds light on the issue but does not claim to provide absolute evidence.

We agree and will take the reviewer's suggestion for the title: *Increasing Water Stress in Chile Revealed by Novel Datasets of Water Availability, Land Use, and Water Use.*

p1, line 13: Clarify why the term "megadrought" is used, as it is not explained in the introduction.

We will mention the megadrought in the revised introduction:

> *(...). This situation is partly due to a long-term drying trend and decreased water availability in the region (Boisier et al., 2018), which has been exacerbated by a decade-long **meteorological** drought since 2010, **the so-called megadrought** (Boisier et al., 2016; Garreaud et al., 2017, 2019). (...)*

p1, line 15: the authors mention that water-intensive agriculture raises questions about the contributions of water extraction to high stress levels. However, they do not come back to this question. Therefore, I think it would make more sense to ask about the contribution of water consumption, as this is what the paper is largely about. Alternatively, the authors could mention the answer to this original question in the conclusion section.

We do not fully understand this comment. In what sense do we not come back to this question? The influence of water use (or consumption) on water stress levels is central to the study, including the particular role of irrigated agriculture.

p1, line 15: "the contributions (...) on high water stress levels"; should be "to".

Corrected.

p1, line 26: using the WSI to assess "one" of the several aspects of water security seems like a vague

description; what aspect of water security is assessed with the WSI?

This point is discussed in detail in various sections of the main text. Pease see our response letter to referees, particularly R2 to RC2. In the abstract, we opted to keep it concise.

p2, line 32: while this definition is important, the opening of the introduction could be stronger by starting with a statement related to water security to catch the reader's attention.

Thank you for the suggestion, but we prefer to use the definition provided in the literature to introduce the topic.

p2, line 41-42: the authors could consider providing a table with an overview of WSI thresholds (no/high/extreme water stress).

In the revised manuscript we will provide a clearer explanation of these thresholds (please refer to our response R2 to RC2). We prefer not to include additional tables in the main text, as these levels are described in several published papers and documents.

p3, line 65: "decade-long drought", what type of drought is referred to? E.g. hydrological/ meteorological/socio-economic/…

It is hydrometeorological, but mostly described from a meteorological perspective. We will add meteorological in this sentence.

p3, line 67: what is meant by Water Use Rights?

We will explain water use rights in the revised manuscript:

> (…). Water scarcity issues have also been attributed to limitations in the water management system **defined in the Chilean Water Code (Congreso Nacional de Chile, 2022). This system is based on Water Use Rights (WURs), the legal entitlements that define the amount and timing of water access for consumptive uses (e.g., drinking water, irrigation) and non-consumptive uses (e.g., hydroelectricity). A major limitation of this allocation scheme is that it does not account for climate variability and decreasing water availability (Alvarez-Garreton et al., 2023a; Barría et al., 2021a). (…)**

p3, line 72: Groundwater overexploitation is mentioned as an important contributor to water stress, but it is not fully addressed in the paper. Consider including it in the discussion or conclusion.

We have addressed this topic in our response to the referee 1 (please see R3 and R4 to RC1).

p4, Table 1, line 96: "irrigation fraction (IV) fraction"; the word "fraction" appears twice.

Corrected.

p6, line 162: "To ensure accurate seasonal representation, (...)"; does this also include important weather phenomena like El Niño?

This sentence states that the method is applied by accounting for the probability distribution of variables separately for each month. This ensures that the mean, percentiles, extremes, etc., are correctly represented according to the seasonal cycle in these parameters. The spread in variables' distribution reflects the actual variability, so the manifestation of ENSO in Chile is inherently accounted for—but mostly at the interannual scale, rather than the seasonal cycle.

p6, line 164: TN should be TN (in subscript).

Corrected.

p7, line 183: "account" should be "accounts".

Corrected.

p7, line 193: "Long-term land-use dynamics is (...)"; should be "are".

Corrected.

p7, line 193-194: "many other factors"; could the authors give an example?

We do not find the phrase "many other factors" in these lines, but rather the sentence: "Long-term land-use dynamics is a key factor to consider when estimating changes in water use over time and for many other applications." An accurate dataset of historical land use allows for studying a wide range of topics related to land-use changes, including their interactions with climate, socioeconomic drivers, historical processes, etc.

p9, line 245: "is" should be "are".

Corrected.

p9, line 247: Subscript "IR" unclear (it looks like eIR now).

It seems correct in our records.

p9, line 250: UNC,LU should be UNC,LU (in subscript).

Corrected.

p10, line 286: "regions in the globe" should be "regions of the globe"

Corrected.

p12, Figure 2: the color used to indicate a ΔP of +25 and -25 mm/yr are too similar. This problem can be solved by making the values above 0 greener (rather than yellow).

Yes, the color palette is predefined, but we will try to modify it to improve clarity.

p12, line 311: the authors refer to Fig. 2 but this should be Fig. 3.

We actually refer to the observed changes depicted in Figure 2. We will change that sentence to make it more clearly related to the figure:

> *Besides geographical differences, Chile's hydroclimate exhibits significant temporal variability.* **Over** *the long term, local precipitation records show a clear downward trend***, as shown by the changes in mean annual precipitation between the periods 1960-1990 and 1990-2020** *across much of the country (Fig. 2).*

p13, line 311-313: "The spatially distributed CR2MET dataset shows a precipitation decline consistent with observations, a match that matters given the subsequent use of this dataset for basin-scale assessments." Is this statement based on a statistical test or purely on visual analysis? The authors say that it matters that the observations match the CR2MET data, but this statement would be stronger if they could show that these datasets show significant correlation.

We argue that, in addition to the day-to-day or year-to-year covariance with observations—validated in CR2MET-related papers and documents—the figure shows consistent behavior regarding long-term trends. In our opinion, this does not require further statistical testing. Please see our response R9 to RC1, where the evaluation of the CR2MET data is discussed.

p13, line 319-328: Could the Andes Cordillera also explain discrepancies between this study and others, given the hydroclimatic effects noted earlier (p2, lines 49-50)?

Yes, some local effects related to the Andes barrier on the westerly flow (not fully captured by ESMs), in addition to other conditions that are systematically biased in the models (e.g., sea surface temperature in southeaster Pacific), may be affecting climate projections for Chile. However, these effects have not yet been clearly constrained by observations.

p15, line 372 and 373: "litters" should be "litres" or "liters".

Corrected.

p23, line 524: "water-demand" should be "water demand" to match other occurrences of this term in the document.

Corrected.

p24, Table 2: in the first scenario description, "(2020-2020)" should be "(2000-2020)"

Corrected.

p25, line 568: "near 30% reduction" should be "nearly a 30% reduction".

Corrected.

p25, line 586-591: the conclusion could be slightly improved by concluding with a strong final statement, e.g. something along the lines of "Achieving long-term water security in Chile will require not only better governance and infrastructure but also a commitment to sustainable water use practices across all sectors. This study provides critical insights into how such goals can be realized."

Thank you for the suggestion. We will consider adding something along these lines in the revised manuscript. Please see the related comment on this section and the proposed changes in our responses to Referee 1.

---

## Author Response (AR1)

**EGUSPHERE-2024-2695**

**Responses to referee comments**

This document includes responses (in blue) to comments from both Referee 1 and Referee 2. Suggested changes in the revised manuscript are shown in *italics* and marked in *bold* when new text is added. All references not already cited in the manuscript are listed at the end.

Juan P. Boisier, on behalf of all co-authors.

**RC1 (Caitlyn Hall)**

**Overall Review**

This paper tackles a critical issue: increasing water stress in Chile and the interplay between climate change and human water use. The integration of novel datasets, long-term historical trends, and future projections offers valuable insight into how water availability, land use, and consumption patterns have shaped the country's current water security challenges. The study is thorough, data-driven, and policy-relevant.

However, key methodological choices need more clarity. The assumptions behind the Water Stress Index (WSI) and water balance model should be explicitly justified, particularly the exclusion of groundwater from availability estimates. Additionally, the manuscript should address uncertainties—both in historical datasets and climate model projections—more directly. The discussion on policy implications is strong but could be sharpened with concrete recommendations for decision-makers.

Despite these areas for improvement, this paper makes a strong contribution to understanding Chile's water security trajectory. Addressing the above concerns will increase its impact and make the findings more actionable for policymakers.

**Strengths**

- 1. Novel, Large-Scale Data Integration: The study compiles and analyzes long-term datasets on water availability, land use, and water use—essential for understanding historical trends and future risks.
- 2. Robust Temporal Scope: By covering water stress from the mid-20th century through the 21st century, the paper provides a long-term perspective often missing in regional water studies.
- 3. Future Scenario Analysis: The evaluation of different climate and socioeconomic pathways strengthens the study's relevance to both researchers and policymakers.
- 4. Clear Geographic Insights: The basin-level approach highlights regional disparities in water stress, making the findings useful for localized decision-making.
- 5. Policy Relevance: The study effectively connects scientific analysis with policy discussions, emphasizing the need for quantitative water security targets in Chile.

We appreciate these highlights on the strengths of our paper, as well as the constructive comments on its weaknesses, which we acknowledge and address below.

**Areas for Improvement**

1. Clarify Methodological Assumptions – The water balance model assumes that water availability equals precipitation minus evapotranspiration, without considering groundwater withdrawals. This may not reflect reality in highly managed basins. The authors should discuss how this limitation affects their results. Similarly, the rationale for using a WSI threshold of 40% as a stress indicator should be explicitly justified.

R1: This point is addressed below, but we would like to clarify a potential misunderstanding of our approach. The datasets used in this paper represent water use estimates, on the one hand, and water availability under near-natural conditions, on the other, which is necessary to properly close the water budget (see Eq. [3]). While we do not account for surface-to-groundwater extraction ratios, our estimates account for total water use independently if actual withdrawals come from groundwater —the predominantly case in arid and semi-arid watersheds— or surface sources. This point, along with the rationale for using a WSI threshold, is further developed in our responses to specific comments.

2. Address Data Uncertainties – The study relies on multiple datasets, some of which have inherent uncertainties due to data gaps, resolution limitations, or modeling assumptions. A clear discussion of these limitations—especially regarding water use estimates—will strengthen the credibility of the findings.

R2: Please refer to our response regarding data uncertainties in the specific comments below.

3. Expand Discussion of Groundwater – Groundwater overuse is a key driver of water stress in Chile, yet it receives limited attention in this study. Incorporating groundwater depletion trends or discussing how aquifer withdrawals might be masking true surface water stress would provide a more complete picture.

R3: We agree on the importance of groundwater overuse, which was the focus of another paper by our group (Alvarez-Garreton et al., 2024). The primary focus of the current paper is on the causes of overuse, specifically examining the long-term role of water use and climate variability in driving water stress. In the revised version, we will include a more in-depth discussion of groundwater depletion, integrating insights from this and previous studies. Specifically, we will add new paragraphs to the revised conclusions and recommendations (see R4 below).

4. Sharpen Policy Recommendations – The paper makes a strong case for integrating WSI into policy but stops short of concrete recommendations. What specific actions should be taken? How should water rights, governance structures, or infrastructure investments be adjusted in response to these findings? Explicit next steps will increase the paper's real-world impact.

R4: We appreciate this comment. We will expand on this in the revised manuscript, as follows:

**Revised Section 6:**

(...) In our opinion, metrics such as the water stress index assessed here are necessary to establish water security goals and monitor the efficacy of potential actions to achieve those goals. The advantage of adopting the WSI is that stress levels can be associated to water scarcity and environmental risks (e.g., Falkenmark et al., 2007; Rockström et al., 2014; Oki and Kanae, 2006) and thus translated into specific water security goals. Of course, a target based solely on WSI is insufficient, as it does not capture the full complexity of factors involved in water security,

such as water quality and accessibility, depletion of groundwater resources, governance and integrated water resources management, among others. However, it offers a straightforward quantification that can be complemented with other metrics.

Revised Section 7 (Conclusions and recommendations):

The following point will be included in the revised conclusions:

4. Renewable water availability supports ecosystems and society for long-term functioning. Natural water reserves, such as aquifers, snow, and glaciers, as well as artificial reserves like reservoirs, complement water availability, mainly by providing temporal regulation. These storage systems enable access to greater water volumes, though this is limited by surface recharge rates. Given this limitation, water uses that approach or exceed renewable water availability will not be sustainable over time, whether the water is accessed through groundwater, surface water, or reservoirs.

The following text will be included in the revised recommendations:

**Based on the evidence provided, we recommend** decision-makers and water users to acknowledge that Chilean regulations have permitted and continue to allow consumptive water uses that exceed recognized sustainability levels in the central-northern regions of the country. **Based on this understanding, measures** can be evaluated and agreed upon to mitigate the impact of high water consumption activities. These actions should be implemented alongside effective adaptation strategies to address current and future climate trends, particularly concerning the impacts on the Andes Cordillera.

*(...)*

The assessed pathways of water use and alternative sources of water availability to achieve a water security target under climate change scenarios, exemplified here by targeting a WSI < 40% by 2050, provide valuable insights for adaptation plans currently being developed. While the determination of basin-specific thresholds for WSI and the strategies to meet them fall within the domain of public policy, the adoption of this index would facilitate the discussion on the main approaches to achieving the target: reducing water use or increasing availability through alternative sources. We recommend that current adaptation plans define goals based on quantifiable indices—such as the WSI— complemented by others that capture additional dimensions of water security (e.g., water access and equality, depletion of groundwater table, ecological flow requirements, among others). Having a comprehensive set of water security indices is essential for evaluating adaptation strategies, however, a key challenge lies in making political decisions to establish goals for these indices and determining the necessary changes and associated costs to meet them.

**Introduction**

The introduction provides strong context for the study, linking water security to climate change and socio-economic factors.

Page 2, Line 35-40 – "A basin is deemed to have high water stress when the Water Stress Index (WSI) – the ratio of water use to availability– exceeds 40% over the medium term (5 to 10 years)."

The definition of WSI is useful, but the rationale for using 40% as a threshold should be explicitly stated with references to support this classification.

R5: We have addressed this comment and proposed modifications in the revised manuscript in our responses to RC2 (please see R2 to RC2)

Page 2, Line 45-50 — "However, obtaining accurate information can be challenging due to the poor quality and limited availability of data and ground observations in some regions."

It would be helpful to discuss how the authors address these data gaps in their study. Are there alternative datasets or validation techniques used to mitigate uncertainties?

R6: To merge a myriad of sources and scattered data into a homogenized (in time and space) dataset of water use, accounting for the historical evolution, was the main action taken to address these data gaps. In our opinion, this is the main innovation of this work, allowing for a straightforward analysis of water stress. No comparable datasets exist for water use in Chile, making uncertainty assessment challenging. However, for near-present-day conditions, some estimates are available—one of which served as a methodological benchmark in the development of CR2WU (DGA, 2017). We acknowledge that a more thorough comparison with these datasets is needed to better quantify uncertainties. Therefore, in the revised version, we will include an appendix with a table summarizing national water use totals by major sectors and comparing them with estimates from other sources (see below). We note that this comparison is limited due to mismatches in the sectors included across the different datasets, as well as the specific periods they cover.

| Source >              | FAO (2021)                                                                                                                                                                    | DGA (2017)                                                                                                                                                                                              | F. Chile (2018)                                                                                                                                                                                                                                                                                                                                                                                                                                        | This study                                                                                                                                                                                                                                                                                                                                                                                                                                                                                                                                                                  |
|-----------------------|-------------------------------------------------------------------------------------------------------------------------------------------------------------------------------|---------------------------------------------------------------------------------------------------------------------------------------------------------------------------------------------------------|--------------------------------------------------------------------------------------------------------------------------------------------------------------------------------------------------------------------------------------------------------------------------------------------------------------------------------------------------------------------------------------------------------------------------------------------------------|-----------------------------------------------------------------------------------------------------------------------------------------------------------------------------------------------------------------------------------------------------------------------------------------------------------------------------------------------------------------------------------------------------------------------------------------------------------------------------------------------------------------------------------------------------------------------------|
| Ref. period or year > | 2020                                                                                                                                                                          | 2015                                                                                                                                                                                                    | ~2016-2017                                                                                                                                                                                                                                                                                                                                                                                                                                             | 2010-2020                                                                                                                                                                                                                                                                                                                                                                                                                                                                                                                                                                   |
| LULUCF (irrigated)    |                                                                                                                                                                               | 7.9                                                                                                                                                                                                     | 5.9                                                                                                                                                                                                                                                                                                                                                                                                                                                    | 8.9                                                                                                                                                                                                                                                                                                                                                                                                                                                                                                                                                                         |
| LULUCF (rainfed)      |                                                                                                                                                                               |                                                                                                                                                                                                         | 13.8                                                                                                                                                                                                                                                                                                                                                                                                                                                   | 2.5                                                                                                                                                                                                                                                                                                                                                                                                                                                                                                                                                                         |
| Drinking Water        | 1.3                                                                                                                                                                           | 1.3                                                                                                                                                                                                     | 1.7                                                                                                                                                                                                                                                                                                                                                                                                                                                    | 1.7                                                                                                                                                                                                                                                                                                                                                                                                                                                                                                                                                                         |
| Energy                | 0.1                                                                                                                                                                           | 0.5                                                                                                                                                                                                     | 0.04                                                                                                                                                                                                                                                                                                                                                                                                                                                   | 1.1                                                                                                                                                                                                                                                                                                                                                                                                                                                                                                                                                                         |
| Manufac. Industry     | 1.7                                                                                                                                                                           | 0.7                                                                                                                                                                                                     | 0.04                                                                                                                                                                                                                                                                                                                                                                                                                                                   | 0.8                                                                                                                                                                                                                                                                                                                                                                                                                                                                                                                                                                         |
| Mining                |                                                                                                                                                                               | 0.4                                                                                                                                                                                                     | 0.3                                                                                                                                                                                                                                                                                                                                                                                                                                                    | 0.5                                                                                                                                                                                                                                                                                                                                                                                                                                                                                                                                                                         |
| Livestock             | 0.1                                                                                                                                                                           | 0.1                                                                                                                                                                                                     | 0.1                                                                                                                                                                                                                                                                                                                                                                                                                                                    | 0.1                                                                                                                                                                                                                                                                                                                                                                                                                                                                                                                                                                         |
| Total                 |                                                                                                                                                                               |                                                                                                                                                                                                         | 22                                                                                                                                                                                                                                                                                                                                                                                                                                                     | 16                                                                                                                                                                                                                                                                                                                                                                                                                                                                                                                                                                          |
| Energy                |                                                                                                                                                                               | 66                                                                                                                                                                                                      | 42                                                                                                                                                                                                                                                                                                                                                                                                                                                     | 88                                                                                                                                                                                                                                                                                                                                                                                                                                                                                                                                                                          |
| LULUCF                |                                                                                                                                                                               |                                                                                                                                                                                                         | 6.9                                                                                                                                                                                                                                                                                                                                                                                                                                                    | 2.7                                                                                                                                                                                                                                                                                                                                                                                                                                                                                                                                                                         |
| Other                 |                                                                                                                                                                               |                                                                                                                                                                                                         | 0.4                                                                                                                                                                                                                                                                                                                                                                                                                                                    |                                                                                                                                                                                                                                                                                                                                                                                                                                                                                                                                                                             |
| Total                 |                                                                                                                                                                               |                                                                                                                                                                                                         | 49                                                                                                                                                                                                                                                                                                                                                                                                                                                     | 91                                                                                                                                                                                                                                                                                                                                                                                                                                                                                                                                                                          |
|                       |                                                                                                                                                                               |                                                                                                                                                                                                         | 71                                                                                                                                                                                                                                                                                                                                                                                                                                                     | 106                                                                                                                                                                                                                                                                                                                                                                                                                                                                                                                                                                         |
| LULUCF                | 29.4                                                                                                                                                                          | 7.9°                                                                                                                                                                                                    | 13                                                                                                                                                                                                                                                                                                                                                                                                                                                     | 12                                                                                                                                                                                                                                                                                                                                                                                                                                                                                                                                                                          |
| Other                 |                                                                                                                                                                               | 68                                                                                                                                                                                                      | 44                                                                                                                                                                                                                                                                                                                                                                                                                                                     | 92                                                                                                                                                                                                                                                                                                                                                                                                                                                                                                                                                                          |
| Total                 | 35b                                                                                                                                                                           | 76                                                                                                                                                                                                      | 57                                                                                                                                                                                                                                                                                                                                                                                                                                                     | 104                                                                                                                                                                                                                                                                                                                                                                                                                                                                                                                                                                         |
|                       | Ref. period or year >  LULUCF (irrigated)  LULUCF (rainfed)  Drinking Water  Energy  Manufac. Industry  Mining  Livestock  Total  Energy  LULUCF  Other  Total  LULUCF  Other | Ref. period or year > 2020  LULUCF (irrigated)  LULUCF (rainfed)  Drinking Water 1.3  Energy 0.1  Manufac. Industry 1.7  Mining  Livestock 0.1  Total  Energy  LULUCF  Other  Total  LULUCF 29.4  Other | Ref. period or year >       2020       2015         LULUCF (irrigated)       7.9         LULUCF (rainfed)       1.3       1.3         Drinking Water       1.3       1.3         Energy       0.1       0.5         Manufac. Industry       1.7       0.7         Mining       0.4       0.1         Livestock       0.1       0.1         Total       66       0.1         LULUCF       0.1       0.1         Other       7.9°         Other       68 | Ref. period or year >       2020       2015       ~2016-2017         LULUCF (irrigated)       7.9       5.9         LULUCF (rainfed)       13.8       1.3       1.7         Energy       0.1       0.5       0.04         Manufac. Industry       1.7       0.7       0.04         Mining       0.4       0.3         Livestock       0.1       0.1       0.1         Total       22         Energy       66       42         LULUCF       6.9         Other       0.4         Total       49         LULUCF       29.4       7.9°       13         Other       68       44 |

<sup>a Blue water withdrawals (exclude rainfed agriculture).

PAO-AQUASTAT's Total Freshwater Withdrawal. It does not account for withdrawals in hydropower generation.

 $^{\circ}\,\text{Does}$  not account for non-consumptive water use in agriculture.

Regarding precipitation please see response to comment on CR2MET below (R9).

**Methods**

Page 6, Line 90-95 — Water availability is derived from a simple water balance approach. Consider discussing whether this approach adequately captures the influence of groundwater withdrawals.

R7: In our methodology, water availability is somewhat independent of water use (though not entirely, as both are influenced by climate variability, but this distinction is not crucial for this point). As defined here, water availability corresponds to Renewable Freshwater Resources (RFWR), according to several references (e.g., Vorosmarty et al., 2000; Alcamo et al., 2003a; Oki and Kanae, 2006; Wada et al, 2011a). Although not explicitly quantified, renewable GW resources are part of RFWR through natural recharges. Whether the use of renewable groundwater is sustainable depends on the balance between RFWR and consumptive water use. That is where the usefulness of the WSI used in this study lies. There is a non-renewable GW component that, by definition, is not included in RFWR.

Please follow the comment from referee 2 and response on this topic (R3 to RC2). We have also enriched the text to clarify the point.

Revised Section 2.3: Hydrological balance and water availability

Water availability (A) is considered here as a naturalized runoff, computed as the remaining flow from precipitation (P) and evapotranspiration (ET), without considering local disturbances (Eq. 2). This definition aligns with Renewable Freshwater Resources (RFWR), according to several references (e.g., Vorosmarty et al., 2000; Alcamo et al., 2003; Oki and Kanae, 2006; Wada et al, 2011a; Kuzma et al., 2023). Renewable water availability includes natural groundwater recharge and thus renewable groundwater resources, however, there is a non-renewable groundwater component that, by definition, is not included in A. It should be noted that groundwater can supply water beyond the groundwater recharge if storage depletion occurs, as reported by recent studies in central Chile (Jódar et al., 2023; Alvarez-Garreton et al., 2024; Taucare et al., 2024).

Are there validation steps comparing modeled ET against observed or remote-sensing-based ET datasets?

R8: Evapotranspiration (ET) is a challenging variable to measure and, consequently, to constrain at larger scales using observational data. Most spatially distributed ET datasets, particularly those derived from satellite retrievals, have large uncertainties due to the indirect nature of their estimates (e.g., Lobos et al., 2020; Melo et al., 2021). In our case, the modeled ET is validated using watershed-scale water balances, with observed river streamflow as a benchmark (Appendix B). Yet, this approach relies on a key assumption: that precipitation data—being the third variable in the water balance—is reasonably accurate (see the next response regarding the CR2MET dataset). Nevertheless, given the large sample of watersheds included in our analysis, this method provides greater confidence in our estimates. Additionally, we incorporated the state-of-the-art reanalysis dataset ERA5-Land as an independent benchmark for water fluxes (P & ET), showing poorer performance than our estimates. While a comprehensive intercomparison with multiple ET products was beyond the scope of this study, it remains a key priority for future improvements to our data products.

Page 7, Line 115-120 – Since CR2MET is a core dataset in the study, a brief discussion of its validation and potential biases would improve confidence in the results.

R9: We agree on this point. Accordingly, we will add the following two paragraphs in the revised manuscript, section 2.1:

A detailed description and validation of CR2MET are provided in DGA (2022a) and references therein. Because CR2MET's statistical models are calibrated using a large observational network in Chile, validation against local meteorological records generally shows low bias and strong covariability in precipitation and temperature compared to other products, particularly in central-southern Chile (DGA, 2018, 2019). Cross-validation analyses further confirm the product's predictive reliability (e.g., DGA, 2018). On a daily time scale, performance tends to decrease in the southern and northern regions of the country, mainly due to weaker covariability between ERA5 and station data in these areas. On monthly or annual time scales, CR2MET generally shows good metrics across the country, making it a useful reference for various applications (e.g., Anderson et al., 2021; Carrasco-Escaff et al., 2023).

The main challenge for spatially distributed meteorological products is their ability to extrapolate and provide accurate estimates in data-sparse regions, particularly across the extratropical Andes in the case of Chile. Hydrological simulations, driven with datasets such as CR2MET and compared against observed river discharges, offer an objective way to assess their performance in areas without meteorological monitoring (e.g., watershed headwaters). This type of validation and the intercomparison with other meteorological datasets consistently demonstrate strong performance (Kling–Gupta Efficiency values typically above 0.6; Baez-Villanueva et al., 2021; DGA, 2022a), with CR2MET outperforming other available products (e.g., Baez-Villanueva et al., 2021). Hydrological balances also allow the identification of systematic biases in precipitation totals within the basins. In this regard, we note that precipitation products—including CR2MET—likely underestimate precipitation in Chile's wettest watersheds, particularly in the southern Andes and western Patagonia (Alvarez-Garreton et al., 2018).

Page 10, Line 175-180 — "Water availability (A) is considered here as a naturalized runoff, that is, the remaining flow from precipitation (P) and evapotranspiration (ET), without considering local disturbances."

The assumption of no local disturbances might oversimplify real-world hydrological conditions, especially in basins with significant groundwater-surface water interactions. Consider discussing the implications of this assumption.

R10: As noted in R7, water availability is defined and considered throughout the paper as RFWR, which establishes the basis for our experimental design and numerical simulations. This corresponds to a methodological framework; it not aims to simplify watershed hydrology but rather to separate components. As in other similar studies (e.g., Wada et al., 2011b; Kuzma et al., 2023), RFWR represents potential water resources and should not include withdrawals, so they can later be accurately contrasted with water uses. Hence, we define water availability as the flow resulting from precipitation and evapotranspiration, without accounting for extra ET losses due to land use or other activities, and regardless of whether the remaining flow manifests as surface runoff, groundwater recharge, or other processes. We will revise this paragraph to clarify this definition (see R7).

**Results & Discussion**

Page 15, Line 285-290 – "On average across continental Chile, the mean annual rates of P and ETN are estimated at 1200 mm and 430 mm, respectively, which leads to a surface water availability of approximately 770 mm yr-1."

How do these values compare to previous studies? Providing some context (like from previous studies, especially linking to the "so what" of this article) would strengthen the reliability of these estimates.

R11: Although these national averages may not be particularly relevant for specific basin-scale analyses —especially in the case of Chile, given its latitudinal extent and diverse climates—they are provided here for context and as reference values (e.g., for comparisons with other regions worldwide). Few studies indicate national-scale water availability estimates for Chile. The recent national water balance update, although not reported explicitly in DGA (2022a), led to very similar values. This is expected, as that study uses a methodology comparable to ours, including CR2MET as the atmospheric driver. By contrast, the FAO-AQUASTAT dataset (FAO, 2021) reports a value of RFWR that is substantially higher (by approximately 35%) than our estimate. However, this difference is difficult to interpret due to contrasted methodologies, time periods considered, and the spatial domain used to compute national averages (in particular, how Patagonian islands and fjords are accounted for).

We have revised that paragraph in the manuscript's section 3 to incorporate these comparisons:

On average across continental Chile, the mean annual rates of P and ETN are estimated at 1200 mm and 430 mm, respectively, which leads to a surface water availability of approximately 770 mm  $yr^{-1}$  (equivalent to a volumetric flux of 680 km3  $yr^{-1}$ ). A similar value is estimated in DGA (2022a), which also uses the CR2MET P dataset, while a higher value is reported by FAO (2021) as Chile's renewable water resources—approximately 1050 mm  $yr^{-1}$ .

Beyond these differences, average freshwater availability in Chile remains high compared to other countries (the global mean over continental areas is, close to 300 mm yr-1; Oki and Kanae, 2006). However, the reality varies greatly ...

Page 18, Line 365-370 — "Agriculture also has a non-consumptive water use component, as some irrigation water returns to the system through infiltration and percolation."

The study could benefit from quantifying this return flow, if possible, to better illustrate the net impact of agricultural water use.

R12: This return flow was estimated (see Eq. 6) and is included in Figure 4 (see non-consumptive uses in panel a). To make this more explicit, in the revised manuscript we will include that component in the new table in the Appendix (see R6), and we will also mention it in the main text and add a reference to the figure following that sentence (section 4):

Agriculture also has a non-consumptive water use component, as some irrigation water returns to the system through infiltration and percolation. For the near-present period, the return flow is estimated at about 2.7 km³ yr¹, corresponding to 23% of the total water withdrawn estimated for irrigated agriculture (Figure 4a and Table A1). This return flow, as a proportion of total irrigation, reflects irrigation efficiency, which tends to decrease with adoption of more advanced technologies and improved management practices.

Page 19, Line 385-390 – The discussion on irrigation efficiency should explicitly address the paradox in which increased efficiency can lead to higher overall water use.

R13: In a revised manuscript we will change that part as follows:

Part of these changes can be attributed to the irrigation efficiency paradox, which refers to cases where improvements in irrigation efficiency reduce non-consumptive water use but do not result in actual water savings at the basin scale. Instead, increased efficiency enables the irrigation of a larger cropping area or the cultivation of more water-intensive crops, ultimately leading to higher consumptive water use—especially when total water extractions are not regulated (Grafton et al., 2018).

Page 21, Line 425-430 – The manuscript states that "water demands approach or even exceed available surface water in a number of basins today." If groundwater use data are available, incorporating them into the discussion would strengthen the argument about unsustainable water use.

R14: We largely agree that incorporating groundwater extraction data would strengthen the discussion. However, this information is incomplete or unavailable for most regions and sectors. Given these limitations, our dataset provides water uses without explicitly accounting for surface-to-groundwater use partitioning—a significant and valuable addition for future updates. Estimations of groundwater storages and extractions are indeed subject to considerable uncertainty due to the lack of independent datasets like the one provided here.

Bookkeeping data exist regarding the allocation of groundwater and surface water use rights (WURs) (e.g., Alvarez-Garreton et al., 2023a), however, the allocated fluxes (fixed in time) do not represent actual water uses since they depend on water access and availability.

Although our current dataset does not include the portion of water uses that comes from groundwater sources, a previous study from our group reported unsustainable groundwater use (i.e., sustained decline of water table) in basins where total water uses approach the RFWR (Alvarez-Garreton et al., 2024). In the revised conclusions we address this more explicit (see R4).

Page 24, Line 495-500 – "Given global climate scenarios, there are only two ways to alleviate water stress in a basin: reducing consumptive water use or increasing water availability through alternative sources."

Consider briefly mentioning the role of integrated water resource management (IWRM) and governance improvements as additional pathways to address water stress.

R15: In that sentence we refer specifically to the physical means of alleviating water stress rather than the mechanisms for implementing solutions. While discussing specific governance modes or pathways is beyond the scope of this paper and our expertise, we agree that the manuscript should more clearly outline recommendations for policymakers. Please see R4 for proposed changes related to this point in sections 6 and 7.

**Conclusion & Policy Implications**

Page 26, Line 545-550 – "It should be noted that non-renewable water reserves, such as aquifers that are not in equilibrium or melting glaciers, are not considered as alternative sources to reduce the WSI."

This is an important distinction. However, the paper could discuss the role of groundwater depletion in temporarily masking the full impact of water stress.

R16: We largely agree. This is a key point that was central in a previous study developed by our research team (Alvarez-Garreton et al., 2024). In the revised manuscript, we will emphasize this aspect and include a related recommendation (see R4).

Page 27, Line 560-565 — "Public policy in Chile is aware of the impacts of droughts and the challenges that climate change poses to water security."

The discussion would benefit from more explicit recommendations on how policy can integrate WSI into national adaptation strategies.

R17: We appreciate this comment. We will expand on this in the revised Section 6 and Conclusions/ Recommendations. Please see our proposed text in R4.

**RC2 (anonymous)**

**General comments**

This manuscript quantifies water stress in all river basins in Chile based on a very detailled estimation of water use (U) and an appropriate high-resolution estimation of annual runoff under naturalized conditions (A) as water stress indiex WSI = U/A, for both historic conditions in the period 1960-2020 and for a potential situation in 2035-2065 under changing climate (as estimated from the median change of runoff as projected by 11 global climate models, after bias-correction of GCM climate variable output with local observations) and water use change (as simple linear extrapolation of the trends observed in the period 2000-2020). They find that in the case of low climate mitigation, precipitation at the end of the 21st century may be similarly low as during the extreme drought that occurred in the 2010s. They find that in central-northern basins of Chile, WSI values of more than 40% were reached by 2020 even though water use is regulated by water use rights. The authors suggest that public policy sets maximum WSI values that represent a threshold for water security so that then actions can be taken to not exceed WSI in each basin, taking into account future climate change.

The quantification of water availability (A) is state-of-the-art and innovative regarding spatial resolution and meteorological data used. An innovative and commendable approach taken by the manuscript is to include increased evapotranspiration due to a change in land cover since 1950 based on a detailed estimation of land use changes between 1950 and 2020 as human water use (consumptive use). It would be even better if the additional evapotranspiration caused by the artificial reservoirs would be quantified as part of the human water use U.

Unfortunately, the manuscript has a number of major weaknesses.

We appreciate the comments highlighting both the strengths and innovations of our study, as well as the constructive feedback on its limitations. Below, we provide responses to each point and propose modifications to the revised manuscript to address them.

First, we would like to clarify that evapotranspiration (ET) from both natural and artificial water bodies is accounted for in our water use dataset (CR2WU). Water use from artificial reservoirs is derived from the land cover maps used in our experiments (see Eq. 4) and, therefore, included under the land use sector in the CR2WU dataset.

1) The definition of the water stress index WSI is not presented according to the literature, and it it not mentioned that the definition in the paper is not the same as in the literature. The component U generally refers to water withdrawals (= abstractions= sum of consumptive use and return flows = sum of consumptive and non-consumptive use), and not to consumptive use as seems to be the case in this manuscript (even though the specific definition is never provided, see below my comments regarding the quantification of U). Even in the publication "Alvarez-Garreton, C., Boisier, J. P., Billi, M., Lefort, I., Marinao, R., and Barría, P.: Protecting environmental flows to achieve long-term water security, J Environ Manage, 328, 116914, https://doi.org/10.1016/j.jenvman.2022.116914, 2023a.", it is clearly stated that U are water withdrawals (and that allocated rights WUR are defined as withdrawals and thus are not comparable to consumptive water use supposed to be shown in Figure 5).

R1: We understand the referee's concern and clarify this point below to avoid misunderstandings.

We respectfully disagree with the assertion that the Water Stress Index (WSI) should, according to the literature, be defined exclusively in terms of total water withdrawals. While it is true that many studies

define WSI using total withdrawals, other studies have raised concerns about this approach. As noted by Rijsberman (2006), relying solely on withdrawals can lead to misleading conclusions, particularly in basins with substantial return flows or water reuse. A typical example is basins with large hydropower plants, where water is withdrawn and returned without significant loss. In such cases, total withdrawals may be high, resulting in elevated WSI values—even though actual pressure on the watershed (i.e., consumptive use) is low.

We acknowledge that past studies have been ambiguous on this matter. For instance, widely cited references such as Vörösmarty et al. (2000) and Oki et al. (2006) use the terms "withdrawal" and "water use" interchangeably. Their estimates include irrigation, domestic, and industrial uses based on reports such as World Resources 1998–99 (UN, World Bank, etc.). However, large non-consumptive water uses—such as those related to hydroelectric power generation—are not explicitly included in these estimates. Similarly, the model used by Alcamo et al. (2003) does not account for water withdrawals from hydroelectric plants in their water use calculations. Instead, they emphasize the higher water use in countries reliant on thermal power plants, which involve consumptive use, whereas countries with greater reliance on recycled water tend to reduce net water consumption in power generation. More explicitly, Wada et al. (2011a, 2011b) discuss this issue and assess WSI using net (consumptive) water demand. Other studies (e.g., Munia et al., 2016) include both gross and net water use estimates in order to provide upper and lower bounds for WSI.

Thus, there are different approaches to computing water use and derived metrics such as WSI, as pointed out by Falkenmark et al. (2007), Damkjaer and Taylor (2017) or Liu et al (2017). Additionally, regarding the agricultural sector, most studies consider only blue (withdrawn) water for irrigation. In our opinion, and in agreement with Liu et al. (2022), an increase in ET losses in rainfed agriculture or forest plantations (green water) should also be accounted for in proper water stress estimates. Despite methodological differences, studies using the WSI aim to quantify the relationship between water use—represented as the portion of water that does not remain available for other uses—and water availability. Building on this understanding, studies have linked WSI with water allocation schemes and environmental flow criteria. For example, Smakhtin (2008) argued that the allocation of water within a basin should stop before 'not a single drop is wasted to the sea' but rather when the remaining flow compromises environmental requirements. It makes sense, then, that returned non-consumptive flows are considered as available for fluvial ecosystems and thus not included in WSI computation.

In our reconstruction we estimate consumptive water uses for several sectors, as well as non-consumptive uses for LULUCF and energy sectors. Similar to Wada et al. (2011a), we aim to maintain a water balance rationale in the WSI assessment, meaning we account only for consumptive uses. Our analyses are conducted at the outlets of major basins of Chile, near their discharge to the sea. This location represents the end of the system, where no further extractions can occur and where all return flows from non-consumptive water uses are assumed to have already taken place at upstream locations within the same basin. Following this approach, the net water use of a basin is represented only by the consumptive portion of total withdrawal.

To be precise, we consider consumptive uses from various sectors (LULUCF, manufacturing industry, livestock, mining and energy) in addition to drinking water (DW). Separating consumptive and non-consumptive components of DW is not straightforward. Return flows after domestic or institutional use are difficult to track. In some cases, DW has a significant consumptive component related to irrigation of public and private green areas within urban zones, which is not accounted for under LULUCF component. Wastewater treatment varies widely across regions, as do the quality and discharge points of return

flows. In most coastal areas, wastewater is discharged directly into the sea, meaning it cannot be considered a return flow. Since these factors are unresolved in our DW estimates, we took a conservative approach and included all DW in our WSI calculations. Additional details are provided in R4.

We acknowledge that ambiguities in past WSI calculation and the approach followed in this study should be stated more clearly. Accordingly, we propose the following revisions to the manuscript:

**Section 1 (Introduction):**

(...)

There are various ways to assess water security in a territory, with methodologies varying in complexity based on the factors considered. A common and straightforward approach is to contrast water requirements and renewable freshwater resources estimates at the basin scale. Following this approach, several studies have quantified a Water Stress Index (WSI) as the ratio of water use to availability and linked its values to the risk of experiencing water scarcity problem (e.g., Falkenmark and Lundqvist, 1998; Vörösmarty et al., 2000; Alcamo et al., 2003; Rijsberman, 2005; Oki et al., 2006; Wada et al., 2011a; Damkjaer & Taylor, 2017).

**Section 2.5 (Methods - Water uses reconstruction):**

The CR2WU product was developed to assess historical water demand in Chile. This dataset includes water uses estimates as volumetric fluxes (U) from various sectors for each commune in continental Chile, with annual resolution from 1960 to 2020. The CR2WU dataset includes consumptive uses for all sectors considered, as well as non-consumptive uses for the two sectors with major return flows: hydroelectric generation and irrigated agriculture. The computation involves two distinct methodologies: one for sectors encompassing land use, landuse change, and forestry (LULUCF) and another for other water-consuming sectors.

(...)

In DGA (2017), drinking water was treated separately, with total withdrawals estimated alongside a return flow component, calculated based on sanitary data. However, in Chile, data on wastewater flows and discharge locations—necessary for reconstructing historical return flows—are not publicly available. Due to this limitation, this component was not estimated in our study, and the WSI calculation includes total withdrawals for this sector, whereas for other sectors only the consumptive component is considered (Sect. 2.6). In some cases, this approach is supported by the fact that wastewater from several treatment plants is discharged directly to the sea via submarine outfalls (Gómez et al., 2025), making drinking water withdrawals effectively a fully consumptive use at the basin scale. In other cases, where wastewater is discharged upstream of the basin outlet, water quality considerations provide additional justification for this choice. Indeed, the Chilean law prohibits the reuse of treated greywater in key sectors such as potable water supply, irrigation of fruits and vegetables that grow at ground level, food production, healthcare facilities, and aquaculture (MOP, 2023).

**Section 2.6 (Methods - Computation of water stress):**

The WSI, as the index characterizing water demand-to-availability ratio (Falkenmark and Lundqvist, 1998), depends on how both quantities are defined, potentially leading to different outcomes. In particular, studies using this index differ in whether they consider gross demand (as a proxy for total withdrawals; e.g., Vörösmarty et al., 2000) or net demand (equivalent to consumptive use; e.g., Wada et al., 2011a), while others assess both cases (e.g., Munia et al.,

2016). Using total withdrawals in basins with significant return flows—such as in Chile, where many basins are modified for hydroelectric power generation—results in very high WSI values that are not necessarily representative of actual water stress. For this reason, and to preserve a water balance rationale, only consumptive uses are considered in the WSI calculation in this study. Then, the WSI is defined as the ratio UCO/A, where UCO and A represent total consumptive water use and near-natural water availability, respectively, as described in previous sections. The WSI is computed at muti-annual timescales for hydrological units, specifically over the major watersheds defined by the DGA in the National Inventory of Watersheds (BNA).

As explained in Section 2.5, we considered the total drinking water requirement as consumptive use, acknowledging that in some watersheds return flows from water treatment can be effectively reused. To account for green water use (e.g., Liu et al., 2022), consumptive water use from rainfed agriculture is also included, estimated as the differential ET loss between land use scenarios (see Section 2.5).

Finally, we would like to clarify that in Alvarez-Garreton et al. (2023a), consumptive and non-consumptive water use rights were treated separately. In that paper, a WSI-like index was computed based only on consumptive water use rights, as stated in the abstract: "For each basin, water scarcity risks were assessed based on water stress indices (WSIs, computed as the ratio of withdrawals to water availability), considering two water-use scenarios: (i)  $WSI_{max}$ , where total withdrawals correspond to the maximum consumptive water allowed by the law, i.e., where only the e-flows protected by law remain in the river, and (ii)  $WSI_{alloc}$ , where total withdrawals correspond to the actual allocated consumptive water uses within the basins." Therefore, the statement made by the referee "... it is clearly stated that U are water withdrawals (and that allocated rights WUR are defined as withdrawals and thus are not comparable to consumptive water use supposed to be shown in Figure 5)" does not fully reflect the distinction made in Alvarez-Garreton et al. (2023a).

2) In various locations of the manuscript, it is wrongly stated the a WSI provides a strong quantititative basis for indicating negative impacts on watersheds (e.g. in line 499-500). In lines 39-42, the author write, without citing any publication

"There are various ways to assess water security in a territory, with methodologies varying in complexity based on the factors considered. A common and straightforward approach is to contrast water use and water availability estimates at the basin scale. A basin is deemed to have high water stress when the Water Stress Index (WSI) –the ratio of water use to availability—exceeds 40% over the medium term (5 to 10 years)."

The authors need to provide some references, and in general water availability computed over a longer time period, e.g. 30 years (see references for WSI in Alvarez-Garreton et al. 2023a). And the authors should mention that the classification of water stress occuring if WSI>0.4 (with abstractions and not consumptive use as U) is not based on evidence but a rough guess. They should clearly state from the beginning that they aim a formulating a type of WSI that is suitable for Chile or for their study, and what type of water use, water withdrawals or consumptive use they apply for WSI. I suggest that the vague term "water use" is replaced throughout the text by either consumptive use or water withdrawals (or abstractions). The term water consumption should be used only very specifically as refering to water volumes reaching households etc., but maybe is not of interest for the manuscript anyway (water consumption is not equal to consumptive use, which would have to be explained if the term consumption is used in the manuscript).

R2: Our aim was not to present the WSI as a definitive indicator of negative impacts on watersheds, but rather as a simple and widely used approach to assess water stress. This is consistent with the studies cited in the same paragraph highlighted by the reviewer—Falkenmark and Lundqvist (1998), Vörösmarty et al. (2000), and Oki et al. (2006)—as well as many others (e.g., Alcamo et al., 2003; Wada et al., 2011a, 2011b; Kusma et al., 2023). As noted in the conclusion section, we recognize that the index does not capture several important local factors influencing water stress, such as water quality, accessibility, and governance.

Despite these limitations, WSI has been extensively applied to assess basin-level water stress, with the 40% threshold commonly cited as a reference point above which a basin is considered to be under stress. This threshold has been used in relation to per capita water availability and environmental flow requirements (e.g., Falkenmark and Lundqvist, 1998; Alcamo et al., 2003; Wada et al., 2013; Falkenmark et al. (2007) suggests that WSI values between 40% and 70% indicate high water stress, while values above 70% correspond to the overuse of water that should ideally be reserved for environmental flows. Pastor et al. (2014) reviewed several methods for estimating ecological flow requirements. On average, they recommend preserving 34% to 56% of the mean annual flow (see Table 5 in Pastor et al., 2014), which corresponds to a maximum WSI of 44% to 66%. These values are consistent with the 40% threshold commonly used to signal potential risk to fluvial ecosystems, and align with the 70% threshold often associated with extreme water stress, where the protection of environmental flows is typically not ensured.

We fully agree that the WSI has its limitations and that the water stress classes derived from it involve a degree of uncertainty (e.g., Rijsberman, 2006; Damkjaer and Taylor, 2017). However, in our view, these thresholds represent more than a "rough guess". WSI offers an objective metric for estimating the degree of water use pressure relative to freshwater availability. We note that one important source of uncertainty in WSI assessments an the meaning of WSI levels stems from the use of total water withdrawals, which can include recycled water and thus overestimate actual pressure. As noted in our response to the previous comment (R1), this is the reason why we rely on consumptive water use in our WSI calculation. Hence, we acknowledge that a WSI > 40% may not represent the same level of risk across all basins, as water scarcity is influenced by multiple factors, including interannual variability, population, infrastructure, and governance. Still, based in previous and our research, we defend that the 40% and other WSI levels serves as a reference to assess water stress.

We understand that there was confusion regarding the calculation of water uses in our study, and we hope that our responses to related comments (see R1 and R4) have helped clarify this issue. In addition to suggested changes described in R1, we will modify the text to clarify WSI limitations and adopted thresholds in the revised manuscript.

Section 2.6 (Methods - Computation of water stress):

(...)

Regarding water stress classification based on the WSI, many studies have adopted a reference threshold of 40%, above which a basin is considered water-stressed (e.g., Falkenmark and Lundqvist, 1998; Vörösmarty et al., 2000; Alcamo et al., 2003; Oki and Kanae, 2006; Wada et al., 2011; Munia et al., 2016; Falkenmark et al., 2007; Kuzma et al., 2023). WSI levels have been linked to the concept of water crowding, with values between 40% and 70% typically indicating high water stress, and values above 70% representing extreme stress—

when water use begins to affect volumes that should be reserved for environmental flows (see Falkenmark et al., 2007, and references therein).

From an ecological flow perspective, Pastor et al. (2014) reviewed several methods for estimating environmental flow requirements. On average, these methods recommend preserving between 34% and 56% of the mean annual flow, which corresponds to a maximum WSI of approximately 44% to 66%. Similar ecological flow requirements have been estimated for watersheds in Chile using different methodologies (Alvarez-Garreton et al., 2022). These findings support the idea that WSI levels above 40% may pose risks to fluvial ecosystems and reinforce the 70% threshold for extreme stress, as ecological flow protection is generally not ensured beyond this point.

Section 6 (How to advance toward water security goals?)

(...) However, some of these instruments, explicitly oriented towards water resources (MOP and DGA, 2024), have not yet defined quantitative metrics of water security. In our opinion, metrics such as the water stress index assessed here are necessary to establish water security goals and monitor the efficacy of potential actions to achieve those goals. The advantage of adopting the WSI, is that stress levels can be associated with water scarcity and environmental risks (e.g., Falkenmark et al., 2007) and thus translated into specific water security targets. Of course, a target based solely on the WSI is insufficient, as it does not capture the full complexity of water security, which also includes factors such as water quality and accessibility, groundwater depletion, governance, and integrated water resources management, among others. Nevertheless, the WSI provides a clear and straightforward metric that can be effectively complemented by additional indicators.

**Section 7 (Conclusions)**

(...)

The assessed pathways of water use and alternative sources of water availability to achieve a water security target under climate change scenarios, exemplified here by targeting a WSI < 40% by 2050, provide valuable insights for adaptation plans currently under development. While setting basin-specific WSI thresholds and strategies to meet them fall within the scope of public policy, adopting this index can facilitate discussions on the main approaches to achieving the target: reducing water use or increasing availability through alternative sources. We recommend that current adaptation plans define goals based on measures of water balance at basin scale —such as the WSI— complemented by other metrics that capture additional dimensions of water security (e.g., water access and equality, depletion of groundwater table, ecological flow requirements, among others). Having a comprehensive set of water security indices is essential for evaluating adaptation strategies, however, a key challenge lies in making political decisions to establish goals for these indices and determining the necessary changes and associated costs to meet them.

3) Usage of the term "surface water avaliability" needs to be corrected to avoid misunderstandings: A, as the difference between precipitation and actual evapotranspiration, includes groundwater recharge and thus renewable groundwater resources and the availability of renewable groundwater. I suggest the manuscript quantifies total availability of renewable water resources, while groundwater can supply water beyond the groundwater recharge if groundwater storage depletion occurs (with constantly falling

groundwater tables). In this context, I suggest explaining shortly the situation in Chile regarding the source of water abstractions (groundwater or surface water) and any occuring groundwater depletion.

R3: As the reviewer correctly points out, water availability in our study aligns with the definition of Renewable Freshwater Resources (RFWR) used in many studies, which includes both surface runoff and the portion that contributes to groundwater recharge. To avoid confusion, we will clarify this in the revised manuscript and explicitly state that what we refer to as "surface water availability" represents RFWR.

We also agree with the reviewer's observation that groundwater can supply water beyond the rate of recharge when groundwater storage is being depleted. This unsustainable groundwater use is indeed occurring in Central Chile, as documented in recent studies (e.g., Taucare et al., Jódar et al., Alvarez-Garreton et al.).

In addition to changes highlighted in R1, the revised manuscript will include the following editions:

**Section 1 (Intro)**

(...) More consistently, recent research has shown that, despite lower **renewable freshwater resources**, increasing water supply has been sustained by the overexploitation of groundwater, leading to continuous depletion of the water table in the major basins of central Chile (Jódar et al., 2023; Alvarez-Garreton et al., 2024; Taucare et al., 2024).

**Section 2.3 (Hydrological balance and water availability)**

Water availability (A) is considered here as a naturalized runoff, computed as the remaining flow from precipitation (P) and evapotranspiration (ET), without considering local disturbances (Eq. 2). This definition aligns with Renewable Freshwater Resources (RFWR), according to several references (e.g., Vorosmarty et al., 2000; Falkenmark & Rockström, 2004; Oki and Kanae, 2006). Renewable water availability includes natural groundwater recharge and thus renewable groundwater resources, however, there is a non-renewable groundwater component that, by definition, is not included in A. It should be noted that groundwater can supply water beyond the groundwater recharge if storage depletion occurs, as reported by recent studies in central Chile (Jódar et al., 2023; Alvarez-Garreton et al., 2024; Taucare et al., 2024).

4) I have my doubts about the consistent handling of diverse types of water uses in the "CR2WU water use reconstruction", section 2.5. On the one hand, consumptive water use for irrigation and land cover change is computed as well as non-consumptive use for irrigation. On the other hand, it appears that water withdrawals are quantfied for the other sectors (which is not explained as the term "consumption" is used starting in line 252), but consumptive uses (i.e. the part of the abstracted water that evapotranpirates during use) are not. Thus the statement in line 263 "The CR2WU dataset includes both consumptive and non-consumptive uses from LULUCF and non-LULUCF sectors." and the entry in Table 1 is misleading. And so is the caption in for Figure 4b and the caption and title of Figure 5, where "consumptive use" appears to refer also to the drinking water, energy, manufacturing, mining and livestock water use, while these apparently refer to withdrawal rates (e.g. 145 l per cap and day for drinking water; this amount of water does not evaporate!). Either a convincing rational for mixing two types of water use (abstractions and consumptive use) is provided, or two different estimates of water use (of all sectors) are used alternatively in the WSI computation: 1) consumptive use/A 2) withdrawals/A. Both are interesting and indicate different types of stress.

R4: This comment reflects a misunderstanding of the CR2WU water use reconstruction and its role in the WSI calculation, which has been addressed in R1. Here, two points are reiterated for clarification:

- The CR2WU dataset includes consumptive uses for all sectors considered, as well as nonconsumptive uses for the two major sectors with significant return flows: hydroelectric generation and irrigated agriculture.
- For the reasons outlined in R1, WSI is computed using consumptive water use from all sectors, with LULUCF being the dominant contributor. Drinking water (DW) is included, though it is not separated into its consumptive and non-consumptive components.

For non-LULUCF sectors, consumptive water use was estimated based on the methodology adopted from DGA (2017). As noted in previous responses, our WSI assessment is based on consumptive water uses for all other sectors and total withdrawals for the drinking water sector. That is, withdrawals for drinking water are considered to be consumptive. Aware that this point may be confusing, we make it explicit throughout the manuscript, such in the label of Fig. 4b (Consumptive uses and drinking water). We will clarify this distinction in the revised manuscript.

As noted in R1, the rationale adopted for the drinking water sector is supported by the fact that wastewater from several treatment plants are discharged directly to the sea through submarine pipes (Gómez et al., 2025), making total withdrawals effectively equivalent to consumptive use at the basin scale. For those cases where wastewater is discharged upstream of the basin outlet, water quality considerations further support this rationale. In this aspect, we note the law in Chile states that greywater cannot be reused in key sectors such as potable water supply, irrigation of fruits and vegetables that grow at ground level, food production industries, healthcare facilities, and aquaculture (MOP, 2023). Consequently, in these situations, total withdrawals can be considered unavailable for further uses within the basin.

To ensure clarity, in the revised manuscript we will better clarify the water use components and the considerations for the drinking water sector, as well as provide a more detailed explanation of the assumptions underlying our WSI computations (see R1).

Please note that, following a comment from Referee 1, we are providing a new table comparing our water use estimates for the near-present period with independent estimates, in which distinctions are made between, sectors, consumptive and non-consumptive uses, as well as a comparison with blue water withdrawals (excluding green water use). Please see R6 to RC1.

5) The paper does not provide any indication how the threshold for WSI (e.g. what WSI should not be exceeded) should be determined. On what basis should stakeholder agree on such a threshold (such as 40%)? I suggest that the threshold should be be based (at least partially) on environmental flow requirements, possibly with higher eflows than today (see Alvarez-Garreton et al. 2023a).

R5: We have addressed this point in relation to environmental flow in R1 and R4.

6) Revise section 6, discussing how to achieve a (basin-specific?) threshold for WSI and also include the constraints of the study (or maybe elsewhere).

R6: Determining a basin-specific threshold for WSI—or any other index—it is beyond the scope of this study. While we are aware of the uncertainties discussed in previous responses, the results presented in this paper represent a first attempt to quantify water stress and its change over a long time period using novel datasets of water availability and use across all watersheds in Chile.

We argue that decisions regarding which metric to implement and what specific thresholds to adopt fall within the domain of public policy and depend on the commitment of authorities to achieve water security goals. The purpose of Section 6 is to illustrate the importance of using quantitative indicators—such as the WSI—to define water security targets and support the design of public policies. At present, no such goals exist, and the baseline regulation (regarding the protection of ecological flow) allows water use equivalent to a WSI of 80% (Alvarez-Garreton et al., 2023a). When an index like the WSI is used with a predefined target (the 40% threshold is used as reference there), there are two main strategies to reach that goal: reducing water use or increasing water availability through alternative sources. The example presented in Section 6 is intended to illustrate this concept.

As discussed earlier, the 40% threshold used in our analysis is based on existing literature, which also supports the classification of WSI values into different stress levels and highlights their implications for ecological flows (R2). Nevertheless, our framework allows for the analysis of any threshold defined by public policy, making it adaptable to different management objectives. We acknowledge that a target based solely on the WSI is insufficient, as it does not capture the full complexity of factors involved in water security. However, it offers a straightforward quantification that can be refined (e.g., with watershed-specific thresholds) or complemented by other metrics. We will add clarification on this point in the revised manuscript (please see the proposed changes in R2).

7) In Figure 4 to 8, state very clearly what water uses are included and why.

R7: Along with changes proposed in text, we will clarify figure captions and labels to avoid confusions.

Specific comments:

In the CRWU2 files, I could only find the non-LULUCF values of water withdrawals but not the LULUCF values.

Thanks! - We realised that only the non-land cover water use sectors were uploaded into CR2WU zenodo repository. We will update the repository to make all sectors are available at https://zenodo.org/records/13324235.

Line 45: should rely on

Ok.

Line 71: due to lower

This suggestion is not adopted. This change would modify the intended meaning of the sentence.

Figure 4: Explain more clearly In caption what type of water use is shown

Figure 4 summarizes all water uses computed in this study, including:

- Total water use (both consumptive and non-consumptive, though dominated by the latter; panel a),
- Details of consumptive use (mainly from LULUCF; panel b), and
- A further breakdown of consumptive use in non-LULCC sectors (panel c).

This will be made clearer in the revised manuscript through the proposed text changes, the new table comparing water use estimates (see R6 to RC1), and improved figure captions.

Figure 4 b title does not seem to be correct.

The title is correct. That bar represents consumptive use from various productive sectors, as well as from the drinking water sector. Drinking water is explicitly labeled along with the other consumptive sectors because it is treated differently than other uses—being accounted for as consumptive use, despite potential return flows in some watersheds—as previously discussed (see R1 and R4).

Figure 4d: replace m3/s by area-specifiv values, e.g. mm/yr, as otherwise values for larger polygons are just larger because of larger area. Does it indluce hydroelectric water use? How is hydroelectric water use computed (in 4 a)?

A larger region may potentially support more population and economic activity with associated water demands, but this also depends on many other factors. In fact, in Chile, the most extensive administrative units to the north and south have relatively low water use. After discussion with coauthors, we prefer to retain volumetric fluxes in Figure 4d to ensure consistency with the national totals shown in panels a—c.

"Total water use" in panels a and d includes both consumptive and non-consumptive uses, with the latter largely explained by hydroelectric power generation. Non-consumptive hydroelectric water use is computed similarly to other non-LULUCF sectors, using activity drivers and specific consumption values (Eq. 7). In this case, the drivers are the actual energy generated by power plants, and the specific consumption values depend on the type of plant (see Table C1). As noted in the response to the first comment, we also account for consumptive use due to reservoir evaporation. This component is automatically estimated based on changes in water body surface area captured in land use maps, and is therefore classified under the LULUCF sector.

Figure 5: Add line for population development.

R: It's a good idea, and we will consider it in a revised version.

Line 393: more water use by increase actual evapotranspiration (add for clarity)

R: We will change this sentence to "The forestry industry, which expanded primarily between the 1970s and 2000s, has significantly contributed to increased water use (approximately 80 m³/s) and heightened pressure on water resources in watersheds. This is due to increased evapotranspiration in intensive Pinus radiata and Eucalyptus plantations."

Line 404: Explain Water Use Rights

R: We will add the following explanation in the revised introduction:

(...). Water scarcity issues have also been attributed to limitations in the water management system defined in the Chilean Water Code (Congreso Nacional de Chile, 2022). This system is based on Water Use Rights (WURs), the legal entitlements that define the amount and timing of water access for consumptive uses (e.g., drinking water, irrigation) and non-consumptive uses (e.g., hydroelectricity). A major limitation of this allocation scheme is that it does not account for climate variability and decreasing water availability (Alvarez-Garreton et al., 2023a; Barría et al., 2021a). (...)

Figure 6: Also show WSI for basins north of 30° S, they do have human water use.

R: There are indeed water uses both to the north and south—although fewer—but Figure 6 presents WSI maps. In the arid regions north of 30°S, water availability approaches zero, causing WSI values to tend toward infinity (shown in grey) and lose interpretative value (water needs in these areas are met through non-renewable groundwater and desalinated seawater, which are not assessed in this study). Conversely, in Patagonia, water use is orders of magnitude lower than water availability, resulting in WSI values close to zero. To maintain the focus on central Chile, as defined in the text for this section, we will clarify this in the figure caption and retain the current figure layout.

Revised caption: "Figure 6. Decadal mean Water Stress Index (WSI) for the major watersheds in central Chile between 1960 and 2020. The names of the basins analyzed in further detail are shown in the right-hand panel. The cross indicates the location of Santiago. Note that most watersheds north of the figure domain are arid, so WSI is undefined, while to the south, all watersheds maintain low WSI levels."

Figure 7: Why do you show here U(cons+DW) (and not e.g. mining), in the caption it says just consumptive use?

R: This point is addressed in R1 and R4. In this study, WSI is calculated using consumptive water uses from all sectors, along with DW, which is treated as consumptive albeit known return flows in some watershed. We will clarify the figure label and caption in the revised manuscript.

Line 613: not clear why is a multiplier of all evapotranspiration components.

Since ET components are computed individually, this factor ensures that the total ET flux does not exceed a maximum rate of  $1.2 \cdot ET_o$ . In other words, it prevents the flux from exceeding atmospheric demand. If the sum of the components in brackets in eq. B1 (now denoted as ET\*) exceeds  $1.2 \cdot ET_o$ , the factor a is set to  $1.2 \cdot ET_o / ET^*$ . Otherwise, its value remains 1.

We acknowledge that this explanation was previously unclear, and the value of the factor a will be explicitly indicated in the revised version of the manuscript.

Line 619: delete "a fraction of"

R: Agree. It will be deleted in a revised version.

Eq. B9: exp (-0.6 LAI): add minus sign

R: Thank you for identifying this error; it will be corrected in the revised version.

Table B2: what is "f"?

R: That is a bug in notation, it refers to  $f_{IWB}$  of eq. B1 (activation flag for irrigated areas or water bodies). It will be corrected in the revised version.

Appendix A and C: move to supplement

The referee does not provide a specific reason for moving these appendices to the supplementary material. We assume this is due to their length, in which case we agree.

Accordingly, we have decided to move the 3 existing tables in the appendices to the supplementary material. Given its relevance, only the newly proposed table comparing water use data (R6 to RC1) will be included in the appendix, along with the existing Appendix B.

**References not included in the main text:**

Alcamo, J., Döll, P., Henrichs, T., Kaspar, F., Lehner, B., Rösch, T., and Siebert, S.: Development and testing of the WaterGAP 2 global model of water use and availability, Hydrol. Sci. J., 48, 317–337, https://doi.org/10.1623/hysj.48.3.317.45290, 2003a.

Alcamo, J., Döll, P., Henrichs, T., Kaspar, F., Lehner, B., Rösch, T., and Siebert, S.: Global estimates of water withdrawals and availability under current and future "business-as-usual" conditions, Hydrol. Sci. J., 48, 339–348, https://doi.org/10.1623/hysj.48.3.339.45278, 2003b.

Anderson, T. G., Christie, D. A., Chávez, R. O., Olea, M., and Anchukaitis, K. J.: Spatiotemporal peatland productivity and climate relationships across the western South American Altiplano, J. Geophys. Res.-Biogeo., 126, e2020JG005994, https://doi.org/10.1029/2020JG005994, 2021.

Carrasco-Escaff, T., Rojas, M., Garreaud, R. D., Bozkurt, D., and Schaefer, M.: Climatic control of the surface mass balance of the Patagonian Icefields, The Cryosphere, 17, 1127–1149, https://doi.org/10.5194/tc-17-1127-2023, 2023.

Damkjaer, S. and Taylor, R.: The measurement of water scarcity: defining a meaningful indicator, Ambio, 46, 513–531, https://doi.org/10.1007/s13280-017-0912-z, 2017.

DGA: Aplicación de la metodología de actualización del Balance Hídrico Nacional en las cuencas de las macrozonas norte y centro, SIT N° 435, Ministerio de Obras Públicas, Dirección General de Aguas, Santiago, Chile, 2018.

DGA: Aplicación de la metodología de actualización del Balance Hídrico Nacional en la macrozona sur y parte norte de la macrozona austral, SIT N° 441, Ministerio de Obras Públicas, Dirección General de Aguas, Santiago, Chile, 2019.

FAO: AQUASTAT – FAO's Global Information System on Water and Agriculture, Food and Agriculture Organization of the United Nations, available at: https://www.fao.org/aquastat/en/, last access: 22 March 2025, 2021.

Falkenmark, M., Berntell, A., Jägerskog, A., Lundqvist, J., Matz, M., and Tropp, H.: On the verge of a new water scarcity: a call for good governance and human ingenuity, SIWI Policy Brief, Stockholm International Water Institute, 2007.

Gómez, G., Álvez, A., Castillo, R., Urrea, J., Díaz, L., González-Saldía, R., and Vidal, G.: Contaminación fecal en el borde costero del país, Serie Comunicacional CRHIAM, Centro de Recursos Hídricos para la Agricultura y la Minería (ANID/FONDAP/1523A0001), ISSN 0718-6460, ISSN 0719-3009, available at: https://www.crhiam.cl/publicaciones/seriescomunicacionales/, 2025.

Kuzma, S., Bierkens, M. F. P., Lakshman, S., Luo, T., Saccoccia, L., Sutanudjaja, E. H., and Van Beek, R.: Aqueduct 4.0: Updated decision-relevant global water risk indicators, Tech. Note, World Resources Institute, Washington, DC, https://doi.org/10.46830/writn.23.00061, 2023.

Liu, W., Liu, X., Yang, H., Ciais, P., and Wada, Y.: Global water scarcity assessment incorporating green water in crop production, Water Resour. Res., 58, e2020WR028570, https://doi.org/10.1029/2020WR028570, 2022.

Melo, D. C. D., Anache, J. A. A., Borges, V. P., Miralles, D. G., Martens, B., Fisher, J. B., et al.: Are remote sensing evapotranspiration models reliable across South American ecoregions?, Water Resour. Res., 57, e2020WR028752, https://doi.org/10.1029/2020WR028752, 2021.

MOP: Ley 21075. Regula la recolección, reutilización y disposición de aguas grises, Ministry of Public Works, Santiago, Chile, available at: https://bcn.cl/G77KeN, 2023.

Moletto-Lobos, I., Mattar, C., and Barichivich, J.: Performance of satellite-based evapotranspiration models in temperate pastures of southern Chile, Water, 12, 3587, https://doi.org/10.3390/w12123587, 2020.

Munia, H., Guillaume, J. H. A., Mirumachi, N., Porkka, M., Wada, Y., and Kummu, M.: Water stress in global transboundary river basins: significance of upstream water use on downstream stress, Environ. Res. Lett., 11, 014002, https://doi.org/10.1088/1748-9326/11/1/014002, 2016.

Pastor, A. V., Ludwig, F., Biemans, H., Hoff, H., and Kabat, P.: Accounting for environmental flow requirements in global water assessments, Hydrol. Earth Syst. Sci., 18, 5041–5059, https://doi.org/10.5194/hess-18-5041-2014, 2014.

Rijsberman, F. R.: Water scarcity: fact or fiction?, Agric. Water Manage., 80, 5–22, https://doi.org/10.1016/j.agwat.2005.07.001, 2006.

Smakhtin, V.: Basin closure and environmental flow requirements, Int. J. Water Resour. Dev., 24, 227–233, https://doi.org/10.1080/07900620701723729, 2008.

Wada, Y., van Beek, L. P. H., Viviroli, D., Dürr, H. H., Weingartner, R., and Bierkens, M. F. P.: Global monthly water stress: 2. Water demand and severity of water stress, Water Resour. Res., 47, W07518, https://doi.org/10.1029/2010WR009792, 2011a.

Wada, Y., van Beek, L. P. H., and Bierkens, M. F. P.: Modelling global water stress of the recent past: on the relative importance of trends in water demand and climate variability, Hydrol. Earth Syst. Sci., 15, 3785–3808, https://doi.org/10.5194/hess-15-3785-2011, 2011b.

---

## Author Response (AR2)

**EGUSPHERE-2024-2695. Responses to Referee Comments**

This document includes responses (in blue) to comments from referee 1 (in black).

RC1

**Overall Review**

The authors have improved the clarity of presentation by addressing the reviewers' comments in mostly suitable ways. However, a few things still need improvement. Line numbers refer to the track change document. A major source of confusion throughout the manuscript remains: the terminology for the different types of water use is not explained clearly enough, and sometimes, the terms are still used incorrectly.

1. I suggest providing, in the introduction, the exact definition of consumptive use (the amount of water evapotranspirated during use), non-consumptive use, and water abstractions/withdrawals, the sum of consumptive and non-consumptive use. "Water use" and "water demand" are only loosely defined and difficult to quantify, and should only be used in this loose or overall meaning. And please provide the definition of the latter two terms, too, as most readers might not be familiar with the exact meaning of these terms.

**We agreed and have added the following text to the revised introduction:**

"Water use estimates typically rely on national inventories, which are often incomplete, unavailable, or even non-existent. As a result, these datasets can carry large uncertainties in some regions, as noted by Wada and Bierkens (2014) for South America. Depending on usage characteristics within a basin, water use can be classified as consumptive, when the water is extracted and not returned to the system (e.g., due to increased evapotranspiration in agriculture or other activities), or as non-consumptive, when the withdrawn water is returned after use and thus does not significantly affect the basin's overall water balance (e.g., hydroelectric power generation)."

2. L317. Revise the newly introduced sentence as you describe later that water abstractions are taken into account, and, from the values in Table S3, I deduce that also livestock water use is rather abstractions than consumptive use. In Table S3, correctly identify which values refer to consumptive use and which to abstraction (Drinking water and livestock). Do not use the term consumption rates, but specify as abstractions or consumptive use.

As we explained in the previous revision and clarified further in the revised text, our water uses estimates are considered consumptive for most secondary sectors (manufacturing industries, mining, and livestock), following the criteria and methods of DGA (2017). In the case of drinking water, we provide a global estimate (i.e., total abstractions), while for LULUCF and energy, we account for both consumptive and non-consumptive uses.

We have added an additional column to Table S3 to clarify the nature of water use.

3. Clarify that the often-used WSI value for water stress of 0.4 refers to water abstractions, not consumptive use.

We modified the following paragraph in the revised Sect. 2.6, L373:

"The WSI, here defined as a water demand-to-availability ratio (Falkenmark and Lundqvist, 1998), depends on how both components are quantified, potentially leading to different outcomes (Liu et al., 2017a). Most previous studies using this index rely on water use estimates based on gross demand (as a proxy for total withdrawals, e.g., Vörösmarty et al., 2000; Oki and Kanae, 2006; Kuzma et al.,

2023), while others consider net demand (equivalent to consumptive use, e.g., Wada et al., 2011), or assess both measures (e.g., Munia et al., 2016). Using total withdrawals in basins with large return flows—such as in Chile, where many watersheds are modified for hydroelectric power generation—results in very high WSI values that do not necessarily reflect actual water stress. For this reason, and to preserve a water balance rationale, only consumptive uses are considered in the WSI calculation in this study."

4. In the new Table B1, FAO values may be wrongly listed as consumptive uses, as, in my understanding, FAO only provides withdrawals for the sectors domestic, industrial, and irrigation.

Yes, FAO-AQUASTAT variable names explicitly indicate that the estimates correspond to withdrawals. Accordingly, we will move them to the withdrawal row in Table B1, while maintaining drinking water as an exception, as it is considered alongside consumptive use from other sectors.

5. While it is now explained that domestic water use refers to abstractions and not consumptive use, the reasons for including abstractions and not consumptive use in the case of domestic water use, I would contend that the given arguments (no reuse possibility due to coastal discharge and laws against reuse) are also true for manufacturing and mining use.

Please refer to our response to comment #2.

State more clearly early in the manuscript that LULUCF also includes the evaporation from artificial reservoirs, and that "energy" water use refers to the cooling of thermal power plants (if correct).

This is already explained in Section 2.5 and highlighted in Section 4. To clarify further, we have modified the following paragraph in Section 2.5 (line 333):

"A third simulation was performed in the same way as  $ET_{FULL}$  but without irrigation ( $ET_{NI}$ ), allowing the estimation of the ET change component driven by rainfed agriculture and forestry (the 'green' water use), as well as by modifications in water bodies—(including hydroelectric reservoirs). We note that the latter includes all types of artificial reservoirs, so consumptive water use due to evaporation losses from hydroelectric reservoirs is included within LULUCF, rather than under the energy sector."

The energy sector does not refer exclusively to thermal power plants (despite the fact that consumptive use in this sector is dominated by thermoelectric facilities, as shown in Fig. 4), but includes all types of energy production, including hydroelectric (the non-consumptive component), thermal, and non-conventional renewable energy (NCRE) sources. The less intuitive choice is that evaporation losses from reservoirs are accounted for within the LULUCC sector (which is also true), because these water bodies are explicitly represented in our land cover maps.

It would be interesting to know which fraction artificial reservoirs contribute to human water use or LULUCF use in Chile; could this be distinguished, e.g., in Fig. 4b?

This would indeed be an interesting analysis for some regions with large reservoirs, and could be estimated using our methodology. However, it would require a specific simulation—one that isolates land cover changes associated only with artificial water bodies—which falls outside the scope of the present study.

Explain, at the latest in section 2.5 but better in the introduction, why you include, in total water use (line 5ß6, Fig. 4a), only the non-consumptive uses of hydro-electricity and irrigation, but not of cooling of thermal power plants, manufacturing and mining, which also should have "major return flows" (line 317 ff).

Please refer to our response to comment #2. In our view, this work represents a substantial effort to estimate dynamic water use across Chile. As with other similar datasets, further refinements could be made to incorporate more detail. However, non-consumptive water use from the additional sectors mentioned (e.g., thermal power plant cooling, manufacturing, and mining) likely accounts for less than 1% of total water use. Neither we account for non-consumptive use from other sectors—such as aquaculture—as their contribution is not expected to significantly affect water stress in Chile.

L549: Also discuss the large discrepancies between the F. Chile (2018) and your estimates for irrigation consumptive and non-consumptive uses, which indicate large uncertainties of irrigation water use efficiencies. Explain whether and why the estimates of your study are more realistic than the values of F. Chile.

The differences in the estimates may arise from a range of factors, including the climate data, land cover data, and crop coefficients used in both studies. The scope of this work does not include a thorough comparative analysis and the attribution of causes for these differences, but rather presents them in broad terms to illustrate the uncertainty, as kindly suggested by a referee in the previous stage.

Unlike the other available studies presented in Table B1, the estimates from our study provide historical data over time, using a consistent methodology in terms of climate, soil moisture, and land cover changes. This represents a comparative advantage and helps fill a critical information gap in Chile.

Check links. For example, links to COCHILCO references do not work anymore.

Thank you. These web addresses have changed since the initial stage of the study. We have now updated the links accordingly.

Check for typos: The revisions introduced new typos, e.g., in lines 15 and 58. In Line 309, the reference to Table B1 seems incorrect.

Typo in L15 corrected. We have not found a typo in L58. Reference to table corrected. Thanks.